# Multinoulli Extension: A Lossless Yet Effective Probabilistic Framework for Subset Selection over Partition Constraints

Qixin Zhang [1]   Wei Huang [2]   Can Jin [3]   Puning Zhao [4]   Yao Shu [5]   Li Shen [4]   Dacheng Tao [1]

## Abstract

Identifying the most representative subset for a *close-to-submodular* objective while satisfying the predefined partition constraint is a fundamental task with numerous applications in machine learning. However, the existing distorted local-search methods are often hindered by their prohibitive query complexities and the rigid requirement for prior knowledge of difficult-to-obtain structural parameters. To overcome these limitations, we introduce a novel algorithm titled **Multinoulli-SCG**, which not only is parameter-free, but also can achieve the same approximation guarantees as the distorted local-search methods with significantly fewer function evaluations. The core of our **Multinoulli-SCG** algorithm is an innovative continuous-relaxation framework named Multinoulli Extension(`ME`), which can effectively convert the discrete subset selection problem subject to partition constraints into a solvable continuous maximization focused on learning the optimal multinoulli priors across the considered partition. In sharp contrast with the well-established multilinear extension for submodular subset selection, a notable advantage of our proposed `ME` is its intrinsic capacity to provide a lossless rounding scheme for any set function. Finally, we validate the practical efficacy of our proposed algorithms by applying them to video summarization, A-optimal design and coverage maximization.

## 1. Introduction

Subset selection aims to identify a small group of representative items from a vast ground set, which finds numerous real-world applications in the fields of machine learning, operations research and statistics, including feature selection (Das & Kempe, 2008; 2011; Qian et al., 2015; 2017), data summarizarion (Lin & Bilmes, 2010; 2011; Wei et al., 2015; Mirzasoleiman et al., 2016), product marketing (Kempe et al., 2003; Tang et al., 2018; Han et al., 2021), sensor placement (Krause et al., 2008; Hashemi et al., 2019; DeValve et al., 2023) and in-context learning (Kumari et al., 2024a;b; Fan et al., 2025). Beyond the aforementioned representational capacity, ensuring the diversity and fairness of the chosen subset is of significant importance. For instance, in various marketing scenarios, it is essential to equitably allocate free products across different communities (Tsang et al., 2019). To this end, partition constraints are often imposed in the process of subset selection, which involves dividing the entire set into non-overlapping sub-classes and then fairly distributing the total budget among them. Motivated by these findings, this paper explores the subset selection problem under partition constraints.

Broadly speaking, the subset selection problem is **NP**-hard (Natarajan, 1995; Feige, 1998), implying that no polynomial-time algorithms can solve it optimally. In light of this hurdle, many studies have focused on designing efficient approximation algorithms to address the subset selection problem. Especially when the utility function associated with the subset selection problem is submodular, a plethora of effective and practical algorithms have been proposed for maximizing this type of functions subject to partition constraints (Fisher et al., 1978; Calinescu et al., 2011; Filmus & Ward, 2012b; 2014). Additionally, it has been frequently observed that there are also many scenarios inducing utility functions that are "*close-to-submodular*", but not strictly submodular. Examples include variable selection for regression (Das & Kempe, 2018; Elenberg et al., 2018), video summarization (Chen et al., 2018a), neural network pruning (El Halabi et al., 2022) and sparse optimal transport (Manupriya et al., 2024).

Compared to the extensive literature on submodular functions, there is a limited amount of research exploring the

[1] Nanyang Technological University, Singapore [2] RIKEN Center for Advanced Intelligence Project(AIP) [3] Rutgers University [4] Shenzhen Campus of Sun Yat-sen University [5] Hong Kong University of Science and Technology (Guangzhou). Email to: Qixin Zhang <qixin.zhang@ntu.edu.sg>  Correspondence to: Li Shen <mathshenli@gmail.com>

*Proceedings of the 42nd International Conference on Machine Learning*, Vancouver, Canada. PMLR 267, 2025. Copyright 2025 by the author(s).

maximization of "*close-to-submodular*" objectives under partition constraints. Notably, Thiery & Ward (2022) recently proposed a distorted local-search algorithm to maximize an important class of "*close-to-submodular*" functions named $(\gamma, \beta)$-weakly submodular functions and demonstrated that this approach can secure a $\frac{\gamma^2(1-e^{-(\beta(1-\gamma)+\gamma^2)})}{\beta(1-\gamma)+\gamma^2}$-approximation under partition constraints, where $\gamma$ and $\beta$ represent the lower and upper submodularity ratio respectively. Subsequently, Lu et al. (2022) extended this local search to another class of "*close-to-submodular*" functions known as $\alpha$-weakly DR-submodular functions and also confirmed a $(1-e^{-\alpha})$-approximation guarantee under partition constraints, where $\alpha$ is the diminishing-return(DR) ratio.

Despite the superior theoretical guarantees of distorted local-search methods, their practical implementation often faces two significant challenges: **i) Reliance on Unknown Parameters:** Distorted local search generally requires prior knowledge of specific structural parameters regarding the objective functions, such as the submodularity ratio and diminishing-return(DR) ratio. However, in practice, accurately estimating these parameters can incur exponential computations. **ii) Prohibitive Query Complexity:** Due to the absence of necessary structural parameters, Lu et al. (2022) and Thiery & Ward (2022) have to adopt a brute-force $\mathcal{O}(1/\epsilon)$-round guesses of these unknown parameters to approximate the distorted local-search methods, which will result in an extremely high $\tilde{\mathcal{O}}(1/\epsilon^6)$ and $\tilde{\mathcal{O}}(1/\epsilon^3)$ number of value queries to the objective function, respectively. Thus, a natural question arises:

> Is it possible to develop a parameter-free and query-efficient algorithm for the "*close-to-submodular*" subset selection problems under partition constraints while keeping strong approximation guarantees?

In this paper, we will provide an affirmative answer to this question by presenting an effective algorithm titled **Multinoulli-SCG**, which not only reduces the strict requirement for the exact knowledge of both submodularity ratio and DR ratio, but also can attain the same approximation guarantees as the aforementioned distorted local-search methods with only $O(1/\epsilon^2)$ function evaluations. The cornerstone of our **Multinoulli-SCG** algorithm is an innovative continuous-relaxation framework termed as the Multinoulli Extension(ME), which aims to learn a multinoulli distribution for each community within the partition constraints and subsequently leverage these distributions to make selection. In sharp contrast with the well-established multi-linear extension (Calinescu et al., 2011), a notable advantage of our proposed ME is its inherent capability to provide a lossless rounding scheme for any set function. Instead, all known lossless rounding schemes for multi-linear extension require

*Table 1.* Comparison of theoretical guarantees for $\alpha$-weakly DR-submodular maximization over partition constraints. Note that '**Para-free**' indicates whether the method does not rely on prior knowledge of $\alpha$, 'OPT' represents the optimal value of the subset selection problem (1), 'Distorted-LS' is the abbreviation for the distorted local-search method, 'Distorted-LS-Guessing' denotes the distorted local-search method with $O(1/\epsilon)$-round guesses, $r$ is the rank of partition constraint, e.g., $r = \sum_{k=1}^{K} B_k$ in problem (1), and $n$ is the size of the ground set, namely, $n = |\mathcal{V}|$ in problem (1).

| Method | Para-free? | Queries | Utility |
|---|---|---|---|
| Standard Greedy (Khashayar & Manuel, 2019) | ✔ | $\mathcal{O}(nr)$ | $\left(\frac{\alpha}{1+\alpha}\right)$OPT |
| Distorted-LS (Lu et al., 2022) | ✘ | $\Omega(nr2^r)$ | $(1-e^{-\alpha})$OPT |
| Distorted-LS-Guessing (Lu et al., 2022) | ✔ | $\tilde{\mathcal{O}}(\frac{nr^4}{\epsilon^6})$ | $(1-e^{-\alpha}-\epsilon)$OPT |
| **Multinoulli-SCG** (Theorem 4&Remark 8) | ✔ | $\mathcal{O}(\frac{r^3n^2}{\epsilon^2})$ | $(1-e^{-\alpha})$OPT$-\epsilon$ |

that the objective set function is *submodular*.

**Our Contributions. i):** This paper introduces a novel probabilistic framework for the subset selection problem under partition constraints, which we refer to as the Multinoulli Extension(ME). Furthermore, we conduct an in-depth exploration of the differentiability, smoothness and monotonicity regarding the ME. More importantly, we establish an upper bound for the gap between the function value of our proposed ME and that of the original set function. **ii):** We propose a novel algorithm named **Multinoulli-SCG**, which effectively integrates the concept of continuous greedy, the path-integrated differential estimator and the relationship between our proposed ME and its original set function. Moreover, we prove that, when the objective function is monotone $\alpha$-weakly DR-submodular or $(\gamma, \beta)$-weakly submodular, our **Multinoulli-SCG** algorithm can attain a value of $(1 - e^{-\alpha})$OPT $- \epsilon$ or $\left(\frac{\gamma^2(1-e^{-(\beta(1-\gamma)+\gamma^2)})}{\beta(1-\gamma)+\gamma^2}\right)$OPT $- \epsilon$ with only $O(1/\epsilon^2)$ function evaluations, where OPT denotes the optimal value. These results not only significantly improve the previous $\tilde{\mathcal{O}}(1/\epsilon^6)$ and $\tilde{\mathcal{O}}(1/\epsilon^3)$ number of function evaluations associated with the distorted local-search methods, but also match the information-theoretic $\mathcal{O}(1/\epsilon^2)$ lower bound (Karbasi et al., 2019; Hassani et al., 2020). **iii):** We demonstrate the practical efficacy of our proposed algorithms by applying them to video summarization, bayesian A-optimal design and maximum coverage.

**Related Work.** Due to space limits, we primarily focus on the most relevant studies, with a more comprehensive discussion provided in Appendix A. Chen et al. (2018a) was the first to investigate weakly submodular maximization beyond simple cardinality constraints, which pointed out that the Residual Random Greedy method of (Buchbinder et al., 2014) can achieve an approximation ratio of $\frac{\gamma^2}{(1+\gamma)^2}$ for the problem of maximizing a monotone $\gamma$-weakly sub-

*Table 2.* Comparison of theoretical guarantees for $(\gamma, \beta)$-weakly submodular maximization over partition constraints. Note that $\phi(\gamma, \beta) = \frac{\gamma^2(1-e^{-(\beta(1-\gamma)+\gamma^2)})}{\beta(1-\gamma)+\gamma^2}$ and '**Para-free**' means whether the method does not rely on prior knowledge of $\gamma$ and $\beta$

| Method | Para-free? | Queries | Utility |
|---|---|---|---|
| Residual Random Greedy (Chen et al., 2018a) | ✔ | $\mathcal{O}(nr)$ | $\left(\frac{\gamma^2}{(1+\gamma)^2}\right)$OPT |
| Standard Greedy (Khashayar & Manuel, 2019) | ✔ | $\mathcal{O}(nr)$ | $\left(\frac{0.4\gamma^2}{\sqrt{\gamma r}+1}\right)$OPT |
| Residual Random Greedy (Thiery & Ward, 2022) | ✔ | $\mathcal{O}(nr)$ | $\left(\frac{\gamma}{\gamma+\beta}\right)$OPT |
| Distorted-LS (Thiery & Ward, 2022) | ✘ | $\Omega(nr2^r)$ | $\phi(\gamma, \beta)$OPT |
| Distorted-LS-Guessing (Thiery & Ward, 2022) | ✔ | $\tilde{\mathcal{O}}(\frac{nr^4}{\epsilon^3})$ | $(\phi(\gamma, \beta) - \epsilon)$OPT |
| **Multinoulli-SCG** (Theorem 4&Remark 8) | ✔ | $\mathcal{O}(\frac{r^3 n^2}{\epsilon^2})$ | $\phi(\gamma, \beta)$OPT$-\epsilon$ |

modular functions subject to a matroid constraint. Note that matroid constraint is a natural generalization of the partition constraints considered in this paper. Subsequently, Khashayar & Manuel (2019) examined the approximation performance of standard greedy algorithm on both $\gamma$-weakly submodular and $\alpha$-weakly DR-submodular maximization over a matroid constraint. Next, in order to improve the approximation performance of the Residual Random Greedy method, Thiery & Ward (2022) introduced the notion of upper submodularity ratio $\beta$ and developed a more powerful distorted local-search algorithm for $(\gamma, \beta)$-weakly submodular maximization. Concurrently, Lu et al. (2022) also proposed a similar local-search method to maximize $\alpha$-weakly DR-submodular functions. A detailed comparison of our proposed **Multinoulli-SCG** algorithm with existing studies is presented in Table 1 and Table 2.

**Remark on Table 2**: Thiery & Ward (2022) has demonstrated that when $\gamma < \frac{1}{7}$, the approximation guarantee of Residual Random Greedy method (Chen et al., 2018a), namely $\frac{\gamma^2}{(1+\gamma)^2}$, will surpass the ratio $\phi(\gamma, \beta)$. To overcome this drawback, Thiery & Ward (2022) initializes their distorted local-search method by the returned subset of the Residual Random Greedy method. Similarly, when $\gamma < \frac{1}{7}$, we also can produce a better subset by comparing the returned subset of our proposed **Multinoulli-SCG** algorithm with that of the Residual Random Greedy method.

## 2. Preliminaries

In this section we present several important notations and concepts that we will frequently use throughout this paper.

**Notations:** For any positive integer $K$, $[K]$ stands for the set $\{1, \ldots, K\}$. The symbol $\langle \cdot, \cdot \rangle$ denotes the inner product. Moreover, $\Delta_m$ represents the standard $m$-dimensional simplex, i.e., the set $\{(x_1, \ldots, x_m) | \sum_{i=1}^m x_i \leq 1 \text{ and } x_i \geq$

$0, \forall i \in [m]\}$. Especially, the symbol 'Multi(**p**)' denotes a multinoulli distribution with $(m + 1)$ possible states where $\mathbf{p} \in \Delta_m$. Note that the multinoulli distribution is also known as the categorical distribution (Murphy, 2012).

**Partition of A Set:** Given a finite ground set $\mathcal{V}$, we say $\{\mathcal{V}_1, \ldots, \mathcal{V}_K\}$ is a partition of set $\mathcal{V}$ if and only if i) $\mathcal{V}_i \cap \mathcal{V}_j = \emptyset$ for any $i \neq j \in [K]$; ii) $\mathcal{V} = \bigcup_{k=1}^K \mathcal{V}_k$.

**Subset Selection under Partition Constraints:** Let $f : 2^{\mathcal{V}} \to \mathbb{R}_+$ be a set function that maps any subset of $\mathcal{V}$ to a non-negative utility. Given a partition $\{\mathcal{V}_1, \ldots, \mathcal{V}_K\}$ of $\mathcal{V}$ and a collection of budgets $\{B_1, \ldots, B_K\}$ where $0 < B_k \leq |\mathcal{V}_k| \ \forall k \in [K]$, the goal of the subset selection problems subject to partition constraints is aimed at finding a subset $S$ from $\mathcal{V}$ such that the utility set function $f$ is maximized within the constraints $|S \cap \mathcal{V}_k| \leq B_k$ for any $k \in [K]$, i.e.,

$$\max_{S \subseteq \mathcal{V}} f(S) \ \text{s.t.} \ |S \cap \mathcal{V}_k| \leq B_k \ \forall k \in [K]. \quad (1)$$

**Monotonicity:** We say that a set function $f : 2^{\mathcal{V}} \to \mathbb{R}_+$ is *monotone* if and only if $f(A) \leq f(B)$ for any $A \subseteq B \subseteq \mathcal{V}$.

**Weak Submodularity:** Given a set function $f : 2^{\mathcal{V}} \to \mathbb{R}_+$ and any two subsets $A, B \subseteq \mathcal{V}$, we denote by $f(B|A)$ the marginal contribution of adding the elements of $B$ to $A$, i.e., $f(B|A) := f(A \cup B) - f(A)$. For simplicity, when $B$ is a singleton set $\{v\}$, we also use $f(v|A)$ to represent $f(\{v\}|A)$. Therefore, we say that a set function $f : 2^{\mathcal{V}} \to \mathbb{R}_+$ is $\gamma$-*weakly submodular from below* for some $\gamma \in (0, 1]$ if and only if, for any two subsets $A \subseteq B \subseteq \mathcal{V}$,

$$\sum_{v \in B \setminus A} f(v|A) \geq \gamma\Big(f(B) - f(A)\Big), \quad (2)$$

where we denote $\gamma$ as the *lower submodularity ratio*. Similarly, we also can define the *weak submodularity from above*, that is, a set function $f : 2^{\mathcal{V}} \to \mathbb{R}_+$ is $\beta$-*weakly submodular from above* for some $\beta \geq 1$ iff, $\forall A \subseteq B \subseteq \mathcal{V}$,

$$\sum_{v \in B \setminus A} f(v|B - \{v\}) \leq \beta\Big(f(B) - f(A)\Big), \quad (3)$$

where $\beta$ is called as the *upper submodularity ratio*. When a set function $f$ satisfies both Eq.(2) and Eq.(3), we say it is $(\gamma, \beta)$-*weakly submodular* (Thiery & Ward, 2022).

**Weak DR-submodularity:** A set function $f : 2^{\mathcal{V}} \to \mathbb{R}_+$ is $\alpha$-*weakly DR-submodular* for some $\alpha \in (0, 1]$ iff $f(v|A) \geq \alpha f(v|B)$ for any two subsets $A \subseteq B \subseteq \mathcal{V}$ and $v \in \mathcal{V} \setminus B$. In particular, $\alpha$ is often called as the diminishing-return(DR) ratio (Kuhnle et al., 2018). Note that, from Eq.(2) and Eq.(3), we can infer that an $\alpha$-weakly DR-submodular function automatically satisfies the conditions for being $(\alpha, \frac{1}{\alpha})$-weakly submodular. Moreover, when $\alpha = 1$, weakly DR-submodular objectives will reduce to the standard submodular functions (Nemhauser et al., 1978; Fujishige, 2005)

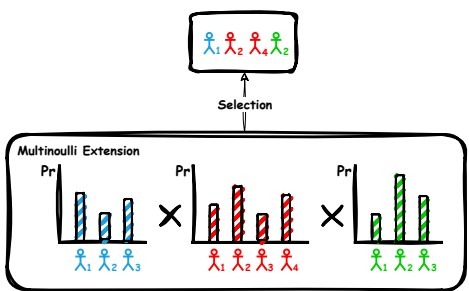

*Figure 1.* Diagram of Multinoulli Extension.

## 3. Multinoulli Extension

Generally speaking, the discrete nature of subset selection problem (1) poses a significant challenge in finding effective solutions. In recent years, compared to discrete optimization, continuous optimization developed an array of efficient and advanced algorithmic tools. Thus, an alternative strategy to address the subset selection problem (1) is to bring it into the world of continuous optimization via *relaxation-rounding* frameworks, which typically involve three critical stages: first, converting the problem (1) into a solvable continuous optimization; second, applying the gradient-based methods to output a high-quality continuous solution; and third, rounding the previous continuous solution back to the partition constraint of Eq.(1) without any loss in terms of the function value. In the subsequent part of this section, we will present a novel relaxation-rounding framework named the Multinoulli Extension(ME) for problem (1).

Prior to this, Calinescu et al. (2011) proposed a continuous relaxation technique known as the multi-linear extension for *submodular* subset selection problems. Unfortunately, this extension cannot be directly applied to the general subset selection problem (1) because most known lossless rounding schemes for multi-linear extension, such as pipage rounding (Ageev & Sviridenko, 2004), swap rounding (Chekuri et al., 2010) and contention resolution (Chekuri et al., 2014), are heavily dependent on the *submodular* assumption. Up to now, how to losslessly round the multi-linear extension of *non-submodular* set functions, e.g. $(\gamma, \beta)$-weakly submodular and $\alpha$-weakly DR-submodular functions, still remains an open question (Thiery & Ward, 2022). Given the unsolved rounding challenge of multi-linear extension, this paper choose to introduce a new relaxation technique named Multinoulli Extension(ME) to address the problem (1).

To provide a clearer exposition of our proposed ME, we first make some assumptions regarding the problem (1): we define $\mathcal{V}_k := \{v_k^1, \ldots, v_k^{n_k}\}$ for any $k \in [K]$ and set $|\mathcal{V}| = n$, i.e., $n = \sum_{k=1}^{K} n_k$. More specifically, the core idea of our ME is to learn a prior multinoulli distribution 'Multi($\mathbf{p}_k$)' for each community $\mathcal{V}_k$, where $\mathbf{p}_k := (p_k^1, \ldots, p_k^{n_k}) \in \Delta_{n_k}$ and each $p_k^m$ denotes the probability that element $v_k^m$ is

selected within its own community $\mathcal{V}_k$ for any $m \in [n_k]$ and $k \in [K]$. Subsequently, ME employs each prior distribution 'Multi($\mathbf{p}_k$)' to conduct $B_k$ independent random selections for every community $\mathcal{V}_k$, which can ultimately yield a subset that adheres to the partition constraint of problem (1). In Figure 1, we present a three-community example of ME. It is noteworthy that, with the probability $1 - \sum_{m=1}^{n_k} p_k^m$, the multinoulli prior 'Multi($\mathbf{p}_k$)' won't pick any member from $\mathcal{V}_k$. In other words, sometimes we might end up with no selection, i.e., $\emptyset$. Formally, we can define the ME as:

**Definition 1** (Multinoulli Extension). Given a set function $f : 2^{\mathcal{V}} \to \mathbb{R}_+$, its Multinoulli Extension $F : \prod_{k=1}^{K} \Delta_{n_k} \to \mathbb{R}_+$ for problem (1) can be defined as:

$$F(\mathbf{p}_1, \ldots, \mathbf{p}_K) := \mathbb{E}_{e_{\hat{k}}^{\hat{b}} \sim \text{Multi}(\mathbf{p}_{\hat{k}})} \Big( f\big( \cup_{\hat{k}=1}^{K} \cup_{\hat{b}=1}^{B_{\hat{k}}} \{e_{\hat{k}}^{\hat{b}}\} \big) \Big)$$

$$= \sum_{e_{\hat{k}}^{\hat{b}} \in \mathcal{V}_{\hat{k}} \cup \{\emptyset\}} \Big( f\big( \cup_{\hat{k}=1}^{K} \cup_{\hat{b}=1}^{B_{\hat{k}}} \{e_{\hat{k}}^{\hat{b}}\} \big) \prod_{\hat{k}=1}^{K} \prod_{\hat{b}=1}^{B_{\hat{k}}} \Pr(e_{\hat{k}}^{\hat{b}} | \mathbf{p}_{\hat{k}}) \Big),$$

where each $e_{\hat{k}}^{\hat{b}}$ denotes the element chosen at the $\hat{b}$-th random trail of community $\mathcal{V}_{\hat{k}}$, $\forall \hat{b} \in [B_{\hat{k}}], \forall \hat{k} \in [K]$, $\mathbf{p}_{\hat{k}} = (p_{\hat{k}}^1, \ldots, p_{\hat{k}}^{n_{\hat{k}}}) \in \Delta_{n_{\hat{k}}}$ is the probability vector for the community $\mathcal{V}_{\hat{k}}$, $\forall \hat{k} \in [K]$, $\Pr(v_{\hat{k}}^m | \mathbf{p}_{\hat{k}}) = p_{\hat{k}}^m$, $\forall m \in [n_{\hat{k}}], \forall \hat{k} \in [K]$ and $\Pr(\emptyset | \mathbf{p}_{\hat{k}}) = 1 - \sum_{m=1}^{n_{\hat{k}}} p_{\hat{k}}^m$, $\forall \hat{k} \in [K]$.

**Remark 1.** The introduction of ME is aimed at converting the general subset selection problem (1) into a continuous maximization task focused on identifying the optimal multinoulli priors across the partition $\{\mathcal{V}_1, \ldots, \mathcal{V}_K\}$. Specifically, we hope to address the following continuous optimization:

$$\max_{p_k^m \geq 0} F(\mathbf{p}_1, \ldots, \mathbf{p}_K) \text{ s.t.} \sum_{m=1}^{n_k} p_k^m \leq 1, \forall k \in [K]. \quad (4)$$

**Remark 2.** In comparison with the multi-linear extension (Calinescu et al., 2011), a notable advantage of our ME is that it does not assign probabilities to any subsets that are out of the partition constraint of problem (1). This means that, for any set function $f : 2^{\mathcal{V}} \to \mathbb{R}_+$ and any given $(\mathbf{p}_1, \ldots, \mathbf{p}_K) \in \prod_{k=1}^{K} \Delta_{n_k}$, we can, through the definition of ME, easily produce a subset that conforms to the partition constraint of problem (1) without any loss in terms of the expected function value $F(\mathbf{p}_1, \ldots, \mathbf{p}_K)$. For more details about multi-linear extension, please refer to Appendix A.1.

Although the ME is naturally endowed with a lossless rounding scheme for any set function, there are two crucial questions that must be answered in order to leverage this tool to tackle the subset selection problem (1) : **i):** *What is the relationship between the function value of our proposed Multinoulli Extension F and the original set function f?* **ii):** *How to solve the relaxed problem (4)?* For the rest of this section, we will focus on the first question. The exploration of the second question will be presented in Section 4.

### 3.1. The Properties of Multinoulli Extension

In this subsection, we will concentrate on characterizing several important properties about our proposed ME. Specifically, we have the following theorem:

**Theorem 1** (Proof provided in Appendix C.1). *For a set function $f : 2^{\mathcal{V}} \rightarrow \mathbb{R}_+$, its Multinoulli Extension $F : \prod_{k=1}^{K} \Delta_{n_k} \rightarrow \mathbb{R}_+$ for problem* (1)*, as described in the Definition 1, satisfies the following properties:*

*1): The first-order partial derivative of $F$ at any point $(\mathbf{p}_1, \ldots, \mathbf{p}_K) \in \prod_{k=1}^{K} \Delta_{n_k}$ can be written as follows:*

$$\frac{\partial F}{\partial p_k^m} := B_k \left( \mathbb{E}_{e_{\hat{k}}^{\hat{b}} \sim \text{Multi}(\mathbf{p}_{\hat{k}})} \left( f\left( v_k^m \Big| \cup_{(\hat{k},\hat{b}) \neq (k,1)} \{e_{\hat{k}}^{\hat{b}}\} \right) \right) \right),$$

*for any $k \in [K]$ and $m \in [n_k]$;*

*2): If $f$ is monotone, then $\frac{\partial F}{\partial p_k^m} \geq 0, \forall k \in [K], m \in [n_k]$;*

*3): If $f$ is $\alpha$-weakly DR-submodular, then $F$ is $\alpha$-weakly continuous DR-submodular (Hassani et al., 2017; Zhang et al., 2022; 2024) over the domain $\prod_{k=1}^{K} \Delta_{n_k}$, that is, for any two point $(\mathbf{p}_1, \ldots, \mathbf{p}_K) \in \prod_{k=1}^{K} \Delta_{n_k}$ and $(\hat{\mathbf{p}}_1, \ldots, \hat{\mathbf{p}}_K) \in \prod_{k=1}^{K} \Delta_{n_k}$, if $\hat{\mathbf{p}}_k \geq \mathbf{p}_k$ for any $k \in [K]$, we have that*

$$\nabla F(\mathbf{p}_1, \ldots, \mathbf{p}_K) \geq \alpha \nabla F(\hat{\mathbf{p}}_1, \ldots, \hat{\mathbf{p}}_K);$$

*4): If $f$ is $\gamma$-weakly submodular from below, then $F$ is upper-linearizable (Pedramfar & Aggarwal, 2024) over the domain $\prod_{k=1}^{K} \Delta_{n_k}$, that is, for any two point $\mathbf{x} := (\mathbf{p}_1, \ldots, \mathbf{p}_K) \in \prod_{k=1}^{K} \Delta_{n_k}$ and $\hat{\mathbf{x}} := (\hat{\mathbf{p}}_1, \ldots, \hat{\mathbf{p}}_K) \in \prod_{k=1}^{K} \Delta_{n_k}$, if $\hat{\mathbf{p}}_k \geq \mathbf{p}_k$ for any $k \in [K]$, we have that*

$$\gamma \left( F(\hat{\mathbf{x}}) - F(\mathbf{x}) \right) \leq \left\langle \hat{\mathbf{x}} - \mathbf{x}, \nabla F(\mathbf{x}) \right\rangle.$$

**Remark 3.** The first point of Theorem 1 provides a specific form about the first-order derivative of our proposed ME. The second point indicates that the mononicity of the set function $f$ can be inheritable by its ME. Furthermore, the third and fourth points reveal that when the set function $f$ exhibits the weak DR-submodularity or weak submodularity, its corresponding ME is weakly continuous DR-submodular or upper-linearizable over the domain $\prod_{k=1}^{K} \Delta_{n_k}$.

**Remark 4.** Note that both upper-linearizable and weakly continuous DR-submodular functions defined over the box constraint $[0, 1]^n$ have been extensively studied by (Hassani et al., 2017; 2020; Zhang et al., 2024; Wan et al., 2023; Pedramfar & Aggarwal, 2024). However, it is crucial to emphasize that these former results cannot be directly applied to our ME. This is because all of them require the domain of objective functions to be closed under the coordinate-wise maximum operation $\vee$, i.e., $\mathbf{x} \vee \mathbf{y} = \max(\mathbf{x}, \mathbf{y})$. Unfortunately, the domain of our ME does not meet this requirement. For further details, please refer to Appendix A.3.

Next, we uncover the relationship between the function value of our proposed ME $F$ and that of the original $f$. To be more precise, we have the following theorem:

**Theorem 2** (Proof provided in Appendix C.2). *When the set function $f : 2^{\mathcal{V}} \rightarrow \mathbb{R}_+$ is monotone and $\alpha$-weakly DR-submodular, for any subset $S$ within the partition constraint of problem* (1) *and any point $\mathbf{x} := (\mathbf{p}_1, \ldots, \mathbf{p}_K) \in \prod_{k=1}^{K} \Delta_{n_k}$, the following inequality holds:*

$$\alpha \left( f(S) - F(\mathbf{x}) \right) \leq \left\langle \sum_{k=1}^{K} \frac{1}{B_k} \mathbf{1}_{S \cap \mathcal{V}_k}, \nabla F(\mathbf{x}) \right\rangle,$$

*where the symbol $\mathbf{1}_S$ is the indicator function over the set $S$, meaning that, for any element $v_k^m \in S$, the corresponding coordinate of its probability $p_k^m$ in $\mathbf{1}_S$ is set to 1; otherwise, 0. Similarly, when the set function $f : 2^{\mathcal{V}} \rightarrow \mathbb{R}_+$ is monotone and $(\gamma, \beta)$-weakly submodular, for any subset $S$ within the partition constraint of problem* (1) *and any point $\mathbf{x} := (\mathbf{p}_1, \ldots, \mathbf{p}_K) \in \prod_{k=1}^{K} \Delta_{n_k}$, we can infer that*

$$\gamma^2 f(S) - (\beta(1-\gamma) + \gamma^2) F(\mathbf{x}) \leq \left\langle \sum_{k=1}^{K} \frac{\mathbf{1}_{S \cap \mathcal{V}_k}}{B_k}, \nabla F(\mathbf{x}) \right\rangle.$$

**Remark 5.** Theorem 2 implies that when $f$ is monotone and $\alpha$-weakly DR-submodular, for any $S$ within the partition constraint of problem (1) and any point $\mathbf{x} \in \prod_{k=1}^{K} \Delta_{n_k}$, the discrepancy between $f(S)$ and $F(\mathbf{x})$ can be bounded by the inner product $\left\langle \sum_{k=1}^{K} \frac{1}{B_k} \mathbf{1}_{S \cap \mathcal{V}_i}, \nabla F(\mathbf{x}) \right\rangle$. Similarly, if $f$ is monotone and $(\gamma, \beta)$-weakly submodular, this inner product $\left\langle \sum_{k=1}^{K} \frac{1}{B_k} \mathbf{1}_{S \cap \mathcal{V}_i}, \nabla F(\mathbf{x}) \right\rangle$ also can bound the gap between $\gamma^2 f(S)$ and $(\beta(1 - \gamma) + \gamma^2) F(\mathbf{x})$.

## 4. Approximation Algorithms for Subset Selection over Partition Constraints

In this section, we delve into the development of efficient approximation algorithms for the subset selection problem (1) based on our introduced Multinoulli Extension $F$.

### 4.1. Stochastic Variant of Continuous Greedy Algorithm

This subsection aims to present an effective method named **Multinoulli-SCG** to maximize our introduced ME, which is primarily inspired by the *continuous greedy*(CG) algorithm for the multi-linear extension (Calinescu et al., 2011; Bian et al., 2017b; 2020; Mokhtari et al., 2018a; 2020).

The CG algorithm typically comprises two critical steps: First, it begins with $\mathbf{x}(1) = \mathbf{0}$ and then, at each iteration $t \in [T]$, the algorithm identifies the optimal ascent direction $\mathbf{v}(t) = \arg\max_{\mathbf{v} \in \mathcal{C}} \left\langle \mathbf{v}, \nabla G(\mathbf{x}(t)) \right\rangle$ to update the current variable $\mathbf{x}(t)$ according to the rule $\mathbf{x}(t+1) = \mathbf{x}(t) + \frac{1}{T} \mathbf{v}(t)$ where $G$ is the multi-linear extension of a set function

---

**Algorithm 1** Stochastic Continuous Greedy Algorithm for Multinoulli Extension(**Multinoulli-SCG**)

---

**Input:** Batch size $L$, number of iterations $T$, set function $f$ and partition $\{\mathcal{V}_1, \ldots, \mathcal{V}_K\}$ of set $\mathcal{V}$

1: **Initialize:** $\mathbf{x}(1) = (\mathbf{p}_1(1), \ldots, \mathbf{p}_K(1)) = \mathbf{0}$;
2: **for** $t = 1, \ldots, T$ **do**
3:    **if** $t = 1$ **then**   ▷ Differential Estimator(Lines 3-12)
4:       Compute $\mathbf{g}(1) := \nabla F(\mathbf{0})$ based on Theorem 1;
5:    **else**
6:       Sample $\{a(1), \ldots, a(L)\}$ uniformly from $[0, 1]$;
7:       Set $\mathbf{x}_l(t) = a(l)\mathbf{x}(t) + (1 - a(l))\mathbf{x}(t-1), \forall l \in [L]$;
8:       Compute the Hessian estimator $\widehat{\nabla}^2 F(\mathbf{x}_l(t))$ based on Remark 6 for any $l \in [L]$;
9:       Compute $\widehat{\nabla}^2_t := \frac{1}{L} \sum_{l=1}^{L} \widehat{\nabla}^2 F(\mathbf{x}_l(t))$;
10:      Compute $\boldsymbol{\xi}_t := \widehat{\nabla}^2_t(\mathbf{x}(t) - \mathbf{x}(t-1))$;
11:      Aggregate the estimator $\mathbf{g}(t) = \mathbf{g}(t-1) + \boldsymbol{\xi}_t$;
12:   **end if**   ▷ Stochastic Continuous Greedy(Lines13-14)
13:    $S(t) = \arg\max_{|S \cap \mathcal{V}_k| \le B_k} \left\langle \mathbf{g}(t), \sum_{k=1}^{K} \frac{1}{B_k} \mathbf{1}_{S \cap \mathcal{V}_k} \right\rangle$;
14:    Update $\mathbf{x}(t+1) = \mathbf{x}(t) + \frac{1}{T} \sum_{k=1}^{K} \frac{1}{B_k} \mathbf{1}_{S(t) \cap \mathcal{V}_k}$;
15: **end for**
16: Return $S$ by rounding $\mathbf{x}(T+1)$;

---

$f$ and $\mathcal{C}$ is a convex domain. The motivation behind the CG algorithm is that when $f$ is *submodular*, the inequality $\langle \mathbf{y}, \nabla G(\mathbf{x}) \rangle \ge G(\mathbf{y}) - G(\mathbf{x})$ holds for any feasible $\mathbf{x}, \mathbf{y}$. Note that, in Theorem 2, we establish a similar relationship between our proposed ME $F$ and its original set function $f$, that is , the inner product $\left\langle \sum_{k=1}^{K} \frac{1}{B_k} \mathbf{1}_{S \cap \mathcal{V}_i}, \nabla F(\mathbf{x}) \right\rangle$ can bound a specific form of weighted discrepancy between $f(S)$ and $F(\mathbf{x})$, when $f$ is monotone $\alpha$-weakly DR-submodular or $(\gamma, \beta)$-weakly submodular. Motivated by these findings, we naturally consider a two-step variant of CG algorithm to maximize our proposed ME: Initially, we set $\mathbf{x}(1) = \mathbf{0}$, and then, at each iteration $t \in [T]$, we find the optimal subset $S(t) := \arg\max_{|S \cap \mathcal{V}_k| \le B_k, \forall k \in [K]} \left\langle \nabla F(\mathbf{x}(t)), \sum_{k=1}^{K} \frac{1}{B_k} \mathbf{1}_{(S \cap \mathcal{V}_k)} \right\rangle$ to update the current variable $\mathbf{x}(t)$ according to the rule $\mathbf{x}(t+1) := \mathbf{x}(t) + \frac{1}{T} \sum_{k=1}^{K} \frac{1}{B_k} \mathbf{1}_{(S(t) \cap \mathcal{V}_k)}$.

However, the implementation of the aforementioned two-step algorithm requires accurately computing the gradients of our proposed ME $F$, which is typically computationally intensive. To circumvent this obstacle, we adopt the stochastic path-integrated differential estimator (Fang et al., 2018; Yurtsever et al., 2019; Hassani et al., 2020), namely, for a sequence of iterations $\{\mathbf{x}(t)\}_{t=1}^{T+1}$, we estimate each gradient $\nabla F(\mathbf{x}(t))$ by the following path-integral form: $\widetilde{\nabla} F(\mathbf{x}(t)) := \nabla F(\mathbf{x}(1)) + \sum_{s=2}^{t} \boldsymbol{\xi}_s$, where each $\boldsymbol{\xi}_s$ is an unbiased estimator for the difference $\mathbf{Diff}_s := \nabla F(\mathbf{x}(s)) - \nabla F(\mathbf{x}(s-1))$. Note that, for any $2 \le s \le T$, the difference $\mathbf{Diff}_s$ can be rewritten

as: $\mathbf{Diff}_s = \int_0^1 \nabla^2 F(\mathbf{x}^a(s)) \mathrm{d}a \left( \mathbf{x}(s) - \mathbf{x}(s-1) \right)$, where $\mathbf{x}^a(s) = a\mathbf{x}(s) + (1-a)\mathbf{x}(s-1), \forall a \in [0, 1]$ and $\nabla^2 F$ is the Hessian of $F$. Hence, if we uniformly sample the parameter $a$ from $[0, 1]$, the difference $\mathbf{Diff}_s$ can be unbiasedly estimated by $\boldsymbol{\xi}_s := \widetilde{\nabla}^2 F(\mathbf{x}^a(s)) \left( \mathbf{x}(s) - \mathbf{x}(s-1) \right)$ where $\widetilde{\nabla}^2 F$ is an unbiased estimator of $\nabla^2 F$. Following this idea, we proceed to demonstrate how to estimate the second-order derivative of our proposed ME, that is to say,

**Theorem 3** (Proof provided in Appendix C.3). *For a set function $f : 2^{\mathcal{V}} \to \mathbb{R}_+$, the second-order derivative of its Multinoulli Extension $F$ at any point $(\mathbf{p}_1, \ldots, \mathbf{p}_K) \in \prod_{k=1}^{K} \Delta_{n_k}$ can be written as follows:*

*1): If $k_1 \ne k_2 \in [K]$, for any $m_1 \in [n_{k_1}]$ and $m_2 \in [n_{k_2}]$,*

$$\frac{\partial^2 F}{\partial p_{k_1}^{m_1} \partial p_{k_2}^{m_2}} = B_{k_1} B_{k_2} \mathbb{E}\left( f\left(v_{k_1}^{m_1} \Big| S \cup \{v_{k_2}^{m_2}\}\right) - f\left(v_{k_1}^{m_1} \Big| S\right) \right),$$

*where $S = \cup_{(\hat{k}, \hat{b}) \ne \{(k_1, 1), (k_2, 1)\}} \{e_{\hat{k}}^{\hat{b}}\}$ and each $e_{\hat{k}}^{\hat{b}}$ is drawn from the multinoulli distribution Multi($\mathbf{p}_{\hat{k}}$);*

*2): As for $k_1 = k_2 = k \in [K]$, if $B_k = 1$, for any $m_1, m_2 \in [n_k]$, we have $\frac{\partial^2 F}{\partial p_k^{m_1} \partial p_k^{m_2}} = 0$; Moreover, when $B_k \ge 2$,*

$$\frac{\partial^2 F}{\partial p_k^{m_1} \partial p_k^{m_2}} = (B_k^2 - B_k) \mathbb{E}\left( f\left(v_k^{m_1} \Big| S \cup \{v_k^{m_2}\}\right) - f\left(v_k^{m_1} \Big| S\right) \right),$$

*where $S = \cup_{(\hat{k}, \hat{b}) \ne \{(k, 1), (k, 2)\}} \{e_{\hat{k}}^{\hat{b}}\}$ and each $e_{\hat{k}}^{\hat{b}}$ is independently drawn from the multinoulli distribution Multi($\mathbf{p}_{\hat{k}}$).*

**Remark 6.** Theorem 3 provides a detailed characterization of the second-order derivative of our proposed ME $F$, which implies that we can estimate the Hessian of $F$ by sampling a sequence of random elements. Specifically, when each $e_{\hat{k}}^{\hat{b}}$ is independently drawn from the multinoulli distribution 'Multi($\mathbf{p}_{\hat{k}}$)' for any $\hat{k} \in [K]$ and $\hat{b} \in [B_{\hat{k}}]$, we can estimate $\frac{\partial^2 F}{\partial p_{k_1}^{m_1} \partial p_{k_2}^{m_2}}(\mathbf{p}_1, \ldots, \mathbf{p}_K)$ as: when $k_1 \ne k_2 \in [K]$,

$$\widehat{\frac{\partial^2 F}{\partial p_{k_1}^{m_1} \partial p_{k_2}^{m_2}}} = B_{k_1} B_{k_2} \left( f\left(v_{k_1}^{m_1} \Big| S \cup \{v_{k_2}^{m_2}\}\right) - f\left(v_{k_1}^{m_1} \Big| S\right) \right),$$

where $S = \cup_{(\hat{k}, \hat{b}) \ne \{(k_1, 1), (k_2, 1)\}} \{e_{\hat{k}}^{\hat{b}}\}$; As for $k_1 = k_2 = k \in [K]$ and $B_k \ge 2$,

$$\widehat{\frac{\partial^2 F}{\partial p_k^{m_1} \partial p_k^{m_2}}} = (B_k^2 - B_k) \left( f\left(v_k^{m_1} \Big| S \cup \{v_k^{m_2}\}\right) - f\left(v_k^{m_1} \Big| S\right) \right),$$

where $S = \cup_{(\hat{k}, \hat{b}) \ne \{(k, 1), (k, 2)\}} \{e_{\hat{k}}^{\hat{b}}\}$.

Incorporating this second-order approximation and the idea of differential estimator into the previously mentioned two-step algorithm, we can develop a stochastic algorithm for maximizing our ME, as detailed in Algorithm 1.

Furthermore, based on the results of Theorem 1 and Theorem 2, we can show that

**Theorem 4** (Proof provided in Appendix D.1). *When the set function $f : 2^{\mathcal{V}} \to \mathbb{R}_+$ is monotone and $\alpha$-weakly DR-submodular, if we set the batch size $L = \mathcal{O}(T)$, the subset $S$ output by Algorithm 1 satisfies:*

$$\mathbb{E}\big(f(S)\big) \geq \Big(1 - e^{-\alpha}\Big)f(S^*) - \mathcal{O}\Big(\frac{r\sqrt{n}}{T}\Big),$$

*where $S^*$ is the optimal solution of problem (1), $r$ is the rank of partition constraint, i.e., $r = \sum_{k=1}^{K} B_k$ and $n = |\mathcal{V}|$. Similarly, if the set function $f : 2^{\mathcal{V}} \to \mathbb{R}_+$ is $(\gamma, \beta)$-weakly monotone submodular and $L = \mathcal{O}(T)$, the subset $S$ output by Algorithm 1 satisfies:*

$$\mathbb{E}\big(f(S)\big) \geq \Big(\frac{\gamma^2(1 - e^{-(\beta(1-\gamma)+\gamma^2)})}{\beta(1-\gamma)+\gamma^2}\Big)f(S^*) - \mathcal{O}\Big(\frac{r\sqrt{n}}{T}\Big).$$

**Remark 7.** Theorem 4 implies that, when the set function $f$ is $\alpha$-weakly monotone DR-submodular, if we set $T = \mathcal{O}(1/\epsilon)$, the subset yielded by our proposed Algorithm 2 can secure a value of $(1 - e^{-\alpha})\text{OPT} - \epsilon$, where OPT is the maximum value of problem (1). Moreover, when $f$ is $(\gamma, \beta)$-weakly monotone submodular, Algorithm 2 also can achieve $\big(\frac{\gamma^2(1-e^{-(\beta(1-\gamma)+\gamma^2)})}{\beta(1-\gamma)+\gamma^2}\big)\text{OPT} - \epsilon$ after $\mathcal{O}(1/\epsilon)$ iterations. Note that, during the process of $\mathcal{O}(1/\epsilon)$ iterations, Algorithm 2 only requires evaluating the set function $\mathcal{O}(1/\epsilon^2)$ times.

**Remark 8.** From Line 13, we know that each $S(t)$ has at most $r$ non-zero entries, where $r = \sum_{k=1}^{K} B_k$. Hence, in Line 10, the computation of $\boldsymbol{\xi}_t$ only utilizes $\mathcal{O}(nr)$ entries in the Hessian estimation $\widehat{\nabla}_t^2$. This implies that it is sufficient to estimate up to $\mathcal{O}(nr)$ second-order partial derivatives for each point $\mathbf{x}_l(t)$ at Line 8.

**Remark 9.** Note that, in Line 16, when rounding $\mathbf{x}(T + 1)$ based on the definition of ME, there is a risk that two distinct random selections may yield the same element. To avoid this issue for monotone set functions, we present an effective rounding-without-replacement method in Appendix D.2.

Compared to the local-search methods (Thiery & Ward, 2022; Lu et al., 2022), our Algorithm 1 not only eliminates the need for prior knowledge of parameters $\alpha$, $\gamma$, and $\beta$, but can achieve the same worst-case approximation guarantees with fewer $\mathcal{O}(1/\epsilon^2)$ value queries to the set function $f$.

### 4.2. Stationary Point and Stochastic Gradient Ascent

Furthermore, an alternative solution for the subset selection problem (1) is to initially apply the off-the-shelf gradient-based methods, like gradient ascent (Nesterov, 2013; Bertsekas, 2015), to maximize our proposed ME $F$, and subsequently, to finalize our selection by rounding the resulting continuous solution. As is well known, under mild conditions, a broad range of first-order gradient algorithms, including the aforementioned gradient ascent, will converge to the stationary points of their target objectives (Ghadimi

& Lan, 2013; Lacoste-Julien, 2016; Allen-Zhu, 2018; Drori & Shamir, 2020; Lan, 2020). Regrettably, we observe that, compared to the previously proposed **Multinoulli-SCG** algorithm, the stationary points of our ME only can guarantee a *sub-optimal* approximation to the maximum value of the subset selection problem (1). Before going into the details, we firstly revisit the definition of stationary points, that is,

**Definition 2.** Given a differentiable objective function $G : \mathcal{K} \to \mathbb{R}$ and a domain $\mathcal{C} \subseteq \mathcal{K}$, a point $\mathbf{x} \in \mathcal{C}$ is called as a stationary point for the function $G$ over the domain $\mathcal{C}$ if and only if $\max_{\mathbf{y} \in \mathcal{C}} \langle \mathbf{y} - \mathbf{x}, \nabla G(\mathbf{x}) \rangle \leq 0$.

Next, we will detail the specific performance of the stationary points of the Multinoulli Extension relative to the maximum value of problem (1). Specifically, we have that

**Theorem 5** (Proof provided in Appendix E.1). *If the set function $f : 2^{\mathcal{V}} \to \mathbb{R}_+$ is monotone and $\alpha$-weakly DR-submodular, then for any stationary point $(\mathbf{p}_1, \ldots, \mathbf{p}_K)$ of its Multinoulli Extension $F$ over the domain $\prod_{k=1}^{K} \Delta_{n_k}$, the following inequality holds:*

$$F(\mathbf{p}_1, \ldots, \mathbf{p}_K) \geq \Big(\frac{\alpha^2}{1 + \alpha^2}\Big)f(S^*),$$

*where $S^*$ is the optimal solution of problem (1). Similarly, when the set function $f : 2^{\mathcal{V}} \to \mathbb{R}_+$ is monotone and $(\gamma, \beta)$-weakly submodular, for any stationary point $(\mathbf{p}_1, \ldots, \mathbf{p}_K)$ over the domain $\prod_{k=1}^{K} \Delta_{n_k}$, we also can show that*

$$F(\mathbf{p}_1, \ldots, \mathbf{p}_K) \geq \Big(\frac{\gamma^2}{\beta + \beta(1-\gamma) + \gamma^2}\Big)f(S^*).$$

**Remark 10.** Theorem 5 suggests that when the set function $f$ is monotone $\alpha$-weakly DR-submodular or $(\gamma, \beta)$-weakly submodular, the direct application of gradient-based methods that aim for stationary points, such as gradient ascent, into the relaxed problem (4) only can ensure a $(\frac{\alpha^2}{1+\alpha^2})$-approximation or $(\frac{\gamma^2}{\beta+\beta(1-\gamma)+\gamma^2})$-approximation to the maximum value of the subset selection problem (1).

Despite these limited theoretical approximation guarantees regarding the stationary points of our proposed ME, we have found that gradient-based methods targeting stationary points, such as the stochastic gradient ascent, can achieve remarkable empirical performance across numerous real-world subset selection problems (See Section 5). Due to space limitations, an in-depth analysis of the stochastic gradient ascent method is presented in Appendix E.2.

## 5. Numerical Experiments

In this section, we empirically compare the performance of our proposed **Multinoulli-SCG** and **Multinoulli-SGA** against the standard greedy method (Khashayar & Manuel, 2019) and the residual random greedy method (Chen et al.,

*Table 3.* Results on video summarization. Note that 'obj' denotes the utility function value, where a higher value is preferable, and 'queries' represents the magnitude of the total number of function evaluations, that is , the $\log_{10}$ of the total number of value queries to the set objective function, with a smaller value being more favorable. 'Distorted-LS-G' is the abbreviation for the distorted local-search method with $O(1/\epsilon)$-round guesses, namely, Algorithm B.1 in (Thiery & Ward, 2022). Both **V1** and **V2** are sourced from websites, which are related with Cooking and Animation, respectively. **V3–V7** are derived from the VSUMM dataset and encompass topics such as Soccer, Live news, Broadcast, Concert as well as TV Show. The time following the name of each video indicates the video duration. In each column of 'obj', ▇ indicates ranking the 1st and ▇ stands for the 2nd.

| Video \ Method | **V1**(3min30s) obj | queries | **V2**(7min45s) obj | queries | **V3**(2min37s) obj | queries | **V4**(2min17s) obj | queries | **V5**(2min58s) obj | queries | **V6**(4min42s) obj | queries | **V7**(2min52s) obj | queries |
|---|---|---|---|---|---|---|---|---|---|---|---|---|---|---|
| Standard Greedy | 46.48 | 3.85 | 79.25 | 3.95 | 47.64 | 3.61 | 19.55 | 3.48 | 52.39 | 3.73 | 96.33 | 4.11 | 44.85 | 3.73 |
| Residual-Greedy | 48.45 | 3.85 | 75.58 | 3.95 | 48.65 | 3.61 | 18.84 | 3.48 | 52.02 | 3.73 | 92.37 | 4.11 | 44.94 | 3.73 |
| Distorted-LS-G | 50.08 | 10.40 | 79.25 | 10.50 | 51.71 | 10.17 | 19.56 | 10.02 | 54.36 | 10.26 | 98.73 | 10.70 | 46.79 | 10.27 |
| **Multinoulli-SGA** | 50.17 | 6.53 | 79.57 | 6.58 | 51.77 | 6.41 | 19.59 | 6.35 | 54.44 | 6.46 | 98.81 | 6.66 | 47.06 | 6.46 |
| **Multinoulli-SCG** | 50.17 | 8.59 | 79.50 | 8.68 | 51.77 | 8.34 | 19.59 | 8.21 | 54.44 | 8.46 | 98.80 | 8.84 | 47.20 | 8.46 |

2018a) as well as the distorted local search (Thiery & Ward, 2022) across three distinct applications: video summarization, bayesian A-optimal design and maximum coverage. Note that **Multinoulli-SGA** represents the stochastic gradient ascent applied to our proposed ME, as detailed in Appendix E.2. Due to the space limits, here we only show the results on video summarization. More discussions about the experiment setup and the results on both maximum coverage and bayesian optimal design are presented in Appendix B.

## 5.1. Video Summarization

The objective of video summarization is aimed at picking a few representative frames from a given video such that these frames can capture as much content as possible. To achieve this, a common strategy is to formulate the frame selection problem as the maximization of a Determinantal Point Process(DPP) objective function (Gong et al., 2014; Mirzasoleiman et al., 2018; Chen et al., 2018a). DPP has recently emerged as a powerful tool that favors subsets of a ground set of items with higher diversity (Kulesza et al., 2012). More specifically, for an $n$-frame video, we represent each frame by a $p$-dimensional vector. Then, we compute the Gramian matrix $X$ of the $n$ resulting vectors by setting each $X_{ij}$ as the Gaussian kernel between the $i$-th and $j$-th vectors. With this matrix $X$, the DPP objective function can be defined as $f(S) = \det(I + X_S)$ where $S \subseteq [n]$, $X_S$ is the principal submatrix of $X$ indexed by $S$ and $I$ is a $|S|$-dimensional identity matrix. Note that Bian et al. (2017a) has proven that this set function $f$ is monotone and weakly submodular from below. Moreover, Nguyen & Thai (2022) also verified the weak DR-submodularity of $f$.

For our experiments, we use five videos from the VSUMM dataset (De Avila et al., 2011) and two videos about 'Animation' and 'Cooking' from websites like YouTube. Additionally, we utilize the method described in (Gong et al., 2014) to prune each video, namely, for long videos($\geq$ 5min), we uniformly sampled one frame per second, and for short videos, we sampled one frame every half second. Subse-

quently, we choose to create a summary of each video by extracting one representative frame from every 25 frames, that is, we consider the following partition constraint:

$$\left| S \cap [25(i-1)+1, 25i] \right| \leq 1 \quad \forall\, 1 \leq i \leq \lceil n/25 \rceil.$$

Table 3 illustrates the performance of our proposed **Multinoulli-SCG** and **Multinoulli-SGA** algorithms against three benchmark methods, namely, 'Standard Greedy', 'Residual-Greedy' and 'Distorted-LS-G'. It is quite easy to observe that our **Multinoulli-SCG** and **Multinoulli-SGA** algorithms produce summaries with higher diversity than the other three baselines. Specifically, **Multinoulli-SCG** achieves Top-1 performance on 5 out of the 7 videos, while **Multinoulli-SGA** attains Top-1 performance on 6 out of the 7 videos. Furthermore, the number of function evaluations required by our **Multinoulli-SCG** and **Multinoulli-SGA** is 2 and 4 orders of magnitude lower than that of the state-of-the-art 'Distorted-LS-G', respectively. This result aligns well with our previous theoretical findings in Section 4.

## 6. Conclusion

This paper introduces a novel continuous-relaxation framework named Multinoulli Extension(ME) for the subset selection problem under a partition constraint. In contrast to the well-known multi-linear extension for submodular subset selection, a notable advantage of our ME is that it can provide a lossless round scheme for any set objective function. Subsequently, base on ME, we develop an efficient algorithm titled **Multinoulli-SCG**, which can achieve a value of $(1-e^{-\alpha})\text{OPT} - \epsilon$ or $\left(\frac{\gamma^2(1-e^{-(\beta(1-\gamma)+\gamma^2)})}{\beta(1-\gamma)+\gamma^2}\right)\text{OPT} - \epsilon$ with only $O(1/\epsilon^2)$ function evaluations for monotone $\alpha$-weakly DR-submodular or $(\gamma, \beta)$-weakly submodular objective functions. This result significantly improves the previous $\tilde{\mathcal{O}}(1/\epsilon^6)$ and $\tilde{\mathcal{O}}(1/\epsilon^3)$ number of function evaluations associated with the distorted local-search methods. Finally, extensive empirical evaluations have been conducted to validate the effectiveness of our proposed algorithms.

## Impact Statement

This paper presents work whose goal is to advance the field of Machine Learning. There are many potential societal consequences of our work, none which we feel must be specifically highlighted here.

## Acknowledgment

This project is supported by the National Research Foundation, Singapore, under its NRF Professorship Award No. NRF-P2024-001.

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

# A. Additional Related Work

## A.1. Submodular Maximization

Submodularity (Fujishige, 2005; Bach et al., 2013) is a fundamental concept, which widely exists in various disciplines, including combinatorial optimization, economics, and machine learning. To be more precise, a set function $f : 2^{\mathcal{V}} \to \mathbb{R}_+$ is called submodular if and only if it satisfies the diminishing-return property, namely, for any two subsets $A \subseteq B \subseteq \mathcal{V}$ and $v \in \mathcal{V} \setminus B$, $f(A \cup \{v\}) - f(A) \geq f(B \cup \{v\}) - f(B)$. In other words, the marginal benefit of adding an element to a set diminishes as the set becomes larger.

Generally speaking, the maximization of submodular functions is **NP**-hard, implying that no polynomial-time algorithms can solve it optimally. To overcome this challenge, Nemhauser et al. (1978) proposed a greedy algorithm for solving the monotone submodular maximization problem under a cardinality constraint and demonstrated that this greedy algorithm can achieve an approximation ratio of $(1 - e^{-1})$. Furthermore, Feige (1998) showed that this $(1 - e^{-1})$-approximation guarantee is tight for monotone submodular maximization under reasonable complexity-theoretic assumptions. After that, Fisher et al. (1978) extended the greedy algorithm to the general matroid constraint and proved that the greedy algorithm only can guarantee a sub-optimal approximation ratio of $1/2$ under matroid constraint. To achieve the tight $(1 - e^{-1})$-approximation under matroid constraint, Calinescu et al. (2011) introduced a continuous greedy algorithm for submodular functions. The core of this continuous greedy algorithm is a novel continuous-relaxation technique known as the multi-linear extension. The problem of maximizing submodular functions also has been studied for the non-monotone case (Feldman et al., 2011; Chekuri et al., 2014; Buchbinder & Feldman, 2019; 2024).

Before delving into the details of other related studies, we introduce the multi-linear extension of (Calinescu et al., 2011) for submodular maximization and simultaneously compare it with our proposed Multinoulli Extension(ME) in Section 3. In order to better illustrate the multi-linear extension, this subsection supposes $|\mathcal{V}| = n$ and set $\mathcal{V} := [n] = \{1, \ldots, n\}$.

**Definition 3.** For a set function $f : 2^{\mathcal{V}} \to \mathbb{R}_+$, we define its multi-linear extension as

$$G(\mathbf{x}) = \sum_{\mathcal{A} \subseteq \mathcal{V}} \left( f(\mathcal{A}) \prod_{a \in \mathcal{A}} x_a \prod_{a \notin \mathcal{A}} (1 - x_a) \right) = \mathbb{E}_{\mathcal{R} \sim \mathbf{x}} \left( f(\mathcal{R}) \right), \tag{5}$$

where $\mathbf{x} = (x_1, \ldots, x_n) \in [0, 1]^n$ and $\mathcal{R} \subseteq \mathcal{V}$ is a random set that contains each element $a \in \mathcal{V}$ independently with probability $x_a$ and excludes it with probability $1 - x_a$. We write $\mathcal{R} \sim \mathbf{x}$ to denote that $\mathcal{R} \subseteq \mathcal{V}$ is a random set sampled according to $\mathbf{x}$.

From the Eq.(5), we can view multi-linear extension $G$ at any point $\mathbf{x} \in [0, 1]^n$ as the expected utility of independently selecting each action $a \in \mathcal{V}$ with probability $x_a$. With this tool, we can cast the previous discrete subset selection problem (1) into a continuous maximization which learns the independent probability for each element $a \in \mathcal{V}$, that is, we consider the following continuous optimization:

$$\max_{\mathbf{x} \in [0,1]^n} G(\mathbf{x}), \quad \text{s.t.} \quad \sum_{a \in \mathcal{V}_k} x_a \leq B_k, \forall k \in [K] \tag{6}$$

where $G(\mathbf{x})$ is the multi-linear extension of $f$.

It is important to note that, if we round any point $\mathbf{x}$ included into the constraint of problem (6) by the definition of multi-linear extension, i.e., Eq.(5), there is a certain probability that the resulting subset will violate the partition constraint of the subset selection problem (1). Therefore, for multi-linear extension, we need to specifically design the rounding methods based on the properties of the set objective functions. However, current known lossless rounding schemes for multi-linear extension, such as pipage rounding (Ageev & Sviridenko, 2004), swap rounding (Chekuri et al., 2010) and contention resolution (Chekuri et al., 2014), are heavily dependent on the *submodular* assumption. Moreover, how to losslessly round the multi-linear extension of *non-submodular* set functions, e.g. $(\gamma, \beta)$-weakly submodular and $\alpha$-weakly DR-submodular functions, still remains an open question (Thiery & Ward, 2022).

In contrast, our ME does not assign probabilities to any subsets that are out of the partition constraint of problem (1), which means that, for any set function $f : 2^{\mathcal{V}} \to \mathbb{R}_+$ and any given $(\mathbf{p}_1, \ldots, \mathbf{p}_K) \in \prod_{k=1}^{K} \Delta_{n_k}$, we can, through the definition of ME, easily produce a subset that conforms to the partition constraint of problem (1) without any loss in terms of the expected function value $F(\mathbf{p}_1, \ldots, \mathbf{p}_K)$.

### A.2. Close-to-Submodular Function Maximization

**Weakly Submodular Maximization:** A particularly important class of "*close-to-submodular*" functions is known as $\gamma$-weakly submodular functions. Specifically, for a set function $f : 2^{\mathcal{V}} \to \mathbb{R}_+$, it is called $\gamma$-weakly submodular if and only if, for any two subsets $A \subseteq B \subseteq \mathcal{V}$, the following inequality holds: $\sum_{v \in B \setminus A} f(v|A) \geq \gamma\big(f(B) - f(A)\big)$. The $\gamma$-weakly submodular functions were originally introduced by a work of (Das & Kempe, 2011), which also demonstrated that the standard greedy algorithm achieves a good approximation ratio of $(1 - e^{-\gamma})$ for the problem of maximizing such functions subject to a cardinality constraint. On the inapproximability side, Harshaw et al. (2019) proved that no polynomial-time algorithm achieves $(1 - e^{-\gamma} + \epsilon)$-approximation for the problem of maximizing a $\gamma$-weakly submodular function subject to a cardinality constraint, for any $\epsilon > 0$. Subsequently, Chen et al. (2018a) investigate weakly submodular maximization beyond simple cardinality constraints and pointed out that the Residual Random Greedy method of (Buchbinder et al., 2014) can achieve an approximation ratio of $\frac{\gamma^2}{(1+\gamma)^2}$ for the problem of maximizing a monotone $\gamma$-weakly submodular functions subject to a matroid constraint. After that, Khashayar & Manuel (2019) examined the approximation performance of standard greedy algorithm on $\gamma$-weakly submodular maximization over matroid constraints, which showed that the standard greedy algorithm can offer an approximation factor of $\frac{0.4\gamma^2}{\sqrt{r}\gamma + 1}$ where $r$ is the rank of the matroid. It is important to note that this approximation ratio $\frac{0.4\gamma^2}{\sqrt{r}\gamma + 1}$ is not a constant guarantee and highly depends on the matroid rank $r$. In order to improve the approximation performance of these greedy-based algorithms, Thiery & Ward (2022) introduced the notion of upper submodularity ratio $\beta$, defined as Eq. (3). Moreover, Thiery & Ward (2022) also developed a more powerful distorted local-search algorithm for $(\gamma, \beta)$-weakly submodular maximization, which is inspired by the non-oblivious search (Filmus & Ward, 2012b; 2014) for submodular maximization and can guarantee a $\frac{\gamma^2(1 - e^{-(\beta(1-\gamma)+\gamma^2)})}{\beta(1-\gamma)+\gamma^2}$-approximation for the problem of maximizing a monotone $(\gamma, \beta)$-weakly submodular functions subject to a matroid constraint. Note that, when the $(\gamma, \beta)$-weakly submodular function is closer to being submodular, namely, $\gamma, \beta \to 1$, the approximation ratio $\frac{\gamma^2(1 - e^{-(\beta(1-\gamma)+\gamma^2)})}{\beta(1-\gamma)+\gamma^2}$ will approach the tight $(1 - 1/e)$. Conversely, when $\gamma, \beta \to 1$, the approximation guarantee $\frac{\gamma^2}{(1+\gamma)^2}$ of the Residual Random Greedy method (Chen et al., 2018a) will trends toward the sub-optimal $1/4$.

**Weakly DR-Submodular Maximization:** Another significant class of "*close-to-submodular*" functions is known as $\alpha$-weakly DR-submodular functions, where $\alpha$ is variously referred to as the diminishing-return(DR) ratio (Kuhnle et al., 2018), the generalized curvature (Bogunovic et al., 2018) or the generic submodularity ratio (Gong et al., 2021). The work of Khashayar & Manuel (2019) is the first to give a study of maximizing a $\alpha$-weakly DR-submodular maximization subject to general matroid constraints. Specifically, Khashayar & Manuel (2019) proved that the greedy algorithm achieves approximation ratios of $\frac{\alpha}{1+\alpha}$ for the matroid-constrained $\alpha$-weakly DR-submodular maximization. After that, Gong et al. (2021) extended this result to $p$-matroid constraints. Moreover, Nguyen & Thai (2022) also considered the impact of curvature (Vondrák, 2010; Sviridenko et al., 2017) on the standard greedy algorithm for $\alpha$-weakly DR-submodular maximization under partition matroid. Recently, Gong et al. (2019) showed that the continuous greedy combined with the contention resolution scheme (Chekuri et al., 2014) can obtain a sub-optimal approaximation ratio of $\alpha(1 - 1/e)(1 - e^{-\alpha})$ for the problem of maximizing a monotone $\alpha$-weakly DR-submodular functions subject to a matroid constraint. To achieve the tight $(1 - e^{-\alpha})$-approximation guarantee, Lu et al. (2022) recently have proposed a novel distorted local-search method, which is also motivated via the non-oblivious search (Filmus & Ward, 2014).

### A.3. Monotone Continuous DR-submodular Maximization

Submodularity can be naturally extended to continuous domains. Generally speaking, a differentiable function $F : [0, 1]^n \to \mathbb{R}_+$ is *DR-submodular* if $\nabla F(\mathbf{x}) \leq \nabla F(\mathbf{y})$ for any $\mathbf{x} \geq \mathbf{y}$. In deterministic setting, Bian et al. (2017b) first proposed a variant of Frank-Wolfe for continuous DR-submodular maximization problem with $(1 - 1/e - \epsilon)$-approximation guarantee after $O(1/\epsilon)$ iterations. When considering the stochastic gradient oracle, Hassani et al. (2017) proved that the stochastic gradient ascent can guarantee a $(1/2)$-approximation after $O(1/\epsilon^2)$ iterations. Then, Mokhtari et al. (2018a) proposed the stochastic continuous greedy algorithm, which achieves a $(1 - 1/e - \epsilon)$-approximation after $O(1/\epsilon^3)$ iterations. Moreover, by assuming the Hessian of objective is Lipschitz continuous and considering non-oblivious stochastic noise, Hassani et al. (2020) proposed the stochastic continuous greedy++ algorithm, which can guarantee a $(1 - 1/e - \epsilon)$-approximation after $O(1/\epsilon^2)$ iterations. After that, Zhang et al. (2022; 2024) proposed a non-oblivious auxiliary function for continuous DR-submodular functions, which can efficiently improve the approximation ratio of stochastic gradient ascent (Hassani et al., 2017) from $1/2$ to $(1 - 1/e)$. In addition, numerous variants of continuous DR-submodular maximization have been extensively studied, for instance, online scenarios (Chen et al., 2018b; Zhang et al., 2019; Pedramfar et al., 2023; 2024;

Zhang et al., 2023a), decentralized environments (Mokhtari et al., 2018b; Xie et al., 2019; Zhu et al., 2021; Gao et al., 2021; Zhang et al., 2023b; 2025), and bandit scenarios (Chen et al., 2020; Zhang et al., 2019; Wan et al., 2023; Pedramfar et al., 2024; Zhao et al., 2025; Zhao & Lai, 2024; Yu et al., 2024a;b; Jin et al., 2024; 2025).

We need to emphasize that, when the set function $f$ is monotone submodular, Theorem 2 implies that our proposed Multinoulli Extension is also monotone continuous DR-submodular over the domain $\prod_{k=1}^{K} \Delta_{n_k}$. However, it is worth noting that these former results about continuous DR-submodular maximization cannot be directly applied to our ME and the relaxed problem (4). This is because all of them highly rely on the following inequality: if $G$ is a monotone continuous DR-submodular function,

$$\langle \mathbf{y} - \mathbf{x}, \nabla G(\mathbf{x}) \rangle \geq \langle \mathbf{y} \vee \mathbf{x} - \mathbf{x}, \nabla G(\mathbf{x}) \rangle \geq G(\mathbf{y} \vee \mathbf{x}) - G(\mathbf{x}) \geq G(\mathbf{y}) - G(\mathbf{x}), \tag{7}$$

where $\vee$ is the coordinate-wise maximum operation, i.e., $\mathbf{x} \vee \mathbf{y} = \max(\mathbf{x}, \mathbf{y})$.

Note that the second inequality in Eq.(7) requires that the vector $\mathbf{y} \vee \mathbf{x}$ is included into the domain of objective function $G$. In other words, the domain of objective function $G$ should be closed under the coordinate-wise maximum operation $\vee$. However, the domain $\prod_{k=1}^{K} \Delta_{n_k}$ of our ME does not meet this requirement. Similarly, to the best of our knowledge, the latest research on both upper-linearizable and weakly continuous DR-submodular maximization also requires the domain of the objective function to be closed under the coordinate-wise maximum operation $\vee$. (See (Hassani et al., 2017; Zhang et al., 2022; Pedramfar & Aggarwal, 2024))

## B. Additional Experimental Results

### B.1. More Discussions on Experiment Setup

In this subsection, we highlight some additional details about the experiments. At first, we describe the parameter setups regarding our proposed 'Multinoulli-SGA', 'Multinoulli-SCG' and 'Distorted-LS-G'. More specifically, we consider the following parameter configurations:

- '**Distorted-LS-G**': Algorithm B.1 in (Thiery & Ward, 2022) where we set the number of guesses $L = 1 + \lceil \log_{(1-\epsilon)}(\frac{3}{16}) \rceil$, the total number of improvements $M = \frac{r}{\epsilon}$, the number of samples $N = \frac{r}{\epsilon^2}$ and $\epsilon = 0.01$ where $r$ is the rank of the partition constraint, namely, $r = \sum_{k=1}^{K} B_k$ in problem (1);

- '**Multinoulli-SGA**': Algorithm 3 is implemented with batch size $L = 20$, the step size $\eta = \frac{1}{\sqrt{T}}$ and the number of iterations $T = 167$. Moreover, we round each continuous solution via Algorithm 2. As for the Euclidean projection of Step 6 in Algorithm 3, we utilize the CVX optimization tool (Grant & Boyd, 2014). It is worth noting that there exists efficient algorithms for handling the projection over partitions if we view it as multiple independent singly constrained quadratic programmings problems(See (Pardalos & Kovoor, 1990) or Appendix B in (Zhou et al., 2022));

- '**Multinoulli-SCG**': Algorithm 1 is implemented with batch size $L = \lceil \frac{T}{2} \rceil$ and the number of iterations $T = 167$. Furthermore, in order to obtain a high-quality subset, we apply Algorithm 2 to round the $\mathbf{x}(T + 1)$ a total of $T^2$ times and then select the best one among these resulting subsets.

Note that, when $\epsilon = 0.01$, the number of guesses $L = 1 + \lceil \log_{(1-\epsilon)}(\frac{3}{16}) \rceil \approx 167$ in 'Distorted-LS-G' such that we set the total of iterations $T = 167$ in both '**Multinoulli-SGA**' and '**Multinoulli-SCG**'. Given the long runtimes of '**Multinoulli-SCG**' and 'Distorted-LS-G', we report their average results in Table 3 based on 5 repeated experiments. In contrast, '**Multinoulli-SGA**' and 'Residual-Greedy' algorithms are repeated 10 times. Finally, all experiments are performed in Python 3.6.5 on a MacBook Pro with Apple M1 Pro and 16GB RAM.

### B.2. Maximum Coverage

In the tasks of video summarization, it is easily observed that our **Multinoulli-SGA** algorithm exhibits exceptional empirical performance. However, according to both Theorem 5 and Theorem 9, we know that our **Multinoulli-SGA** only can ensure a sub-optimal approximation guarantee in the worst case. To verify the theoretical correctness of our proposed **Multinoulli-SGA** algorithm, we consider a special maximum coverage problem as discussed in (Filmus & Ward, 2012a).

Let the universe set $U$ consist of $n - 1$ elements $\{x_1, \ldots, x_{n-1}\}$ and $n - k$ elements $\{y_1, \ldots, y_{n-k}\}$, all of weight 1, and $n - 1$ elements $\{\epsilon_1, \ldots, \epsilon_{n-1}\}$ of arbitrarily small weight $\epsilon > 0$. Then, we define two different set $A_i$ and $B_i$ for any $i \in [n]$,

*Table 4.* Results on coverage maximization(2nd-6th columns) and bayesian A-optimal design(the final column). Note that 'obj' denotes the utility function value, where a higher value is preferable, and 'queries' represents the magnitude of the total number of function evaluations, that is , the $\log_{10}$ of the total number of value queries to the set objective function, with a smaller value being more favorable. 'Distorted-LS-G' is the abbreviation for the distorted local-search method with $O(1/\epsilon)$-round guesses, namely, Algorithm B.1 in (Thiery & Ward, 2022). In each column of 'obj', ▇ indicates ranking the 1st and ▇ stands for the 2nd.

| Setting / Method | n=15, k=5 | | n=20, k=5 | | n=30, k=6 | | n=40, k=8 | | n=50, k=10 | | Housing | |
|---|---|---|---|---|---|---|---|---|---|---|---|---|
| | obj | queries | obj | queries | obj | queries | obj | queries | obj | queries | obj | queries |
| Standard Greedy | 14.14 | 2.65 | 19.19 | 2.90 | 29.29 | 3.26 | 39.39 | 3.51 | 49.49 | 3.70 | 61.295 | 3.70 |
| Residual-Greedy | 21.04 | 2.65 | 32.52 | 2.90 | 48.25 | 3.26 | 64.67 | 3.51 | 77.13 | 3.70 | 61.104 | 3.70 |
| Distorted-LS-G | 24.00 | 9.06 | 34.00 | 9.44 | 53.00 | 9.98 | 71.00 | 10.36 | 89.00 | 10.65 | 61.285 | 10.26 |
| **Multinoulli-SGA** | 14.14 | 5.45 | 19.18 | 5.56 | 29.26 | 5.72 | 39.34 | 5.83 | 49.43 | 5.92 | 61.745 | 6.61 |
| **Multinoulli-SCG** | 24.00 | 7.39 | 34.00 | 7.64 | 53.00 | 7.99 | 71.00 | 8.24 | 89.00 | 8.43 | 61.456 | 8.44 |

that is to say ,

$$A_i = \{\epsilon_i\} \text{ for } 1 \le i \le n-1, \qquad A_n = \{x_1, \ldots, x_{n-1}\},$$
$$B_i = \{x_i\} \text{ for } 1 \le i \le n-1, \qquad B_n = \{y_1, \ldots, y_{n-k}\}.$$

After that, we define a coverage set function $f : 2^{\mathcal{V}} \to \mathbb{R}_+$ over these $2n$ distinct set $\{A_1, \ldots, A_n, B_1, \ldots, B_n\}$ where $\mathcal{V} = [2n]$. More specifically, we have that, for any subset $\mathcal{F} \subseteq \mathcal{V}$,

$$f(\mathcal{F}) = \sum_{v \in \bigcup_{j \in \mathcal{F}} S_j} w(v), \tag{8}$$

where $w(v)$ is the weight of element $v$, $S_j = A_j$ when $1 \le j \le n$ and $S_j = B_{j-n}$ as for $n+1 \le j \le 2n$.

Moreover, we consider a partition constraint that contains at most one of $\{A_i, B_i\}$ for any $i \in [n]$. If we set $\mathcal{V}_i = \{i, i+n\}$($\mathcal{V} = \bigcup_{i \in [n]} \mathcal{V}_i = [2n]$), we naturally obtain the following coverage maximization problem:

$$\max_{\mathcal{F} \subseteq \mathcal{V}} f(\mathcal{F}) \text{ s.t. } |\mathcal{F} \cap \mathcal{V}_i| \le 1 \ \forall i \in [n]. \tag{9}$$

From the result of (Filmus & Ward, 2012a), we know that the problem (9) is a submodular maximization problem subject to a partition matroid constraint, namely, $\alpha = \beta = \gamma = 1$. A key feature of this coverage maximization problem is that Filmus & Ward (2012a) found that the standard greedy (Nemhauser et al., 1978) will be stuck at a local maximum subset $\{A_1, \ldots, A_n\}$ where $\mathcal{F} = [n]$ and $f(\mathcal{F}) = (1+\epsilon)n$. In contrast, when $\epsilon$ is very small, the optimal subset for problem (9) is $\{B_1, \ldots, B_n\}$ where $\mathcal{F} = \{n+1, \ldots, 2n\}$ and $f(\mathcal{F}) = 2n - k - 1$. Note that $\lim_{n \to \infty} \lim_{\epsilon \to 0 \text{ and } k \to 0} \frac{(1+\epsilon)n}{2n-k-1} = \frac{1}{2}$.

Motivated by this finding of (Filmus & Ward, 2012a) and the correspondence between the set $[n]$ and $\{A_1, \ldots, A_n\}$ in problem (9), we conjecture that the point $\mathbf{1}_{[n]}$ may be a local stationary point of the Multinoulli Extension of the set function $f$ in Eq.(8). Before showing the rigorous proof of the previous conjecture, we firstly compare the empirical performance of our proposed **Multinoulli-SCG** and **Multinoulli-SGA** against the standard greedy method and the residual random greedy method as well as the distorted local search across distinct coverage maximization (9) with different $n$ and $k$, where we uniformly set the weight $\epsilon = 0.01$.

From the results in Table 4, we found that both our proposed '**Multinoulli-SCG**' and 'Distorted-LS-G' algorithm eventually select the optimal subset, i.e., $\{B_1, \ldots, B_n\}$. In contrast, in all settings of maximum coverage problems, '**Multinoulli-SGA**' and 'Standard-Greedy' algorithm are trapped around the local-optimal set $\{A_1, \ldots, A_n\}$. Moreover, the 'Residual-Greedy' method oscillates between the optimal subset $\{B_1, \ldots, B_n\}$ and the local maximum set $\{A_1, \ldots, A_n\}$. Similar to the video summarization in Section 5.1, Table 4 also demonstrated that the number of function evaluations required by our **Multinoulli-SCG** is 2 orders of magnitude lower than that of the 'Distorted-LS-G' algorithm,which highlights the efficiency and effectiveness of **Multinoulli-SCG** algorithm.

Now, we show that the point $\mathbf{1}_{[n]}$ is a stationary point for the ME of the set function $f$ in Eq.(8). Before that, we detail the ME $F$ of the set function $f$ of Eq.(8), that is,

$$F(\mathbf{p}_1, \ldots, \mathbf{p}_n) = \sum_{i=1}^{n} \sum_{e_i^1 \in \{i, i+n, \emptyset\}} \left( f\left( \cup_{i=1}^{n} \{e_i^1\} \right) \prod_{i=1}^{n} \Pr(e_i^1 | \mathbf{p}_i) \right), \tag{10}$$

where we set $\mathbf{p}_i = (p_i^1, p_i^2)$, $\Pr(i|\mathbf{p}_i) = p_i^1$, $\Pr(n+i|\mathbf{p}_i) = p_i^2$ and $\Pr(\emptyset|\mathbf{p}_i) = 1 - p_i^1 - p_i^2$.

From Theorem 1, we know that,

$$\frac{\partial F}{\partial(p_i^1)}(\mathbf{1}_{[n]}) = \epsilon \text{ for } 1 \leq i \leq n-1, \qquad \frac{\partial F}{\partial(p_n^1)}(\mathbf{1}_{[n]}) = n-1,$$

$$\frac{\partial F}{\partial(p_i^2)}(\mathbf{1}_{[n]}) = 0 \text{ for } 1 \leq i \leq n-1, \qquad \frac{\partial F}{\partial(p_n^2)}(\mathbf{1}_{[n]}) = n-k.$$

As a result, for any $(\mathbf{p}_1, \ldots, \mathbf{p}_n) \in \prod_{i=1}^n \triangle_2$, we have

$$\langle(\mathbf{p}_1, \ldots, \mathbf{p}_n) - \mathbf{1}_{[n]}, \nabla F(\mathbf{1}_{[n]})\rangle$$

$$= \sum_{i=1}^n \left(p_i^1 \frac{\partial F}{\partial(p_i^1)}(\mathbf{1}_{[n]}) + p_i^2 \frac{\partial F}{\partial(p_i^2)}(\mathbf{1}_{[n]})\right) - (1+\epsilon)(n-1)$$

$$= \epsilon \sum_{i=1}^{n-1} p_i^1 + (n-1)p_n^1 + (n-k)p_n^2 - (1+\epsilon)(n-1)$$

$$= \epsilon \sum_{i=1}^{n-1} (p_i^1 - 1) + (n-1)p_n^1 + (n-k)p_n^2 - (n-1)$$

$$= \epsilon \sum_{i=1}^{n-1} (p_i^1 - 1) + \big((n-k) - (n-1)\big)p_n^2 - (n-1)(1 - p_n^1 - p_n^2) \leq 0.$$

As a result, $\mathbf{1}_{[n]}$ is a stationary point of the ME $F$. Note that $\frac{F(\mathbf{1}_{[n]})}{F(\mathbf{1}_{\{n+1,\ldots,2n\}})} = \frac{(1+\epsilon)(n-1)}{2n-1-k} \to \frac{1}{2}$ when $k \to 0$, $\epsilon \to 0$ and $n \to \infty$, which implies that the approximation guarantee of Theorem 5 is tight when $\alpha = \beta = \gamma = 1$.

### B.3. Bayesian A-Optimal Design

In this subsection, we consider a classical subset selection problem in experimental design (Hashemi et al., 2019; Borsos et al., 2020; 2024), namely, bayesian A-optimal design.

We first describe the details of the bayesian A-optimal design. Suppose that $\theta \in \mathbb{R}^d$ is an unknown parameter vector that we wish to estimate from noisy linear measurements using least squares regression. Our goal is to choose a set $S$ of linear measurements (the so-called experiments) which have low cost and also maximally reduce the variance of our OLS estimate. More precisely, let $X = [x_1, \ldots, x_n] \in \mathbb{R}^{d \times n}$ be a matrix including $n$ different measurements. Given a set of measurement vectors $S \subseteq [n]$, we may run the experiments and obtain the noisy linear observations $y_S = X_S^T \theta + \epsilon_S$, where $\epsilon_S \sim N(\mathbf{0}, \sigma^2 I_{|S|})$. We estimate $\theta$ using the OLS $\hat{\theta} = (X_S X_S^T)^{-1} X_S y_S$. After assuming a normal Bayesian prior distribution on the unknown parameter $\theta \sim N(\mathbf{0}, \Sigma)$, we can compute the sum of the variance of the coefficients as $r(S) = tr(\Sigma + \frac{1}{\sigma^2} X_S X_S^T)^{-1}$ where $tr(\cdot)$ is the trace of a matrix. If we redefine the $g(S) = r(\emptyset) - r(S) = tr(\Sigma) - tr(\Sigma + \frac{1}{\sigma^2} X_S X_S^T)^{-1}$, we can reformulate the bayesian A-optimal design as a $(\gamma, \beta)$-weakly submodular maximization problem (Thiery & Ward, 2022) with $\beta = 1/\gamma$.

In our experiment, we use the Boston Housing dataset (Harrison Jr & Rubinfeld, 1978), a standard benchmark dataset containing $d = 14$ features of $n = 506$ Boston homes, including average number of rooms per dwelling, proximity to the Charles River, and crime rate per capita. Like Harshaw et al. (2019), we preprocessed the data by normalizing the features to have a zero mean and a standard deviation of 1. After that, we set $\sigma = 1/\sqrt{d}$ and randomly generated a normal prior with covariance $\Sigma = ADA^T$ where $A \sim N(\mathbf{0}, I)$ and $D$ is a diagonal matrix with $D_{i,i} = (i/d)^2$. As for the partition constraint, we randomly cut the all 506 data points into 10 different groups and then select at most one element from each group. Then, we present the results of bayesian A-optimal design in the final column in Table 4.

From the final column in Table 4, the performance of our proposed **Multinoulli-SCG** and **Multinoulli-SGA** algorithms surpasses that of three benchmark methods, namely, 'Standard Greedy', 'Residual-Greedy' and 'Distorted-LS-G'. Furthermore, like the previous experiments about video summarization and maximum coverage, the number of value queries to the set function required by our **Multinoulli-SCG** and **Multinoulli-SGA** is 2 and 4 orders of magnitude lower than that of the 'Distorted-LS-G', respectively.

## C. The Properties of Multinoulli Extension

### C.1. Proof of Theorem 1

In this section, we prove the Theorem 1.

*Proof.* **1):** At first, we review the definition of Multinoulli Extension. For any given set function $f : 2^{\mathcal{V}} \to \mathbb{R}_+$ and any point $(\mathbf{p}_1, \ldots, \mathbf{p}_K) \in \prod_{k=1}^{K} \Delta_{n_k}$, we define its Multinoulli Extension for the subset selection problem (1) as:

$$F(\mathbf{p}_1, \ldots, \mathbf{p}_K) := \sum_{e_{\hat{k}}^{\hat{b}} \in \mathcal{V}_{\hat{k}} \cup \{\emptyset\}, \forall \hat{b} \in [B_{\hat{k}}], \forall \hat{k} \in [K]} \left( f\big( \cup_{\hat{k}=1}^{K} \cup_{\hat{b}=1}^{B_{\hat{k}}} \{e_{\hat{k}}^{\hat{b}}\}\big) \prod_{\hat{k}=1}^{K} \prod_{\hat{b}=1}^{B_{\hat{k}}} \Pr(e_{\hat{k}}^{\hat{b}} | \mathbf{p}_{\hat{k}}) \right), \tag{11}$$

where $\Pr(v_{\hat{k}}^m | \mathbf{p}_{\hat{k}}) = p_{\hat{k}}^m$ and $\Pr(\emptyset | \mathbf{p}_{\hat{k}}) = 1 - \sum_{m=1}^{n_{\hat{k}}} p_{\hat{k}}^m$ for any $m \in [n_{\hat{k}}]$ and $\hat{k} \in [K]$.

From Eq.(11), for any parameter $p_k^m$ where $m \in [n_k]$ and $k \in [K]$, we have the following equality:

$$\frac{\partial F}{\partial p_k^m}(\mathbf{p}_1, \ldots, \mathbf{p}_K) = \sum_{e_{\hat{k}}^{\hat{b}} \in \mathcal{V}_{\hat{k}} \cup \{\emptyset\}, \forall \hat{b} \in [B_{\hat{k}}], \forall \hat{k} \in [K]} \left( f\big( \cup_{\hat{k}=1}^{K} \cup_{\hat{b}=1}^{B_{\hat{k}}} \{e_{\hat{k}}^{\hat{b}}\}\big) \frac{\partial \big( \prod_{\hat{k}=1}^{K} \prod_{\hat{b}=1}^{B_{\hat{k}}} \Pr(e_{\hat{k}}^{\hat{b}} | \mathbf{p}_{\hat{k}})\big)}{\partial p_k^m} \right). \tag{12}$$

Then, according to the definition of $\Pr(e_{\hat{k}}^{\hat{b}} | \mathbf{p}_{\hat{k}})$, we can show that, if $k \neq \hat{k}$, $\frac{\partial \Pr(e_{\hat{k}}^{\hat{b}} | \mathbf{p}_{\hat{k}})}{\partial p_k^m} = 0$ for any $e_{\hat{k}}^{\hat{b}} \in \mathcal{V}_{\hat{k}} \cup \{\emptyset\}$. When $k = \hat{k}$, we also can show that, if $e_{\hat{k}}^{\hat{b}} = v_k^m$, $\frac{\partial \Pr(e_{\hat{k}}^{\hat{b}} | \mathbf{p}_{\hat{k}})}{\partial p_k^m} = 1$ and if $e_{\hat{k}}^{\hat{b}} = \emptyset$, $\frac{\partial \Pr(e_{\hat{k}}^{\hat{b}} | \mathbf{p}_{\hat{k}})}{\partial p_k^m} = -1$. As for $e_{\hat{k}}^{\hat{b}} \notin \{v_k^m, \emptyset\}$, when $k = \hat{k}$, $\frac{\partial \Pr(e_{\hat{k}}^{\hat{b}} | \mathbf{p}_{\hat{k}})}{\partial p_k^m} = 0$. As a result, we can rewrite the Eq.(12) as:

$$\begin{aligned}
\frac{\partial F}{\partial p_k^m} &= \sum_{e_{\hat{k}}^{\hat{b}} \in \mathcal{V}_{\hat{k}} \cup \{\emptyset\}, \forall \hat{b} \in [B_{\hat{k}}], \forall \hat{k} \in [K]} \left( f\big( \cup_{\hat{k}=1}^{K} \cup_{\hat{b}=1}^{B_{\hat{k}}} \{e_{\hat{k}}^{\hat{b}}\}\big) \sum_{\hat{b}_1 \in [B_k]} \left( \Big( \prod_{(\hat{k}, \hat{b}) \neq (k, \hat{b}_1)} \Pr(e_{\hat{k}}^{\hat{b}} | \mathbf{p}_{\hat{k}}) \Big) \frac{\partial \Pr(e_k^{\hat{b}_1} | \mathbf{p}_k)}{\partial p_k^m} \right) \right) \\
&= \sum_{\hat{b}_1 \in [B_k]} \left( \sum_{e_{\hat{k}}^{\hat{b}} \in \mathcal{V}_{\hat{k}} \cup \{\emptyset\}, \forall \hat{b} \in [B_{\hat{k}}], \forall \hat{k} \in [K]} \left( f\big( \cup_{\hat{k}=1}^{K} \cup_{\hat{b}=1}^{B_{\hat{k}}} \{e_{\hat{k}}^{\hat{b}}\}\big) \frac{\partial \Pr(e_k^{\hat{b}_1} | \mathbf{p}_k)}{\partial p_k^m} \Big( \prod_{(\hat{k}, \hat{b}) \neq (k, \hat{b}_1)} \Pr(e_{\hat{k}}^{\hat{b}} | \mathbf{p}_{\hat{k}}) \Big) \right) \right) \\
&= \sum_{\hat{b}_1 \in [B_k]} \left( \sum_{e_{\hat{k}}^{\hat{b}} \in \mathcal{V}_{\hat{k}} \cup \{\emptyset\}, (\hat{k}, \hat{b}) \neq (k, \hat{b}_1)} \left( \sum_{e_k^{\hat{b}} \in \mathcal{V}_k \cup \{\emptyset\}} \left( f\big( \cup_{\hat{k}=1}^{K} \cup_{\hat{b}=1}^{B_{\hat{k}}} \{e_{\hat{k}}^{\hat{b}}\}\big) \frac{\partial \Pr(e_k^{\hat{b}_1} | \mathbf{p}_k)}{\partial p_k^m} \prod_{(\hat{k}, \hat{b}) \neq (k, \hat{b}_1)} \Pr(e_{\hat{k}}^{\hat{b}} | \mathbf{p}_{\hat{k}}) \right) \right) \right) \\
&= \sum_{\hat{b}_1 \in [B_k]} \left( \sum_{e_{\hat{k}}^{\hat{b}} \in \mathcal{V}_{\hat{k}} \cup \{\emptyset\}, (\hat{k}, \hat{b}) \neq (k, \hat{b}_1)} \left( f\Big( v_k^m \big| \cup_{(\hat{k}, \hat{b}) \neq (k, \hat{b}_1)} \{e_{\hat{k}}^{\hat{b}}\}\Big) \prod_{(\hat{k}, \hat{b}) \neq (k, \hat{b}_1)} \Pr(e_{\hat{k}}^{\hat{b}} | \mathbf{p}_{\hat{k}}) \right) \right) \\
&= B_k \left( \sum_{e_{\hat{k}}^{\hat{b}} \in \mathcal{V}_{\hat{k}} \cup \{\emptyset\}, (\hat{k}, \hat{b}) \neq (k, 1)} \left( f\Big( v_k^m \big| \cup_{(\hat{k}, \hat{b}) \neq (k, 1)} \{e_{\hat{k}}^{\hat{b}}\}\Big) \prod_{(\hat{k}, \hat{b}) \neq (k, 1)} \Pr(e_{\hat{k}}^{\hat{b}} | \mathbf{p}_{\hat{k}}) \right) \right) \\
&= B_k \left( \Big( \sum_{e_k^1 \in \mathcal{V}_k \cup \{\emptyset\}} \Pr(e_k^1 | \mathbf{p}_k) \Big) \sum_{e_{\hat{k}}^{\hat{b}} \in \mathcal{V}_{\hat{k}} \cup \{\emptyset\}, (\hat{k}, \hat{b}) \neq (k, 1)} \left( f\Big( v_k^m \big| \cup_{(\hat{k}, \hat{b}) \neq (k, 1)} \{e_{\hat{k}}^{\hat{b}}\}\Big) \prod_{(\hat{k}, \hat{b}) \neq (k, 1)} \Pr(e_{\hat{k}}^{\hat{b}} | \mathbf{p}_{\hat{k}}) \right) \right) \\
&= B_k \left( \sum_{e_{\hat{k}}^{\hat{b}} \in \mathcal{V}_{\hat{k}} \cup \{\emptyset\}, \forall \hat{b} \in [B_{\hat{k}}], \forall \hat{k} \in [K]} \left( f\Big( v_k^m \big| \cup_{(\hat{k}, \hat{b}) \neq (k, 1)} \{e_{\hat{k}}^{\hat{b}}\}\Big) \prod_{\hat{k} \in [K]} \prod_{\hat{b} \in [B_k]} \Pr(e_{\hat{k}}^{\hat{b}} | \mathbf{p}_{\hat{k}}) \right) \right) \\
&= B_k \left( \mathbb{E}_{e_{\hat{k}}^{\hat{b}} \sim Multi(\mathbf{p}_{\hat{k}})} \Big( f\Big( v_k^m \big| \cup_{(\hat{k}, \hat{b}) \neq (k, 1)} \{e_{\hat{k}}^{\hat{b}}\}\Big) \Big) \right),
\end{aligned} \tag{13}$$

where the fifth equality comes from that each random element $e_k^{\hat{b}_1}$ is independently drawn the same multinoulli distribution Multi$(\mathbf{p}_k)$ for any $\hat{b}_1 \in [B_k]$ and the sixth equality follows from $\sum_{e_k^1 \in \mathcal{V}_k \cup \{\emptyset\}} \Pr(e_k^1 | \mathbf{p}_k) = 1$.

**2):** Next, we prove the second point of Theorem 1. At first, the monotonicity of $f$ implies that for any two subsets $A \subseteq B \subseteq \mathcal{V}$, $f(A) \leq f(B)$ such that we know that $f\left(v_k^m \middle| \cup_{(\hat{k},\hat{b}) \neq (k,1)} \{e_{\hat{k}}^{\hat{b}}\}\right) \geq 0$ for any random elements $e_{\hat{k}}^{\hat{b}}, \forall \hat{b} \in [B_{\hat{k}}], \forall \hat{k} \in [K]$. As a result, we have If $f$ is monotone, then $\frac{\partial F}{\partial p_k^m} \geq 0, \forall k \in [K], m \in [n_k]$.

**3):** As for the third point, we firstly unify the process of generating random elements regarding two different multinoulli distributions with parameters $\mathbf{p}_k = (p_k^1, \ldots, p_k^{n_k}) \in \Delta_{n_k}$ and $\hat{\mathbf{p}}_k = (\hat{p}_k^1, \ldots, \hat{p}_k^{n_k}) \in \Delta_{n_k}$ where $\hat{p}_k^m \geq p_k^m$ for any $m \in [n_k]$ and $k \in [K]$. More specifically, we will transform the sampling process from each multinoulli distribution Multi$(\mathbf{p}_k)$ or Multi$(\hat{\mathbf{p}}_k)$ into a function of two independent uniform random variables on the interval $[0,1]$. Namely, for any two independent uniform random variables $X, Y$ on the interval $[0,1]$, we define that

$$
e(X, Y, \mathbf{p}_k, \hat{\mathbf{p}}_k) = \begin{cases} v_k^1 & \text{If } X \in [0, p_k^1) \\ v_k^c & \text{If } X \in [\sum_{m=1}^{c-1} p_k^m, \sum_{m=1}^{c} p_k^m) \text{ for some integer } c \in [2, n_k] \\ \emptyset & \text{If } X \geq \sum_{m=1}^{n_k} p_k^m \end{cases}
$$

We can easily check that, when $X$ and $Y$ are uniform random variables over the interval $[0,1]$, the random element $e(X, Y, \mathbf{p}_k, \hat{\mathbf{p}}_k)$ follows the multinoulli distribution Multi$(\mathbf{p}_k)$ over the community $\mathcal{V}_k := \{v_k^1, \ldots, v_k^{n_k}\}$, where $\mathbf{p}_k = (p_k^1, \ldots, p_k^{n_k}) \in \Delta_{n_k}$ and $\Pr\left(e(X, Y, \mathbf{p}_k, \hat{\mathbf{p}}_k) = v_k^m\right) = p_k^m$. Similarly, we also can generate a random element $\hat{e}$ from the community $\mathcal{V}_k$ according to the multinoulli distribution Multi$(\hat{\mathbf{p}}_k)$ if we set

$$
\hat{e}(X, Y, \mathbf{p}_k, \hat{\mathbf{p}}_k) = \begin{cases} e & \text{If } X < \sum_{m=1}^{n_k} p_k^m \\ v_k^1 & \text{If } X \geq \sum_{m=1}^{n_k} p_k^m \text{ and } Y \in [0, \frac{\hat{p}_k^1 - p_k^1}{1 - \sum_{m=1}^{n_k} p_k^m}) \\ v_k^c & \text{If } X \geq \sum_{m=1}^{n_k} p_k^m \text{ and } Y \in \left[\frac{\sum_{m=1}^{c-1}(\hat{p}_k^m - p_k^m)}{1 - \sum_{m=1}^{n_k} p_k^m}, \frac{\sum_{m=1}^{c}(\hat{p}_k^m - p_k^m)}{1 - \sum_{m=1}^{n_k} p_k^m}\right) \text{ for } c \in [2, n_k] \\ \emptyset & \text{If } X \geq \sum_{m=1}^{n_k} p_k^m \text{ and } Y \geq \sum_{m=1}^{n_k}(\hat{p}_k^m - p_k^m) \end{cases}
$$

From the definition of $e(X, Y, \mathbf{p}_k, \hat{\mathbf{p}}_k)$ and $\hat{e}(X, Y, \mathbf{p}_k, \hat{\mathbf{p}}_k)$, we can show that, for any fixed $X, Y$, $\{e(X, Y, \mathbf{p}_k, \hat{\mathbf{p}}_k)\} \subseteq \{\hat{e}(X, Y, \mathbf{p}_k, \hat{\mathbf{p}}_k)\}$. Thus, if we generate multiple independent pairs $(X_{\hat{k}}^{\hat{b}}, Y_{\hat{k}}^{\hat{b}})$ for any $\hat{b} \in [B_{\hat{k}}]$ and $\hat{k} \in [K]$, from the first point of Theorem 1, we have that,

$$
\frac{\partial F}{\partial p_k^m}(\mathbf{p}_1, \ldots, \mathbf{p}_K) = B_k\left(\mathbb{E}\left(f\left(v_k^m \middle| \cup_{(\hat{k},\hat{b}) \neq (k,1)} \{e_{\hat{k}}^{\hat{b}}(X_{\hat{k}}^{\hat{b}}, Y_{\hat{k}}^{\hat{b}}, \mathbf{p}_k, \hat{\mathbf{p}}_k)\}\right)\right)\right)
$$

$$
\frac{\partial F}{\partial p_k^m}(\hat{\mathbf{p}}_1, \ldots, \hat{\mathbf{p}}_K) = B_k\left(\mathbb{E}\left(f\left(v_k^m \middle| \cup_{(\hat{k},\hat{b}) \neq (k,1)} \{\hat{e}_{\hat{k}}^{\hat{b}}(X_{\hat{k}}^{\hat{b}}, Y_{\hat{k}}^{\hat{b}}, \mathbf{p}_k, \hat{\mathbf{p}}_k)\}\right)\right)\right).
$$

Due to $\{e_{\hat{k}}^{\hat{b}}(X_{\hat{k}}^{\hat{b}}, Y_{\hat{k}}^{\hat{b}}, \mathbf{p}_k, \hat{\mathbf{p}}_k)\} \subseteq \{\hat{e}_{\hat{k}}^{\hat{b}}(X_{\hat{k}}^{\hat{b}}, Y_{\hat{k}}^{\hat{b}}, \mathbf{p}_k, \hat{\mathbf{p}}_k)\}$, we can show that,

$$
\left(\cup_{(\hat{k},\hat{b}) \neq (k,1)} \{e_{\hat{k}}^{\hat{b}}(X_{\hat{k}}^{\hat{b}}, Y_{\hat{k}}^{\hat{b}}, \mathbf{p}_k, \hat{\mathbf{p}}_k)\}\right) \subseteq \left(\cup_{(\hat{k},\hat{b}) \neq (k,1)} \{\hat{e}_{\hat{k}}^{\hat{b}}(X_{\hat{k}}^{\hat{b}}, Y_{\hat{k}}^{\hat{b}}, \mathbf{p}_k, \hat{\mathbf{p}}_k)\}\right).
$$

As a result, from the definition of weakly DR-submodularity, we know that if $f$ is $\alpha$-weakly DR-submodular, we have $\frac{\partial F}{\partial p_k^m}(\mathbf{p}_1, \ldots, \mathbf{p}_K) \geq \alpha \frac{\partial F}{\partial p_k^m}(\hat{\mathbf{p}}_1, \ldots, \hat{\mathbf{p}}_K)$ such that $\nabla F(\mathbf{p}_1, \ldots, \mathbf{p}_K) \geq \alpha \nabla F(\hat{\mathbf{p}}_1, \ldots, \hat{\mathbf{p}}_K)$.

**4):** We prove the fourth point of Theorem 1. From the definition of Multinoulli Extension and previous definitions of $e(X, Y, \mathbf{p}_k, \hat{\mathbf{p}}_k)$ and $\hat{e}(X, Y, \mathbf{p}_k, \hat{\mathbf{p}}_k)$, we can infer that the Multinoulli Extension $F$ of set function $f$ satisfies the following relationships:

$$F(\mathbf{p}_1, \ldots, \mathbf{p}_K) = \mathbb{E}\left( f\left( \cup_{\hat{k}=1}^{K} \cup_{\hat{b}=1}^{B_k} \{e_{\hat{k}}^{\hat{b}}(X_{\hat{k}}^{\hat{b}}, Y_{\hat{k}}^{\hat{b}}, \mathbf{p}_k, \hat{\mathbf{p}}_k)\} \right) \right)$$

$$F(\hat{\mathbf{p}}_1, \ldots, \hat{\mathbf{p}}_K) = \mathbb{E}\left( f\left( \cup_{\hat{k}=1}^{K} \cup_{\hat{b}=1}^{B_k} \{\hat{e}_{\hat{k}}^{\hat{b}}(X_{\hat{k}}^{\hat{b}}, Y_{\hat{k}}^{\hat{b}}, \mathbf{p}_k, \hat{\mathbf{p}}_k)\} \right) \right).$$

As a result, we have

$$F(\hat{\mathbf{p}}_1, \ldots, \hat{\mathbf{p}}_K) - F(\mathbf{p}_1, \ldots, \mathbf{p}_K) = \mathbb{E}\left( f\left( \cup_{\hat{k}=1}^{K} \cup_{\hat{b}=1}^{B_k} \{\hat{e}_{\hat{k}}^{\hat{b}}(X_{\hat{k}}^{\hat{b}}, Y_{\hat{k}}^{\hat{b}}, \mathbf{p}_k, \hat{\mathbf{p}}_k)\} \right) - f\left( \cup_{\hat{k}=1}^{K} \cup_{\hat{b}=1}^{B_k} \{e_{\hat{k}}^{\hat{b}}(X_{\hat{k}}^{\hat{b}}, Y_{\hat{k}}^{\hat{b}}, \mathbf{p}_k, \hat{\mathbf{p}}_k)\} \right) \right).$$

If $f$ is $\gamma$-weakly submodular from below and $\hat{\mathbf{p}}_k \geq \mathbf{p}_k$ for any $k \in [K]$, we have that

$$\gamma\left( F(\hat{\mathbf{p}}_1, \ldots, \hat{\mathbf{p}}_K) - F(\mathbf{p}_1, \ldots, \mathbf{p}_K) \right) = \gamma \mathbb{E}\left( f\left( \cup_{\hat{k}=1}^{K} \cup_{\hat{b}=1}^{B_k} \{\hat{e}_{\hat{k}}^{\hat{b}}(X_{\hat{k}}^{\hat{b}}, Y_{\hat{k}}^{\hat{b}}, \mathbf{p}_k, \hat{\mathbf{p}}_k)\} \right) - f\left( \cup_{\hat{k}=1}^{K} \cup_{\hat{b}=1}^{B_k} \{e_{\hat{k}}^{\hat{b}}(X_{\hat{k}}^{\hat{b}}, Y_{\hat{k}}^{\hat{b}}, \mathbf{p}_k, \hat{\mathbf{p}}_k)\} \right) \right)$$

$$\leq \sum_{\bar{k}=1}^{K} \sum_{\bar{b}=1}^{B_{\bar{k}}} \mathbb{E}\left( f\left( \hat{e}_{\bar{k}}^{\bar{b}}(X_{\bar{k}}^{\bar{b}}, Y_{\bar{k}}^{\bar{b}}, \mathbf{p}_{\bar{k}}, \hat{\mathbf{p}}_{\bar{k}}) \middle| \cup_{\hat{k}=1}^{K} \cup_{\hat{b}=1}^{B_k} \{e_{\hat{k}}^{\hat{b}}(X_{\hat{k}}^{\hat{b}}, Y_{\hat{k}}^{\hat{b}}, \mathbf{p}_k, \hat{\mathbf{p}}_k)\} \right) \right),$$

where the final inequality follows from the $\gamma$-weakly submodular property of $f$, namely, for any two subsets $A \subseteq B \subseteq \mathcal{V}$, $\sum_{v \in B} f(v|A) = \sum_{v \in B \setminus A} f(v|A) \geq \gamma\left( f(B) - f(A) \right)$.

Then, if $X_{\bar{k}}^{\bar{b}} < \sum_{m=1}^{n_{\bar{k}}} p_{\bar{k}}^m$, we know that $\hat{e}_{\bar{k}}^{\bar{b}}(X_{\bar{k}}^{\bar{b}}, Y_{\bar{k}}^{\bar{b}}, \mathbf{p}_{\bar{k}}, \hat{\mathbf{p}}_{\bar{k}}) = e_{\bar{k}}^{\bar{b}}(X_{\bar{k}}^{\bar{b}}, Y_{\bar{k}}^{\bar{b}}, \mathbf{p}_{\bar{k}}, \hat{\mathbf{p}}_{\bar{k}})$ such that $f\left( \hat{e}_{\bar{k}}^{\bar{b}}(X_{\bar{k}}^{\bar{b}}, Y_{\bar{k}}^{\bar{b}}, \mathbf{p}_{\bar{k}}, \hat{\mathbf{p}}_{\bar{k}}) \middle| \cup_{\hat{k}=1}^{K} \cup_{\hat{b}=1}^{B_k} \{e_{\hat{k}}^{\hat{b}}(X_{\hat{k}}^{\hat{b}}, Y_{\hat{k}}^{\hat{b}}, \mathbf{p}_k, \hat{\mathbf{p}}_k)\} \right) = 0$. As for the case that $X_{\bar{k}}^{\bar{b}} \geq \sum_{m=1}^{n_{\bar{k}}} p_{\bar{k}}^m$, namely, $e_{\bar{k}}^{\bar{b}}(X_{\bar{k}}^{\bar{b}}, Y_{\bar{k}}^{\bar{b}}, \mathbf{p}_{\bar{k}}, \hat{\mathbf{p}}_{\bar{k}}) = \emptyset$, which means that we have the following equality:

$$\begin{aligned}
& f\left( \hat{e}_{\bar{k}}^{\bar{b}}(X_{\bar{k}}^{\bar{b}}, Y_{\bar{k}}^{\bar{b}}, \mathbf{p}_{\bar{k}}, \hat{\mathbf{p}}_{\bar{k}}) \middle| \cup_{\hat{k}=1}^{K} \cup_{\hat{b}=1}^{B_k} \{e_{\hat{k}}^{\hat{b}}(X_{\hat{k}}^{\hat{b}}, Y_{\hat{k}}^{\hat{b}}, \mathbf{p}_k, \hat{\mathbf{p}}_k)\} \right) \\
& = f\left( \hat{e}_{\bar{k}}^{\bar{b}}(X_{\bar{k}}^{\bar{b}}, Y_{\bar{k}}^{\bar{b}}, \mathbf{p}_{\bar{k}}, \hat{\mathbf{p}}_{\bar{k}}) \middle| \cup_{(\hat{k}, \hat{b}) \neq (\bar{k}, \bar{b})} \{e_{\hat{k}}^{\hat{b}}(X_{\hat{k}}^{\hat{b}}, Y_{\hat{k}}^{\hat{b}}, \mathbf{p}_k, \hat{\mathbf{p}}_k)\} \right).
\end{aligned} \tag{14}$$

Therefore, we have

$$\gamma\left( F(\hat{\mathbf{p}}_1, \ldots, \hat{\mathbf{p}}_K) - F(\mathbf{p}_1, \ldots, \mathbf{p}_K) \right)$$

$$\leq \sum_{\bar{k}=1}^{K} \sum_{\bar{b}=1}^{B_{\bar{k}}} \mathbb{E}\left( f\left( \hat{e}_{\bar{k}}^{\bar{b}}(X_{\bar{k}}^{\bar{b}}, Y_{\bar{k}}^{\bar{b}}, \mathbf{p}_{\bar{k}}, \hat{\mathbf{p}}_{\bar{k}}) \middle| \cup_{\hat{k}=1}^{K} \cup_{\hat{b}=1}^{B_k} \{e_{\hat{k}}^{\hat{b}}(X_{\hat{k}}^{\hat{b}}, Y_{\hat{k}}^{\hat{b}}, \mathbf{p}_k, \hat{\mathbf{p}}_k)\} \right) \right)$$

$$= \sum_{\bar{k}=1}^{K} \sum_{\bar{b}=1}^{B_{\bar{k}}} \mathbb{E}\left( \mathbb{E}\left( f\left( \hat{e}_{\bar{k}}^{\bar{b}}(X_{\bar{k}}^{\bar{b}}, Y_{\bar{k}}^{\bar{b}}, \mathbf{p}_{\bar{k}}, \hat{\mathbf{p}}_{\bar{k}}) \middle| \cup_{\hat{k}=1}^{K} \cup_{\hat{b}=1}^{B_k} \{e_{\hat{k}}^{\hat{b}}(X_{\hat{k}}^{\hat{b}}, Y_{\hat{k}}^{\hat{b}}, \mathbf{p}_k, \hat{\mathbf{p}}_k)\} \right) \middle| (X_k^b, Y_k^b), (k, b) \neq (\bar{k}, \bar{b}) \right) \right)$$

$$= \sum_{\bar{k}=1}^{K} \sum_{\bar{b}=1}^{B_{\bar{k}}} \mathbb{E}\left( \mathbb{E}\left( f\left( \hat{e}_{\bar{k}}^{\bar{b}}(X_{\bar{k}}^{\bar{b}}, Y_{\bar{k}}^{\bar{b}}, \mathbf{p}_{\bar{k}}, \hat{\mathbf{p}}_{\bar{k}}) \middle| \cup_{\hat{k}=1}^{K} \cup_{\hat{b}=1}^{B_k} \{e_{\hat{k}}^{\hat{b}}(X_{\hat{k}}^{\hat{b}}, Y_{\hat{k}}^{\hat{b}}, \mathbf{p}_k, \hat{\mathbf{p}}_k)\} \right) I(X_{\bar{k}}^{\bar{b}} < \sum_{m=1}^{n_{\bar{k}}} p_{\bar{k}}^m) \middle| (X_k^b, Y_k^b), (k, b) \neq (\bar{k}, \bar{b}) \right) \right)$$

$$+ \sum_{\bar{k}=1}^{K} \sum_{\bar{b}=1}^{B_{\bar{k}}} \mathbb{E}\left( \mathbb{E}\left( f\left( \hat{e}_{\bar{k}}^{\bar{b}}(X_{\bar{k}}^{\bar{b}}, Y_{\bar{k}}^{\bar{b}}, \mathbf{p}_{\bar{k}}, \hat{\mathbf{p}}_{\bar{k}}) \middle| \cup_{\hat{k}=1}^{K} \cup_{\hat{b}=1}^{B_k} \{e_{\hat{k}}^{\hat{b}}(X_{\hat{k}}^{\hat{b}}, Y_{\hat{k}}^{\hat{b}}, \mathbf{p}_k, \hat{\mathbf{p}}_k)\} \right) I(X_{\bar{k}}^{\bar{b}} \geq \sum_{m=1}^{n_{\bar{k}}} p_{\bar{k}}^m) \middle| (X_k^b, Y_k^b), (k, b) \neq (\bar{k}, \bar{b}) \right) \right)$$

$$= \sum_{\bar{k}=1}^{K} \sum_{\bar{b}=1}^{B_{\bar{k}}} \mathbb{E}\left( \mathbb{E}\left( f\left( \hat{e}_{\bar{k}}^{\bar{b}}(X_{\bar{k}}^{\bar{b}}, Y_{\bar{k}}^{\bar{b}}, \mathbf{p}_{\bar{k}}, \hat{\mathbf{p}}_{\bar{k}}) \middle| \cup_{\hat{k}=1}^{K} \cup_{\hat{b}=1}^{B_k} \{e_{\hat{k}}^{\hat{b}}(X_{\hat{k}}^{\hat{b}}, Y_{\hat{k}}^{\hat{b}}, \mathbf{p}_k, \hat{\mathbf{p}}_k)\} \right) I(X_{\bar{k}}^{\bar{b}} \geq \sum_{m=1}^{n_{\bar{k}}} p_{\bar{k}}^m) \middle| (X_k^b, Y_k^b), (k, b) \neq (\bar{k}, \bar{b}) \right) \right),$$

where the final equality comes from $f\left( \hat{e}_{\bar{k}}^{\bar{b}}(X_{\bar{k}}^{\bar{b}}, Y_{\bar{k}}^{\bar{b}}, \mathbf{p}_{\bar{k}}, \hat{\mathbf{p}}_{\bar{k}}) \middle| \cup_{\hat{k}=1}^{K} \cup_{\hat{b}=1}^{B_k} \{e_{\hat{k}}^{\hat{b}}(X_{\hat{k}}^{\hat{b}}, Y_{\hat{k}}^{\hat{b}}, \mathbf{p}_k, \hat{\mathbf{p}}_k)\} \right) = 0$ when $X_{\bar{k}}^{\bar{b}} < \sum_{m=1}^{n_{\bar{k}}} p_{\bar{k}}^m$. $\square$

Then, from Eq.(14), we also can show that

$$
\mathbb{E}\Bigg( f\Big(\hat{e}_{\bar{k}}^{\bar{b}}(X_{\bar{k}}^{\bar{b}}, Y_{\bar{k}}^{\bar{b}}, \mathbf{p}_{\bar{k}}, \hat{\mathbf{p}}_{\bar{k}})\Big| \cup_{\hat{k}=1}^{K} \cup_{\hat{b}=1}^{B_k} \{e_{\hat{k}}^{\hat{b}}(X_{\hat{k}}^{\hat{b}}, Y_{\hat{k}}^{\hat{b}}, \mathbf{p}_k, \hat{\mathbf{p}}_k)\}\Big) I(X_{\bar{k}}^{\bar{b}} \geq \sum_{m=1}^{n_{\bar{k}}} p_{\bar{k}}^m)\Big| (X_k^b, Y_k^b), (k,b) \neq (\bar{k}, \bar{b}) \Bigg)
$$

$$
= \sum_{c=1}^{n_{\bar{k}}} \Pr\Big(\hat{e}_{\bar{k}}^{\bar{b}}(X_{\bar{k}}^{\bar{b}}, Y_{\bar{k}}^{\bar{b}}, \mathbf{p}_{\bar{k}}, \hat{\mathbf{p}}_{\bar{k}}) = v_{\bar{k}}^c, X_{\bar{k}}^{\bar{b}} \geq \sum_{m=1}^{n_{\bar{k}}} p_{\bar{k}}^m\Big) f\Big(v_{\bar{k}}^c\Big| \cup_{(\hat{k},\hat{b}) \neq (\bar{k},\bar{b})} \{e_{\hat{k}}^{\hat{b}}(X_{\hat{k}}^{\hat{b}}, Y_{\hat{k}}^{\hat{b}}, \mathbf{p}_k, \hat{\mathbf{p}}_k)\}\Big) \tag{15}
$$

$$
= \sum_{c=1}^{n_{\bar{k}}} (\hat{p}_{\bar{k}}^c - p_{\bar{k}}^c) f\Big(v_{\bar{k}}^c\Big| \cup_{(\hat{k},\hat{b}) \neq (\bar{k},\bar{b})} \{e_{\hat{k}}^{\hat{b}}(X_{\hat{k}}^{\hat{b}}, Y_{\hat{k}}^{\hat{b}}, \mathbf{p}_k, \hat{\mathbf{p}}_k)\}\Big),
$$

where the equality comes from that

$$
\Pr\Big(\hat{e}_{\bar{k}}^{\bar{b}}(X_{\bar{k}}^{\bar{b}}, Y_{\bar{k}}^{\bar{b}}, \mathbf{p}_{\bar{k}}, \hat{\mathbf{p}}_{\bar{k}}) = v_{\bar{k}}^c, X_{\bar{k}}^{\bar{b}} \geq \sum_{m=1}^{n_{\bar{k}}} p_{\bar{k}}^m\Big) = \Pr\Big(X_{\bar{k}}^{\bar{b}} \geq \sum_{m=1}^{n_{\bar{k}}} p_{\bar{k}}^m, Y_{\bar{k}}^{\bar{b}} \in \Big[\frac{\sum_{m=1}^{c-1}(\hat{p}_{\bar{k}}^m - p_{\bar{k}}^m)}{1 - \sum_{m=1}^{n_{\bar{k}}} p_{\bar{k}}^m}, \frac{\sum_{m=1}^{c}(\hat{p}_{\bar{k}}^m - p_{\bar{k}}^m)}{1 - \sum_{m=1}^{n_{\bar{k}}} p_{\bar{k}}^m}\Big)\Big)
$$

$$
= (\hat{p}_{\bar{k}}^m - p_{\bar{k}}^m).
$$

As a result, we have

$$
\gamma\Big(F(\hat{\mathbf{p}}_1, \ldots, \hat{\mathbf{p}}_K) - F(\mathbf{p}_1, \ldots, \mathbf{p}_K)\Big)
$$

$$
= \sum_{\bar{k}=1}^{K} \sum_{\bar{b}=1}^{B_{\bar{k}}} \mathbb{E}\Bigg( \sum_{m=1}^{n_{\bar{k}}} (\hat{p}_{\bar{k}}^m - p_{\bar{k}}^m) f\Big(v_{\bar{k}}^m\Big| \cup_{(\hat{k},\hat{b}) \neq (\bar{k},\bar{b})} \{e_{\hat{k}}^{\hat{b}}(X_{\hat{k}}^{\hat{b}}, Y_{\hat{k}}^{\hat{b}}, \mathbf{p}_k, \hat{\mathbf{p}}_k)\}\Big) \Bigg)
$$

$$
= \sum_{\bar{k}=1}^{K} \sum_{\bar{b}=1}^{B_{\bar{k}}} \mathbb{E}\Bigg( \sum_{m=1}^{n_{\bar{k}}} (\hat{p}_{\bar{k}}^m - p_{\bar{k}}^m) f\Big(v_{\bar{k}}^m\Big| \cup_{(\hat{k},\hat{b}) \neq (\bar{k},1)} \{e_{\hat{k}}^{\hat{b}}(X_{\hat{k}}^{\hat{b}}, Y_{\hat{k}}^{\hat{b}}, \mathbf{p}_k, \hat{\mathbf{p}}_k)\}\Big) \Bigg)
$$

$$
= \sum_{\bar{k}=1}^{K} \sum_{m=1}^{n_{\bar{k}}} (\hat{p}_{\bar{k}}^m - p_{\bar{k}}^m) B_{\bar{k}} \mathbb{E}\Bigg( f\Big(v_{\bar{k}}^m\Big| \cup_{(\hat{k},\hat{b}) \neq (\bar{k},1)} \{e_{\hat{k}}^{\hat{b}}(X_{\hat{k}}^{\hat{b}}, Y_{\hat{k}}^{\hat{b}}, \mathbf{p}_k, \hat{\mathbf{p}}_k)\}\Big) \Bigg)
$$

$$
= \sum_{\bar{k}=1}^{K} \sum_{m=1}^{n_{\bar{k}}} (\hat{p}_{\bar{k}}^m - p_{\bar{k}}^m) \frac{\partial F}{\partial p_{\bar{k}}^m}(\mathbf{p}_1, \ldots, \mathbf{p}_K) = \Big\langle (\hat{\mathbf{p}}_1, \ldots, \hat{\mathbf{p}}_K) - (\mathbf{p}_1, \ldots, \mathbf{p}_K), \nabla F(\mathbf{p}_1, \ldots, \mathbf{p}_K) \Big\rangle,
$$

where the second equality follows from the independence of $e_{\hat{k}}^{\hat{b}}$ for any $\hat{b} \in [B_{\hat{k}}]$ and $k \in [K]$.

### C.2. Proof of Theorem 2

In this section, we verify the Theorem 2.

*Proof.* At first, we recall that $\mathcal{V}_k := \{v_k^1, \ldots, v_k^{n_k}\}$ for any $k \in [K]$. Therefore, for any subset $S$ within the partition constraint of problem (1), we assume $|S \cap \mathcal{V}_k| = s_k \leq B_k$ and we can represent each $S \cap \mathcal{V}_k$ as

$$
S \cap \mathcal{V}_k = \{v_k^{m_k^1}, \ldots, v_k^{m_k^{s_k}}\},
$$

where $m_k^{b_1} \neq m_k^{b_2} \in [n_k]$ for any $b_1, b_2 \in [s_k]$ for any $k \in [K]$.

As a result, we can rewrite that

$$
\Bigg\langle \sum_{k=1}^{K} \frac{\mathbf{1}_{S \cap \mathcal{V}_k}}{B_k}, \nabla F(\mathbf{p}_1, \ldots, \mathbf{p}_K) \Bigg\rangle = \sum_{k=1}^{K} \sum_{b=1}^{s_k} \frac{1}{B_k} \frac{\partial F}{\partial p_k^{m_k^b}}(\mathbf{p}_1, \ldots, \mathbf{p}_K)
$$

$$
= \sum_{k=1}^{K} \sum_{b=1}^{s_k} \Bigg( \mathbb{E}_{e_{\hat{k}}^{\hat{b}} \sim Multi(\mathbf{p}_{\hat{k}})}\Big( f\Big(v_k^{m_k^b}\Big| \cup_{(\hat{k},\hat{b}) \neq (k,1)} \{e_{\hat{k}}^{\hat{b}}\}\Big)\Big) \Bigg) \tag{16}
$$

$$
= \sum_{k=1}^{K} \sum_{b=1}^{s_k} \Bigg( \mathbb{E}_{e_{\hat{k}}^{\hat{b}} \sim Multi(\mathbf{p}_{\hat{k}})}\Big( f\Big(v_k^{m_k^b}\Big| \cup_{(\hat{k},\hat{b}) \neq (k,b)} \{e_{\hat{k}}^{\hat{b}}\}\Big)\Big) \Bigg),
$$

where the final equality follows from that, for any fixed $\hat{k} \in [K]$, the elements $e_{\hat{k}}^b, \forall b \in [B_{\hat{k}}]$ are independently drawn from the same multinoulli distribution Multi($\mathbf{p}_{\hat{k}}$).

When the set function $f$ is $\alpha$-weakly monotone DR-submodular, we can show that

$$
\begin{aligned}
&\left\langle \sum_{k=1}^K \frac{\mathbf{1}_{S \cap \mathcal{V}_k}}{B_k}, \nabla F(\mathbf{p}_1, \dots, \mathbf{p}_K) \right\rangle \\
&= \sum_{k=1}^K \sum_{b=1}^{s_k} \left( \mathbb{E}_{e_{\hat{k}}^{\hat{b}} \sim Multi(\mathbf{p}_{\hat{k}})} \left( f\left( v_k^{m_k^b} \middle| \cup_{(\hat{k},\hat{b}) \neq (k,b)} \{e_{\hat{k}}^{\hat{b}}\} \right) \right) \right) \\
&\geq \alpha \sum_{k=1}^K \sum_{b=1}^{s_k} \left( \mathbb{E}_{e_{\hat{k}}^{\hat{b}} \sim Multi(\mathbf{p}_{\hat{k}})} \left( f\left( v_k^{m_k^b} \middle| \cup_{\hat{k}=1}^K \cup_{\hat{b}=1}^{B_{\hat{k}}} \{e_{\hat{k}}^{\hat{b}}\} \bigcup \left( \cup_{(\bar{k},\bar{b}) < (k,b)} \{v_{\bar{k}}^{m_{\bar{k}}^{\bar{b}}}\} \right) \right) \right) \right) \\
&= \alpha \left( \mathbb{E}_{e_{\hat{k}}^{\hat{b}} \sim Multi(\mathbf{p}_{\hat{k}})} \left( f\left( \cup_{k=1}^K \cup_{b=1}^{s_k} \{v_k^{m_k^b}\} \middle| \cup_{\hat{k}=1}^K \cup_{\hat{b}=1}^{B_{\hat{k}}} \{e_{\hat{k}}^{\hat{b}}\} \right) \right) \right) \\
&\geq \alpha \left( \mathbb{E}_{e_{\hat{k}}^{\hat{b}} \sim Multi(\mathbf{p}_{\hat{k}})} \left( f\left( \cup_{k=1}^K \cup_{b=1}^{s_k} \{v_k^{m_k^b}\} \right) - f\left( \cup_{\hat{k}=1}^K \cup_{\hat{b}=1}^{B_{\hat{k}}} \{e_{\hat{k}}^{\hat{b}}\} \right) \right) \right) \quad \text{(Monotonicity)} \\
&= \alpha \left( f(S) - F(\mathbf{p}_1, \dots, \mathbf{p}_K) \right),
\end{aligned}
\tag{17}
$$

where the first inequality follows from the $\alpha$-weakly DR-submodularity, namely, $f(v|A) \geq \alpha f(v|B)$ for any two subsets $A \subseteq B \subseteq \mathcal{V}$ and the partially ordered set $\{(\bar{k},\bar{b}) < (k,b)\} = \{(\bar{k},\bar{b}) | \bar{k} < k \text{ or } \bar{b} < b \text{ when } \bar{k} = k\}$.

As for the settings that the set function $f$ is $(\gamma, \beta)$-weakly monotone submodular, we firstly can show that for any two elements $e_1, e_2 \in \mathcal{V}$ and the subset $B \subseteq \mathcal{V}$, we have that

$$
\gamma \left( f(e_1 | B \cup \{e_2\}) + f(e_2|B) \right) = \gamma \left( f(B \cup \{e_1, e_2\}) - f(B) \right) \leq f(e_1|B) + f(e_2|B),
$$

such that

$$
f(e_1|B) \geq \gamma f(e_1 | B \cup \{e_2\}) - (1-\gamma) f(e_2|B).
\tag{18}
$$

From Eq.(18), we can show that

$$
\begin{aligned}
&\left\langle \sum_{k=1}^K \frac{\mathbf{1}_{S \cap \mathcal{V}_k}}{B_k}, \nabla F(\mathbf{p}_1, \dots, \mathbf{p}_K) \right\rangle \\
&= \sum_{k=1}^K \sum_{b=1}^{s_k} \left( \mathbb{E}_{e_{\hat{k}}^{\hat{b}} \sim Multi(\mathbf{p}_{\hat{k}})} \left( f\left( v_k^{m_k^b} \middle| \cup_{(\hat{k},\hat{b}) \neq (k,b)} \{e_{\hat{k}}^{\hat{b}}\} \right) \right) \right) \\
&= \sum_{k=1}^K \sum_{b=1}^{s_k} \left( \mathbb{E}_{e_{\hat{k}}^{\hat{b}} \sim Multi(\mathbf{p}_{\hat{k}})} \left( \gamma f\left( v_k^{m_k^b} \middle| \cup_{\hat{k}=1}^K \cup_{\hat{b}=1}^{B_{\hat{k}}} \{e_{\hat{k}}^{\hat{b}}\} \right) - (1-\gamma) f\left( e_k^b \middle| \cup_{(\hat{k},\hat{b}) \neq (k,b)} \{e_{\hat{k}}^{\hat{b}}\} \right) \right) \right)
\end{aligned}
\tag{19}
$$

Then, from the $\gamma$-weakly submodularity,

$$
\begin{aligned}
&\left( \mathbb{E}_{e_{\hat{k}}^{\hat{b}} \sim Multi(\mathbf{p}_{\hat{k}})} \left( f\left( v_k^{m_k^b} \middle| \cup_{\hat{k}=1}^K \cup_{\hat{b}=1}^{B_{\hat{k}}} \{e_{\hat{k}}^{\hat{b}}\} \right) \right) \right) \\
&\geq \gamma \mathbb{E}_{e_{\hat{k}}^{\hat{b}} \sim Multi(\mathbf{p}_{\hat{k}})} \left( f\left( \cup_{k=1}^K \cup_{b=1}^{s_k} \{v_k^{m_k^b}\} \middle| \cup_{\hat{k}=1}^K \cup_{\hat{b}=1}^{B_{\hat{k}}} \{e_{\hat{k}}^{\hat{b}}\} \right) \right) \\
&\geq \gamma \left( f(S) - F(\mathbf{p}_1, \dots, \mathbf{p}_K) \right),
\end{aligned}
\tag{20}
$$

where the final inequality follows from the monotonicity.

Furthermore, from the $\beta$-weakly upper submodularity, we also have,

$$\sum_{k=1}^{K}\sum_{b=1}^{s_k}\left(\mathbb{E}_{e_{\hat{k}}^{\hat{b}}\sim Multi(\mathbf{p}_{\hat{k}})}\left(f\left(e_k^b\Big|\cup_{(\hat{k},\hat{b})\neq(k,b)}\{e_{\hat{k}}^{\hat{b}}\}\right)\right)\right)$$
$$\leq \sum_{k=1}^{K}\sum_{b=1}^{B_k}\left(\mathbb{E}_{e_{\hat{k}}^{\hat{b}}\sim Multi(\mathbf{p}_{\hat{k}})}\left(f\left(e_k^b\Big|\cup_{(\hat{k},\hat{b})\neq(k,b)}\{e_{\hat{k}}^{\hat{b}}\}\right)\right)\right) \qquad (21)$$
$$\leq \beta\mathbb{E}_{e_{\hat{k}}^{\hat{b}}\sim Multi(\mathbf{p}_{\hat{k}})}\left(f\left(\cup_{\hat{k}=1}^{K}\cup_{\hat{b}=1}^{B_{\hat{k}}}\{e_{\hat{k}}^{\hat{b}}\}\right)-f(\emptyset)\right)\leq \beta F(\mathbf{p}_1,\dots,\mathbf{p}_K),$$

where the second inequality follows from the definition of $\beta$-upper submodularity, namely, Eq.(3).

Merging Eq.(21) and Eq.(20) into Eq.(19), we have that

$$\left\langle\sum_{k=1}^{K}\frac{\mathbf{1}_{S\cap\mathcal{V}_k}}{B_k},\nabla F(\mathbf{p}_1,\dots,\mathbf{p}_K)\right\rangle$$
$$=\sum_{k=1}^{K}\sum_{b=1}^{s_k}\left(\mathbb{E}_{e_{\hat{k}}^{\hat{b}}\sim Multi(\mathbf{p}_{\hat{k}})}\left(\gamma f\left(v_k^{m_k^b}\Big|\cup_{\hat{k}=1}^{K}\cup_{\hat{b}=1}^{B_{\hat{k}}}\{e_{\hat{k}}^{\hat{b}}\}\right)-(1-\gamma)f\left(e_k^b\Big|\cup_{(\hat{k},\hat{b})\neq(k,b)}\{e_{\hat{k}}^{\hat{b}}\}\right)\right)\right)$$
$$\geq \gamma^2(f(S)-F(\mathbf{p}_1,\dots,\mathbf{p}_K))-(1-\gamma)\beta F(\mathbf{p}_1,\dots,\mathbf{p}_K)=\gamma^2 f(S)-(\beta(1-\gamma)+\gamma^2)F(\mathbf{p}_1,\dots,\mathbf{p}_K).$$

$\square$

### C.3. Proof of Theorem 3

In this section, we verify the Theorem 3.

*Proof.* Firstly, from the fifth equality in Eq.(13), we know that

$$\frac{\partial F}{\partial p_k^m}(\mathbf{p}_1,\dots,\mathbf{p}_K)=B_k\left(\sum_{e_{\hat{k}}^{\hat{b}}\in\mathcal{V}_{\hat{k}}\cup\{\emptyset\},(\hat{k},\hat{b})\neq(k,1)}\left(f\left(v_k^m|\cup_{(\hat{k},\hat{b})\neq(k,1)}\{e_{\hat{k}}^{\hat{b}}\}\right)\prod_{(\hat{k},\hat{b})\neq(k,1)}\Pr(e_{\hat{k}}^{\hat{b}}|\mathbf{p}_{\hat{k}})\right)\right).$$

Therefore, if $k_1\neq k_2\in[K]$, for any $m_1\in[n_{k_1}]$ and $m_2\in[n_{k_2}]$,

$$\frac{\partial^2 F}{\partial p_{k_1}^{m_1}\partial p_{k_2}^{m_2}}(\mathbf{p}_1,\dots,\mathbf{p}_K)$$
$$=B_{k_1}\left(\sum_{e_{\hat{k}}^{\hat{b}}\in\mathcal{V}_{\hat{k}}\cup\{\emptyset\},(\hat{k},\hat{b})\neq(k_1,1)}\left(f\left(v_{k_1}^{m_1}|\cup_{(\hat{k},\hat{b})\neq(k_1,1)}\{e_{\hat{k}}^{\hat{b}}\}\right)\frac{\partial\left(\prod_{(\hat{k},\hat{b})\neq(k_1,1)}\Pr(e_{\hat{k}}^{\hat{b}}|\mathbf{p}_{\hat{k}})\right)}{\partial p_{k_2}^{m_2}}\right)\right)$$
$$=B_{k_1}\left(\sum_{e_{\hat{k}}^{\hat{b}}\in\mathcal{V}_{\hat{k}}\cup\{\emptyset\},(\hat{k},\hat{b})\neq(k_1,1)}\left(f\left(v_{k_1}^{m_1}|\cup_{(\hat{k},\hat{b})\neq(k_1,1)}\{e_{\hat{k}}^{\hat{b}}\}\right)\sum_{\hat{b}_1\in[B_{k_2}]}\frac{\partial\Pr(e_{k_2}^{\hat{b}_1}|\mathbf{p}_{k_2})}{\partial p_{k_2}^{m_2}}\left(\prod_{(\hat{k},\hat{b})\neq\{(k_1,1),(k_2,\hat{b}_1)\}}\Pr(e_{\hat{k}}^{\hat{b}}|\mathbf{p}_{\hat{k}})\right)\right)\right)$$
$$=B_{k_1}\sum_{\hat{b}_1\in[B_{k_2}]}\left(\sum_{e_{\hat{k}}^{\hat{b}}\in\mathcal{V}_{\hat{k}}\cup\{\emptyset\},(\hat{k},\hat{b})\neq(k_1,1)}\left(f\left(v_{k_1}^{m_1}|\cup_{(\hat{k},\hat{b})\neq(k_1,1)}\{e_{\hat{k}}^{\hat{b}}\}\right)\frac{\partial\Pr(e_{k_2}^{\hat{b}_1}|\mathbf{p}_{k_2})}{\partial p_{k_2}^{m_2}}\left(\prod_{(\hat{k},\hat{b})\neq\{(k_1,1),(k_2,\hat{b}_1)\}}\Pr(e_{\hat{k}}^{\hat{b}}|\mathbf{p}_{\hat{k}})\right)\right)\right)$$
$$=B_{k_1}B_{k_2}\left(\sum_{(\hat{k},\hat{b})\neq\{(k_1,1),(k_2,1)\}}\left(\sum_{e_{k_2}^1\in\mathcal{V}_{k_2}\cup\{\emptyset\}}f\left(v_{k_1}^{m_1}|\cup_{(\hat{k},\hat{b})\neq(k_1,1)}\{e_{\hat{k}}^{\hat{b}}\}\right)\frac{\partial\Pr(e_{k_2}^1|\mathbf{p}_{k_2})}{\partial p_{k_2}^{m_2}}\left(\prod_{(\hat{k},\hat{b})\neq\{(k_1,1),(k_2,1)\}}\Pr(e_{\hat{k}}^{\hat{b}}|\mathbf{p}_{\hat{k}})\right)\right)\right).$$

Due to that $\Pr(v_{\hat{k}}^m|\mathbf{p}_{\hat{k}})=p_{\hat{k}}^m$ and $\Pr(\emptyset|\mathbf{p}_{\hat{k}})=1-\sum_{m=1}^{n_{\hat{k}}}p_{\hat{k}}^m$, we can show that

$$\sum_{e_{k_2}^1\in\mathcal{V}_{k_2}\cup\{\emptyset\}}f\left(v_{k_1}^{m_1}|\cup_{(\hat{k},\hat{b})\neq(k_1,1)}\{e_{\hat{k}}^{\hat{b}}\}\right)\frac{\partial\Pr(e_{k_2}^1|\mathbf{p}_{k_2})}{\partial p_{k_2}^{m_2}}$$
$$=f\left(v_{k_1}^{m_1}|\cup_{(\hat{k},\hat{b})\neq\{(k_1,1),(k_2,1)\}}\{e_{\hat{k}}^{\hat{b}}\}\bigcup\{v_{k_2}^{m_2}\}\right)-f\left(v_{k_1}^{m_1}|\cup_{(\hat{k},\hat{b})\neq\{(k_1,1),(k_2,1)\}}\{e_{\hat{k}}^{\hat{b}}\}\right).$$

Therefore, if we set $S = \cup_{(\hat{k},\hat{b}) \neq \{(k_1,1),(k_2,1)\}} \{e_{\hat{k}}^{\hat{b}}\}$, we can show that

$$
\frac{\partial^2 F}{\partial p_{k_1}^{m_1} \partial p_{k_2}^{m_2}}(\mathbf{p}_1, \ldots, \mathbf{p}_K)
$$

$$
= B_{k_1} B_{k_2} \left( \sum_{(\hat{k},\hat{b}) \neq \{(k_1,1),(k_2,1)\}} \left( \sum_{e_{k_2}^1 \in \mathcal{V}_{k_2} \cup \{\emptyset\}} f\left(v_{k_1}^{m_1} | \cup_{(\hat{k},\hat{b}) \neq (k_1,1)} \{e_{\hat{k}}^{\hat{b}}\}\right) \frac{\partial \mathrm{Pr}(e_{k_2}^1 | \mathbf{p}_{k_2})}{\partial p_{k_2}^{m_2}} \left( \prod_{(\hat{k},\hat{b}) \neq \{(k_1,1),(k_2,1)\}} \mathrm{Pr}(e_{\hat{k}}^{\hat{b}} | \mathbf{p}_{\hat{k}}) \right) \right) \right)
$$

$$
= B_{k_1} B_{k_2} \left( \sum_{e_{\hat{k}}^{\hat{b}} \in \mathcal{V}_{\hat{k}} \cup \{\emptyset\}, (\hat{k},\hat{b}) \neq \{(k_1,1),(k_2,1)\}} \left( f\left(v_{k_1}^{m_1} | S \cup \{v_{k_2}^{m_2}\}\right) - f\left(v_{k_1}^{m_1} | S\right) \left( \prod_{(\hat{k},\hat{b}) \neq \{(k_1,1),(k_2,1)\}} \mathrm{Pr}(e_{\hat{k}}^{\hat{b}} | \mathbf{p}_{\hat{k}}) \right) \right) \right)
$$

$$
= B_{k_1} B_{k_2} \mathbb{E}_{e_{\hat{k}}^{\hat{b}} \sim \mathrm{Multi}(\mathbf{p}_{\hat{k}})} \left( f\left(v_{k_1}^{m_1} | S \cup \{v_{k_2}^{m_2}\}\right) - f\left(v_{k_1}^{m_1} | S\right) \right),
$$

where the final equality comes from that the subset $S = \cup_{(\hat{k},\hat{b}) \neq \{(k_1,1),(k_2,1)\}} \{e_{\hat{k}}^{\hat{b}}\}$ is unrelated with the random elements $e_{k_1}^1$ and $e_{k_2}^1$.

When $k_1 = k_2 = k \in [K]$ and $B_k = 1$, it is easy to verify that $\frac{\partial^2 F}{\partial p_k^{m_1} \partial p_k^{m_2}}(\mathbf{p}_1, \ldots, \mathbf{p}_K) = 0$. As for $B_k \geq 2$, we have that

$$
\frac{\partial^2 F}{\partial p_k^{m_1} \partial p_k^{m_2}}(\mathbf{p}_1, \ldots, \mathbf{p}_K)
$$

$$
= B_k \left( \sum_{e_{\hat{k}}^{\hat{b}} \in \mathcal{V}_{\hat{k}} \cup \{\emptyset\}, (\hat{k},\hat{b}) \neq (k,1)} \left( f\left(v_k^{m_1} | \cup_{(\hat{k},\hat{b}) \neq (k,1)} \{e_{\hat{k}}^{\hat{b}}\}\right) \frac{\partial \left( \prod_{(\hat{k},\hat{b}) \neq (k,1)} \mathrm{Pr}(e_{\hat{k}}^{\hat{b}} | \mathbf{p}_{\hat{k}}) \right)}{\partial p_k^{m_2}} \right) \right)
$$

$$
= B_k \left( \sum_{e_{\hat{k}}^{\hat{b}} \in \mathcal{V}_{\hat{k}} \cup \{\emptyset\}, (\hat{k},\hat{b}) \neq (k,1)} \left( f\left(v_k^{m_1} | \cup_{(\hat{k},\hat{b}) \neq (k,1)} \{e_{\hat{k}}^{\hat{b}}\}\right) \sum_{2 \leq \hat{b}_1 \leq B_k} \frac{\partial \mathrm{Pr}(e_k^{\hat{b}_1} | \mathbf{p}_k)}{\partial p_k^{m_2}} \left( \prod_{(\hat{k},\hat{b}) \neq \{(k,1),(k,\hat{b}_1)\}} \mathrm{Pr}(e_{\hat{k}}^{\hat{b}} | \mathbf{p}_{\hat{k}}) \right) \right) \right)
$$

$$
= B_k \sum_{2 \leq \hat{b}_1 \leq B_k} \left( \sum_{e_{\hat{k}}^{\hat{b}} \in \mathcal{V}_{\hat{k}} \cup \{\emptyset\}, (\hat{k},\hat{b}) \neq (k,1)} \left( f\left(v_k^{m_1} | \cup_{(\hat{k},\hat{b}) \neq (k,1)} \{e_{\hat{k}}^{\hat{b}}\}\right) \frac{\partial \mathrm{Pr}(e_k^{\hat{b}_1} | \mathbf{p}_k)}{\partial p_k^{m_2}} \left( \prod_{(\hat{k},\hat{b}) \neq \{(k,1),(k,\hat{b}_1)\}} \mathrm{Pr}(e_{\hat{k}}^{\hat{b}} | \mathbf{p}_{\hat{k}}) \right) \right) \right)
$$

$$
= B_k(B_k - 1) \left( \sum_{e_{\hat{k}}^{\hat{b}} \in \mathcal{V}_{\hat{k}} \cup \{\emptyset\}, (\hat{k},\hat{b}) \neq (k,1)} \left( f\left(v_k^{m_1} | \cup_{(\hat{k},\hat{b}) \neq (k,1)} \{e_{\hat{k}}^{\hat{b}}\}\right) \frac{\partial \mathrm{Pr}(e_k^2 | \mathbf{p}_k)}{\partial p_k^{m_2}} \left( \prod_{(\hat{k},\hat{b}) \neq \{(k,1),(k,2)\}} \mathrm{Pr}(e_{\hat{k}}^{\hat{b}} | \mathbf{p}_{\hat{k}}) \right) \right) \right)
$$

$$
= B_k(B_k - 1) \left( \sum_{(\hat{k},\hat{b}) \neq \{(k,1),(k,2)\}} \left( \sum_{e_k^2 \in \mathcal{V}_k \cup \{\emptyset\}} f\left(v_k^{m_1} | \cup_{(\hat{k},\hat{b}) \neq (k,1)} \{e_{\hat{k}}^{\hat{b}}\}\right) \frac{\partial \mathrm{Pr}(e_k^2 | \mathbf{p}_k)}{\partial p_k^{m_2}} \left( \prod_{(\hat{k},\hat{b}) \neq \{(k,1),(k,2)\}} \mathrm{Pr}(e_{\hat{k}}^{\hat{b}} | \mathbf{p}_{\hat{k}}) \right) \right) \right).
$$

Also, because $\mathrm{Pr}(v_{\hat{k}}^m | \mathbf{p}_{\hat{k}}) = p_{\hat{k}}^m$ and $\mathrm{Pr}(\emptyset | \mathbf{p}_{\hat{k}}) = 1 - \sum_{m=1}^{n_{\hat{k}}} p_{\hat{k}}^m$, we can show that

$$
\sum_{e_k^2 \in \mathcal{V}_k \cup \{\emptyset\}} f\left(v_k^{m_1} | \cup_{(\hat{k},\hat{b}) \neq (k,1)} \{e_{\hat{k}}^{\hat{b}}\}\right) \frac{\partial \mathrm{Pr}(e_k^2 | \mathbf{p}_k)}{\partial p_{k_2}^{m_2}}
$$

$$
= f\left(v_k^{m_1} | \cup_{(\hat{k},\hat{b}) \neq \{(k,1),(k,2)\}} \{e_{\hat{k}}^{\hat{b}}\} \bigcup \{v_k^{m_2}\}\right) - f\left(v_k^{m_1} | \cup_{(\hat{k},\hat{b}) \neq \{(k,1),(k,2)\}} \{e_{\hat{k}}^{\hat{b}}\}\right).
$$

As a result, if we set $S = \cup_{(\hat{k},\hat{b}) \neq \{(k,1),(k,2)\}} \{e_{\hat{k}}^{\hat{b}}\}$, we have that

$$
\frac{\partial^2 F}{\partial p_k^{m_1} \partial p_k^{m_2}}(\mathbf{p}_1, \ldots, \mathbf{p}_K)
$$

$$
= B_k(B_k - 1) \left( \sum_{e_{\hat{k}}^{\hat{b}} \in \mathcal{V}_{\hat{k}} \cup \{\emptyset\}, (\hat{k},\hat{b}) \neq \{(k,1),(k,2)\}} \left( \left( f\left(v_k^{m_1} | S \cup \{v_k^{m_2}\}\right) - f\left(v_k^{m_1} | S\right) \right) \left( \prod_{(\hat{k},\hat{b}) \neq \{(k,1),(k,2)\}} \mathrm{Pr}(e_{\hat{k}}^{\hat{b}} | \mathbf{p}_{\hat{k}}) \right) \right) \right)
$$

$$
= (B_k^2 - B_k) \mathbb{E}_{e_{\hat{k}}^{\hat{b}} \sim \mathrm{Multi}(\mathbf{p}_{\hat{k}})} \left( f\left(v_k^{m_1} | S \cup \{v_k^{m_2}\}\right) - f\left(v_k^{m_1} | S\right) \right),
$$

where the final equality comes from that the subset $S = \cup_{(\hat{k},\hat{b}) \neq \{(k,1),(k,2)\}} \{e_{\hat{k}}^{\hat{b}}\}$ is unrelated with the random elements $e_k^1$ and $e_k^2$. $\qquad\square$

## D. Stochastic Variant of Continuous Greedy Method for Mutinoulli Extension

### D.1. Proof of Theorem 4

In this section, we prove the Theorem 4.

Before going into the details, we firstly bound each second-order derivative of our proposed ME $F$, that is to say,

**Lemma 1.** *Given a monotone set function $f : 2^{\mathcal{V}} \to \mathbb{R}_+$, if we denote the maximum marginal value of $f$ as $M_f = \max_{S \subseteq \mathcal{V}, e \in \mathcal{V} \setminus S} \left( f(e|S) \right)$, we have that, for any $k_1, k_2 \in [K]$, $m_1 \in [n_{k_1}]$ and $m_2 \in [n_{k_2}]$,*

$$\left| \frac{\partial^2 F}{\partial p_{k_1}^{m_1} \partial p_{k_2}^{m_2}}(\mathbf{p}_1, \ldots, \mathbf{p}_K) \right| \leq \bar{B}^2 M_f,$$

*where $(\mathbf{p}_1, \ldots, \mathbf{p}_K) \in \prod_{k=1}^{K} \Delta_{n_k}$ and $\bar{B} = \max_{k=1}^{K} B_k$ is the maximum budget over the $K$ communities $\{\mathcal{V}_1, \ldots, \mathcal{V}_K\}$.*

*Proof.* From the Theorem 3, we know that, If $k_1 \neq k_2 \in [K]$, the second-order derivative of the Multinoulli Extension $F$ at any point $(\mathbf{p}_1, \ldots, \mathbf{p}_K) \in \prod_{k=1}^{K} \Delta_{n_k}$ can be written as follows:

$$\frac{\partial^2 F}{\partial p_{k_1}^{m_1} \partial p_{k_2}^{m_2}}(\mathbf{p}_1, \ldots, \mathbf{p}_K) = B_{k_1} B_{k_2} \mathbb{E}_{e_{\hat{k}}^{\hat{b}}} \left( f\left(v_{k_1}^{m_1} | S \cup \{v_{k_2}^{m_2}\}\right) - f\left(v_{k_1}^{m_1} | S\right) \right),$$

where $S = \cup_{(\hat{k},\hat{b}) \neq \{(k_1,1),(k_2,1)\}} \{e_{\hat{k}}^{\hat{b}}\}$ and each $e_{\hat{k}}^{\hat{b}}$ is drawn from the multinoulli distribution $\text{Multi}(\mathbf{p}_{\hat{k}})$. Furthermore, for any monotone set function $f$ and any subset $S \subseteq \mathcal{V}$, we can easily know that:

$$-M_f \leq -f\left(v_{k_1}^{m_1}|S\right) \leq f\left(v_{k_1}^{m_1}|S \cup \{v_{k_2}^{m_2}\}\right) - f\left(v_{k_1}^{m_1}|S\right) \leq f\left(v_{k_1}^{m_1}|S \cup \{v_{k_2}^{m_2}\}\right) \leq M_f$$

such that, when $k_1 \neq k_2 \in [K]$,

$$\left| \frac{\partial^2 F}{\partial p_{k_1}^{m_1} \partial p_{k_2}^{m_2}}(\mathbf{p}_1, \ldots, \mathbf{p}_K) \right| \leq B_{k_1} B_{k_2} M_f \leq \bar{B}^2 M_f.$$

Similarly, we also can verify that when $k_1 = k_2 = k \in [K]$, for any $m_1 \in [n_{k_1}]$ and $m_2 \in [n_{k_2}]$, we have that $\left| \frac{\partial^2 F}{\partial p_k^{m_1} \partial p_k^{m_2}}(\mathbf{p}_1, \ldots, \mathbf{p}_K) \right| \leq \bar{B}^2 M_f$. $\qquad\square$

Similarly, we also can verify that the estimations of the second-order derivative of Multinoulli Extension $F$ in Remark 6 are also bounded by $\bar{B}^2 M_f$, that is to say,

**Lemma 2.** *Given a monotone set function $f : 2^{\mathcal{V}} \to \mathbb{R}_+$, if we denote the maximum marginal value of $f$ as $M_f = \max_{S \subseteq \mathcal{V}, e \in \mathcal{V} \setminus S} \left( f(e|S) \right)$, we can infer that each second-order estimators in Remark 6 is also bounded by $\bar{B}^2 M_f$, i.e., for any $k_1, k_2 \in [K]$, $m_1 \in [n_{k_1}]$ and $m_2 \in [n_{k_2}]$, we have that*

$$\left| \frac{\widehat{\partial^2 F}}{\partial p_{k_1}^{m_1} \partial p_{k_2}^{m_2}}((\mathbf{p}_1, \ldots, \mathbf{p}_K)) \right| \leq \bar{B}^2 M_f$$

*where $(\mathbf{p}_1, \ldots, \mathbf{p}_K) \in \prod_{k=1}^{K} \Delta_{n_k}$ and $\bar{B} = \max_{k=1}^{K} B_k$ is the maximum budget over the $K$ communities $\{\mathcal{V}_1, \ldots, \mathcal{V}_K\}$.*

As a result, we also show that

**Lemma 3.** *Given a monotone set function $f : 2^{\mathcal{V}} \to \mathbb{R}_+$, if we denote the maximum marginal value of $f$ as $M_f = \max_{S \subseteq \mathcal{V}, e \in \mathcal{V} \setminus S} \left( f(e|S) \right)$, we can infer that each Hessian approximation $\widehat{\nabla}_t^2 := \frac{1}{L} \sum_{l=1}^{L} \widehat{\nabla}^2 F(\mathbf{x}_l(t))$ in Line 9 of Algorithm 1 satisfies that,*

$$\|\widehat{\nabla}_t^2\|_{2,\infty}^2 \leq n\bar{B}^2 M_f$$

*where $t \in [T]$, $n = |\mathcal{V}|$, $L$ is a positive integer, $\bar{B} = \max_{k=1}^{K} B_k$ is the maximum budget over the $K$ communities $\{\mathcal{V}_1, \ldots, \mathcal{V}_K\}$.*

**Remark 11.** For any matrix $A \in \mathbb{R}^{n \times n}$, the $(2, \infty)$-norm of $A$ is defined as $\|A\|_{2,\infty} = \sup\{\|A\mathbf{x}\|_{\infty} : \mathbf{x} \in \mathbb{R}^n, \|\mathbf{x}\|_2 = 1\}$ where $\| \cdot \|$ denotes the L2 norm.

*Proof.* From the definition of the norm $\| \cdot \|_{2,\infty}$, we can show that

$$\|\widehat{\nabla}_t^2\|_{2,\infty}^2 = \max_{i \in [n]} \|\widehat{\nabla}_t^2(i,:)\|_2^2 \leq n\bar{B}^2 M_f,$$

where $\widehat{\nabla}_t^2(i,:)$ is the $i$-th line of the Hessian approximation $\widehat{\nabla}_t^2$ and the final inequality follows from the Lemma 2. $\qquad \square$

With the Lemma 3, we next verify the gap between our gradient estimator $\mathbf{g}(t)$ and the exact gradient $\nabla F(\mathbf{x}(t))$, that is,

**Lemma 4.** *Given a monotone set function $f : 2^{\mathcal{V}} \to \mathbb{R}_+$, if we denote the maximum marginal value of $f$ as $M_f = \max_{S \subseteq \mathcal{V}, e \in \mathcal{V} \setminus S} \left(f(e|S)\right)$, we can show that each gradient estimator $\mathbf{g}(t)$ in Line 11 of Algorithm 1 satisfies that, for any $t \in [T]$*

$$\mathbb{E}\left(\|\mathbf{g}(t) - \nabla F(\mathbf{x}(t))\|_2^2\right) \leq \frac{nr\bar{B}^2 M_f}{LT},$$

*where $L$ is the batch size, $n = |\mathcal{V}|$, $\bar{B} = \max_{k=1}^K B_k$ is the maximum budget over the $K$ communities $\{\mathcal{V}_1, \ldots, \mathcal{V}_K\}$ and the rank $r = \sum_{k=1}^K B_k$.*

*Proof.* Note that in Lines 4 of Algorithm 1, we compute the exact gradient of our proposed Multinoulli Extension $F$ at the point $\mathbf{0}$ and then assign this value to $\mathbf{g}(1)$. Therefore, we know that when $t = 1$,

$$\|\mathbf{g}(1) - \nabla F(\mathbf{x}(1))\| = \|\mathbf{g}(1) - \nabla F(\mathbf{0})\| = 0 \leq \frac{nr\bar{B}^2 M_f}{LT}.$$

When $t > 1$, we have that

$$\mathbb{E}\left(\|\mathbf{g}(t) - \nabla F(\mathbf{x}(t))\|_2^2\right)$$
$$= \mathbb{E}\left(\|\mathbf{g}(t-1) + \boldsymbol{\xi}_t - \nabla F(\mathbf{x}(t))\|_2^2\right)$$
$$= \mathbb{E}\left(\|\mathbf{g}(t-1) - \nabla F(\mathbf{x}(t-1))\|_2^2\right) + \mathbb{E}\left(\|\boldsymbol{\xi}_t - \left(\nabla F(\mathbf{x}(t)) - \nabla F(\mathbf{x}(t-1))\right)\|_2^2\right)$$
$$+ \mathbb{E}\left(\left\langle \boldsymbol{g}(t-1) - \nabla F(\mathbf{x}(t-1)), \boldsymbol{\xi}_t - \left(\nabla F(\mathbf{x}(t)) - \nabla F(\mathbf{x}(t-1))\right)\right\rangle\right).$$

Note that

$$\mathbb{E}\left(\left\langle \boldsymbol{g}(t-1) - \nabla F(\mathbf{x}(t-1)), \boldsymbol{\xi}_t - \left(\nabla F(\mathbf{x}(t)) - \nabla F(\mathbf{x}(t-1))\right)\right\rangle\right)$$
$$= \mathbb{E}\left(\mathbb{E}\left(\left\langle \boldsymbol{g}(t-1) - \nabla F(\mathbf{x}(t-1)), \boldsymbol{\xi}_t - \left(\nabla F(\mathbf{x}(t)) - \nabla F(\mathbf{x}(t-1))\right)\right\rangle \Big| \mathbf{x}(t)\right)\right)$$
$$= \mathbb{E}\left(\left\langle \boldsymbol{g}(t-1) - \nabla F(\mathbf{x}(t-1)), \mathbb{E}\left(\boldsymbol{\xi}_t | \mathbf{x}(t)\right) - \left(\nabla F(\mathbf{x}(t)) - \nabla F(\mathbf{x}(t-1))\right)\right\rangle\right)$$
$$= 0.$$

Therefore, we have

$$
\mathbb{E}\left(\|\mathbf{g}(t) - \nabla F(\mathbf{x}(t))\|_2^2\right)
$$

$$
= \mathbb{E}\left(\|\mathbf{g}(t-1) - \nabla F(\mathbf{x}(t-1))\|_2^2\right) + \mathbb{E}\left(\|\boldsymbol{\xi}_t - \left(\nabla F(\mathbf{x}(t)) - \nabla F(\mathbf{x}(t-1))\right)\|_2^2\right)
$$

$$
= \mathbb{E}\left(\|\mathbf{g}(t-1) - \nabla F(\mathbf{x}(t-1))\|_2^2\right) + \mathbb{E}\left(\left\|\widehat{\nabla}_t^2(\mathbf{x}(t) - \mathbf{x}(t-1)) - \left(\nabla F(\mathbf{x}(t)) - \nabla F(\mathbf{x}(t-1))\right)\right\|_2^2\right)
$$

$$
= \mathbb{E}\left(\|\mathbf{g}(t-1) - \nabla F(\mathbf{x}(t-1))\|_2^2\right) + \mathbb{E}\left(\left\|\left(\frac{1}{L}\sum_{l=1}^{L}\left(\widehat{\nabla}^2 F(\mathbf{x}_l(t))(\mathbf{x}(t) - \mathbf{x}(t-1))\right)\right) - \left(\nabla F(\mathbf{x}(t)) - \nabla F(\mathbf{x}(t-1))\right)\right\|_2^2\right)
$$

$$
= \mathbb{E}\left(\|\mathbf{g}(t-1) - \nabla F(\mathbf{x}(t-1))\|_2^2\right) + \frac{1}{L}\mathbb{E}\left(\left\|\left(\widehat{\nabla}^2 F(\mathbf{x}_1(t))(\mathbf{x}(t) - \mathbf{x}(t-1))\right) - \left(\nabla F(\mathbf{x}(t)) - \nabla F(\mathbf{x}(t-1))\right)\right\|_2^2\right)
$$

$$
\leq \mathbb{E}\left(\|\mathbf{g}(t-1) - \nabla F(\mathbf{x}(t-1))\|_2^2\right) + \frac{1}{L}\mathbb{E}\left(\|\widehat{\nabla}^2 F(\mathbf{x}_1(t))(\mathbf{x}(t) - \mathbf{x}(t-1))\|_2^2\right)
$$

$$
\leq \mathbb{E}\left(\|\mathbf{g}(t-1) - \nabla F(\mathbf{x}(t-1))\|_2^2\right) + \frac{1}{L}\mathbb{E}\left(\|\widehat{\nabla}^2 F(\mathbf{x}_1(t))\|_{2,\infty}^2 \|\mathbf{x}(t) - \mathbf{x}(t-1)\|_2^2\right)
$$

$$
= \mathbb{E}\left(\|\mathbf{g}(t-1) - \nabla F(\mathbf{x}(t-1))\|_2^2\right) + \frac{1}{L}\mathbb{E}\left(\|\widehat{\nabla}^2 F(\mathbf{x}_1(t))\|_{2,\infty}^2 \|\frac{1}{T}\sum_{k=1}^{K}\frac{1}{B_k}\mathbf{1}_{S(t-1)\cap\mathcal{V}_k}\|_2^2\right)
$$

$$
\leq \mathbb{E}\left(\|\mathbf{g}(t-1) - \nabla F(\mathbf{x}(t-1))\|_2^2\right) + \frac{nr\bar{B}^2 M_f}{LT^2}
$$

$$
\cdots
$$

$$
\leq \mathbb{E}\left(\|\mathbf{g}(1) - \nabla F(\mathbf{x}(1))\|_2^2\right) + \frac{nr\bar{B}^2 M_f}{LT^2}(t-1) \leq \frac{nr\bar{B}^2 M_f}{LT},
$$

where the first inequality follows from $\mathbb{E}\left(X - \mathbb{E}(X)\right)^2 \leq \mathbb{E}(X^2)$ for any random variable $X$; the second inequality comes from the definition of the norm $\|\cdot\|_{2,\infty}$; the third inequality follows from the Lemma 3 and the ascent direction $\sum_{k=1}^{K}\frac{1}{B_k}\mathbf{1}_{S(t-1)\cap\mathcal{V}_k}$ has at most $r$ non-zero elements. $\qquad\square$

Now, we verify the Theorem 4.

*Proof.* From calculus, we know that, there exist a constant $a \in [0,1]$ such that

$$
F(\mathbf{x}(t+1)) - F(\mathbf{x}(t)) - \langle \nabla F(\mathbf{x}(t)), \mathbf{x}(t+1) - \mathbf{x}(t)\rangle = \frac{1}{2}\left\langle \nabla^2 F(\mathbf{x}^a(t))\left(\mathbf{x}(t+1) - \mathbf{x}(t)\right), \mathbf{x}(t+1) - \mathbf{x}(t)\right\rangle,
$$

where $\mathbf{x}^a(t) = a\mathbf{x}(t) + (1-a)\mathbf{x}(t-1)$. Therefore, we can show that

$$
\begin{aligned}
F(\mathbf{x}(t+1)) & \\
&\geq F(\mathbf{x}(t)) + \langle \nabla F(\mathbf{x}(t)), \mathbf{x}(t+1) - \mathbf{x}(t)\rangle - \frac{1}{2}\left|\left\langle \nabla^2 F(\mathbf{x}^a(t))\left(\mathbf{x}(t+1) - \mathbf{x}(t)\right), \mathbf{x}(t+1) - \mathbf{x}(t)\right\rangle\right| \\
&\geq F(\mathbf{x}(t)) + \langle \nabla F(\mathbf{x}(t)), \mathbf{x}(t+1) - \mathbf{x}(t)\rangle - \frac{\|\nabla^2 F(\mathbf{x}^a(t))\|_{2,\infty}}{2}\|\mathbf{x}(t+1) - \mathbf{x}(t)\|_2^2 \\
&\geq F(\mathbf{x}(t)) + \langle \nabla F(\mathbf{x}(t)), \mathbf{x}(t+1) - \mathbf{x}(t)\rangle - \frac{\bar{B}\sqrt{nM_f}}{2}\|\mathbf{x}(t+1) - \mathbf{x}(t)\|_2^2,
\end{aligned}
\tag{22}
$$

where the final inequality follows from Lemma 3.

With Eq.(22) and $\mathbf{x}(t+1) = \mathbf{x}(t) + \frac{1}{T}\mathbf{v}(t)$, we also have that, for any subset $S$ within the partition constraint of problem (1),

$$F(\mathbf{x}(t+1))$$

$$\geq F(\mathbf{x}(t)) + \langle \nabla F(\mathbf{x}(t)), \mathbf{x}(t+1) - \mathbf{x}(t) \rangle - \frac{\bar{B}\sqrt{nM_f}}{2}\|\mathbf{x}(t+1) - \mathbf{x}(t)\|_2^2$$

$$= F(\mathbf{x}(t)) + \frac{1}{T}\left\langle \nabla F(\mathbf{x}(t)), \sum_{k=1}^{K}\frac{1}{B_k}\mathbf{1}_{S(t)\cap\mathcal{V}_k} \right\rangle - \frac{\bar{B}\sqrt{nM_f}}{2T^2}\|\sum_{k=1}^{K}\frac{1}{B_k}\mathbf{1}_{S(t)\cap\mathcal{V}_k}\|_2^2$$

$$= F(\mathbf{x}(t)) + \frac{1}{T}\left\langle \boldsymbol{g}(t), \sum_{k=1}^{K}\frac{\mathbf{1}_{S(t)\cap\mathcal{V}_k}}{B_k} \right\rangle + \frac{1}{T}\left\langle \nabla F(\mathbf{x}(t)) - \boldsymbol{g}(t), \sum_{k=1}^{K}\frac{1}{B_k}\mathbf{1}_{S(t)\cap\mathcal{V}_k} \right\rangle - \frac{\bar{B}\sqrt{nM_f}}{2T^2}\|\sum_{k=1}^{K}\frac{1}{B_k}\mathbf{1}_{S(t)\cap\mathcal{V}_k}\|_2^2$$

$$\geq F(\mathbf{x}(t)) + \frac{1}{T}\left\langle \boldsymbol{g}(t), \sum_{k=1}^{K}\frac{\mathbf{1}_{S\cap\mathcal{V}_k}}{B_k} \right\rangle + \frac{1}{T}\left\langle \nabla F(\mathbf{x}(t)) - \boldsymbol{g}(t), \sum_{k=1}^{K}\frac{1}{B_k}\mathbf{1}_{S(t)\cap\mathcal{V}_k} \right\rangle - \frac{\bar{B}\sqrt{nM_f}}{2T^2}\|\sum_{k=1}^{K}\frac{1}{B_k}\mathbf{1}_{S(t)\cap\mathcal{V}_k}\|_2^2,$$

where the final inequality follows from $\sum_{k=1}^{K}\frac{\mathbf{1}_{S\cap\mathcal{V}_k}}{B_k} \in \prod_{k=1}^{T}\Delta_{n_k}$ if the subset $S$ is included into the partition constraint of problem (1) and Line 13 in Algorithm 1.

As a result, we can show that, in expectation,

$$\mathbb{E}\big(F(\mathbf{x}(t+1))\big)$$

$$\geq \mathbb{E}\big(F(\mathbf{x}(t))\big) + \frac{1}{T}\mathbb{E}\big(\langle \boldsymbol{g}(t), \sum_{k=1}^{K}\frac{\mathbf{1}_{S\cap\mathcal{V}_k}}{B_k} \rangle\big) + \frac{1}{T}\mathbb{E}\big(\langle \nabla F(\mathbf{x}(t)) - \boldsymbol{g}(t), \sum_{k=1}^{K}\frac{1}{B_k}\mathbf{1}_{S(t)\cap\mathcal{V}_k} \rangle\big)$$

$$- \frac{\bar{B}\sqrt{nM_f}}{2T^2}\mathbb{E}\big(\|\sum_{k=1}^{K}\frac{\mathbf{1}_{S(t)\cap\mathcal{V}_k}}{B_k}\|_2^2\big)$$

$$= \mathbb{E}\big(F(\mathbf{x}(t))\big) + \frac{1}{T}\left\langle \mathbb{E}\big(\boldsymbol{g}(t)\big), \sum_{k=1}^{K}\frac{\mathbf{1}_{S\cap\mathcal{V}_k}}{B_k} \right\rangle + \frac{1}{T}\mathbb{E}\big(\langle \nabla F(\mathbf{x}(t)) - \boldsymbol{g}(t), \sum_{k=1}^{K}\frac{1}{B_k}\mathbf{1}_{S(t)\cap\mathcal{V}_k} \rangle\big) \quad (23)$$

$$- \frac{\bar{B}\sqrt{nM_f}}{2T^2}\mathbb{E}\big(\|\sum_{k=1}^{K}\frac{\mathbf{1}_{S(t)\cap\mathcal{V}_k}}{B_k}\|_2^2\big)$$

$$= \mathbb{E}\big(F(\mathbf{x}(t))\big) + \frac{1}{T}\left\langle \nabla F(\mathbf{x}(t)), \sum_{k=1}^{K}\frac{\mathbf{1}_{S\cap\mathcal{V}_k}}{B_k} \right\rangle + \frac{1}{T}\mathbb{E}\big(\langle \nabla F(\mathbf{x}(t)) - \boldsymbol{g}(t), \sum_{k=1}^{K}\frac{1}{B_k}\mathbf{1}_{S(t)\cap\mathcal{V}_k} \rangle\big)$$

$$- \frac{\bar{B}\sqrt{nM_f}}{2T^2}\mathbb{E}\big(\|\sum_{k=1}^{K}\frac{\mathbf{1}_{S(t)\cap\mathcal{V}_k}}{B_k}\|_2^2\big),$$

where the final equality follows from $\boldsymbol{g}(\mathbf{x}(t))$ is the unbiased estimator of $\nabla F(\mathbf{x}(t))$ for any $t \in [T]$.

**1):** When the set function $f : 2^{\mathcal{V}} \to \mathbb{R}_+$ is monotone and $\alpha$-weakly DR-submodular, from Theorem 2, we can show that

$$\mathbb{E}\big(F(\mathbf{x}(t+1))\big)$$

$$= \mathbb{E}\big(F(\mathbf{x}(t))\big) + \frac{1}{T}\left\langle \nabla F(\mathbf{x}(t)), \sum_{k=1}^{K}\frac{\mathbf{1}_{S\cap\mathcal{V}_k}}{B_k} \right\rangle + \frac{1}{T}\mathbb{E}\big(\langle \nabla F(\mathbf{x}(t)) - \boldsymbol{g}(t), \sum_{k=1}^{K}\frac{\mathbf{1}_{S(t)\cap\mathcal{V}_k}}{B_k} \rangle\big) - \frac{\bar{B}\sqrt{nM_f}}{2T^2}\mathbb{E}\big(\|\sum_{k=1}^{K}\frac{\mathbf{1}_{S(t)\cap\mathcal{V}_k}}{B_k}\|_2^2\big)$$

$$\geq \mathbb{E}\big(F(\mathbf{x}(t))\big) + \frac{\alpha}{T}\big(f(S) - \mathbb{E}\big(F(\mathbf{x}(t))\big)\big) + + \frac{1}{T}\mathbb{E}\big(\langle \nabla F(\mathbf{x}(t)) - \boldsymbol{g}(t), \sum_{k=1}^{K}\frac{\mathbf{1}_{S(t)\cap\mathcal{V}_k}}{B_k} \rangle\big) - \frac{\bar{B}\sqrt{nM_f}}{2T^2}\mathbb{E}\big(\|\sum_{k=1}^{K}\frac{\mathbf{1}_{S(t)\cap\mathcal{V}_k}}{B_k}\|_2^2$$

$$\geq \mathbb{E}\big(F(\mathbf{x}(t))\big) + \frac{\alpha}{T}\big(f(S) - \mathbb{E}\big(F(\mathbf{x}(t))\big)\big) - \frac{1}{2\bar{B}\sqrt{nM_f}}\mathbb{E}\big(\|F(\mathbf{x}(t)) - \boldsymbol{g}(t)\|_2^2\big) - \frac{\bar{B}\sqrt{nM_f}}{T^2}\mathbb{E}\big(\|\sum_{k=1}^{K}\frac{1}{B_k}\mathbf{1}_{S(t)\cap\mathcal{V}_k}\|_2^2\big)\big)$$

$$\geq \mathbb{E}\big(F(\mathbf{x}(t))\big) + \frac{\alpha}{T}\big(f(S) - \mathbb{E}\big(F(\mathbf{x}(t))\big)\big) - \frac{r\bar{B}\sqrt{nM_f}}{2LT} - \frac{r\bar{B}\sqrt{nM_f}}{T^2}$$

$$(24)$$

where the second inequality follows from the Young's inequality.

By rearranging the Eq.(24), we can show that

$$\Big( f(S) - \mathbb{E}\big(F(\mathbf{x}(t+1))\big) \Big)$$

$$\leq (1-\tfrac{\alpha}{T})\Big(f(S) - \mathbb{E}\big(F(\mathbf{x}(t))\big)\Big) + \frac{r\bar{B}\sqrt{nM_f}}{2LT} + \frac{r\bar{B}\sqrt{nM_f}}{T^2}$$

$$\leq \ldots$$

$$\leq (1-\tfrac{\alpha}{T})^t\Big(f(S) - \mathbb{E}\big(F(\mathbf{x}(1))\big)\Big) + \Big(\frac{r\bar{B}\sqrt{nM_f}}{2LT} + \frac{r\bar{B}\sqrt{nM_f}}{T^2}\Big)\sum_{i=0}^{t-1}(1-\tfrac{\alpha}{T})^i$$

$$\leq (1-\tfrac{\alpha}{T})^t\Big(f(S) - \mathbb{E}\big(F(\mathbf{x}(1))\big)\Big) + \Big(\frac{r\bar{B}\sqrt{nM_f}}{2LT} + \frac{r\bar{B}\sqrt{nM_f}}{T^2}\Big)\frac{T}{\alpha}$$

$$\leq (1-\tfrac{\alpha}{T})^t f(S) + \frac{r\bar{B}\sqrt{nM_f}}{2\alpha L} + \frac{r\bar{B}\sqrt{nM_f}}{\alpha T}.$$

Finally, we have that

$$\mathbb{E}\big(F(\mathbf{x}(T+1))\big)$$

$$\geq \Big(1 - (1-\tfrac{\alpha}{T})^T\Big)f(S) - \frac{r\bar{B}\sqrt{nM_f}}{2\alpha L} - \frac{r\bar{B}\sqrt{nM_f}}{\alpha T}$$

$$\geq \Big(1 - e^{-\alpha}\Big)f(S) - \frac{r\bar{B}\sqrt{nM_f}}{2\alpha L} - \frac{r\bar{B}\sqrt{nM_f}}{\alpha T},$$

where the final inequality follows from $(1-\tfrac{\alpha}{T})^T \leq e^{-\alpha}$ when $T \geq 3$.

Therefore, when $L = \tfrac{T}{2}$, we have that

$$\mathbb{E}\big(F(\mathbf{x}(T+1))\big) \geq \Big(1 - e^{-\alpha}\Big)f(S^*) - \frac{2r\bar{B}\sqrt{nM_f}}{\alpha T}.$$

**2:** When the set function $f : 2^{\mathcal{V}} \to \mathbb{R}_+$ is monotone and $(\gamma, \beta)$-weakly submodular, from the Theorem 2, we can show that

$$\mathbb{E}\big(F(\mathbf{x}(t+1))\big) \geq \mathbb{E}\big(F(\mathbf{x}(t))\big) + \frac{\gamma^2}{T}f(S) - \frac{\beta(1-\gamma)+\gamma^2}{T}\mathbb{E}\big(F(\mathbf{x}(t))\big) - \frac{r\bar{B}\sqrt{nM_f}}{2LT} - \frac{r\bar{B}\sqrt{nM_f}}{T^2}. \tag{25}$$

By rearranging the Eq.(25), we can have that

$$\Big(\gamma^2 f(S) - (\beta(1-\gamma)+\gamma^2)\mathbb{E}\big(F(\mathbf{x}(t+1))\big)\Big)$$

$$(1-\frac{\beta(1-\gamma)+\gamma^2}{T})\Big(\gamma^2 f(S) - (\beta(1-\gamma)+\gamma^2)\mathbb{E}\big(F(\mathbf{x}(t))\big)\Big) + \Big(\frac{r\bar{B}\sqrt{nM_f}}{2LT} + \frac{r\bar{B}\sqrt{nM_f}}{T^2}\Big)(\beta(1-\gamma)+\gamma^2)$$

$$\leq \ldots$$

$$\leq (1-\frac{\beta(1-\gamma)+\gamma^2}{T})^t\Big(f\gamma^2 f(S) - (\beta(1-\gamma)+\gamma^2)\mathbb{E}\big(F(\mathbf{x}(0))\big)\Big) + \Big(\frac{r\bar{B}\sqrt{nM_f}}{2LT} + \frac{r\bar{B}\sqrt{nM_f}}{T^2}\Big)T(\beta(1-\gamma)+\gamma^2)$$

$$\leq (1-\frac{\beta(1-\gamma)+\gamma^2}{T})^t\gamma^2 f(S) + \frac{r\bar{B}\sqrt{nM_f}}{2L} + \frac{r\bar{B}\sqrt{nM_f}}{T}.$$

Finally, we have that

$$(\beta(1-\gamma)+\gamma^2)\mathbb{E}\big(F(\mathbf{x}(T+1))\big)$$

$$\geq \Big(1 - (1-\frac{\beta(1-\gamma)+\gamma^2}{T})^T\Big)\gamma^2 f(S) + \frac{r\bar{B}\sqrt{nM_f}}{2L} + \frac{r\bar{B}\sqrt{nM_f}}{T}$$

$$\geq \gamma^2(1 - e^{-(\beta(1-\gamma)+\gamma^2)})f(S) + \frac{r\bar{B}\sqrt{nM_f}}{2L} + \frac{r\bar{B}\sqrt{nM_f}}{T},$$

where the final inequality follows from $(1-\frac{\beta(1-\gamma)+\gamma^2}{T})^T \leq e^{-(\beta(1-\gamma)+\gamma^2)}$ when $T \geq 3$.

---

**Algorithm 2** Rounding Without Replacement

---

**Input:** Point $(\mathbf{p}_1, \ldots, \mathbf{p}_K) \in \prod_{k=1}^{K} \Delta_{n_k}$ where $\mathbf{p}_k = (p_k^1, \ldots, p_k^{n_k})$ and $\sum_{m=1}^{n_k} p_k^m = 1$ for any $k \in [K]$, Partition $(\mathcal{V}_1, \ldots, \mathcal{V}_K)$ of set $\mathcal{V}$ where $\mathcal{V}_k := \{v_k^1, \ldots, v_k^{n_k}\}$ for any $k \in [K]$, Budget set $\{B_1, \ldots, B_K\}$

1: **Initialize** $S = \emptyset$;
2: **for** $\hat{k} = 1, \ldots, K$ **do**
3:     $P = 0$
4:     **for** $\hat{b} = 1, \ldots, B_k$ **do**
5:         **if** $\hat{b} = 1$ **then**
6:             Sampling a number $N_1$ from $[n_k]$ according to the probability $\Pr(N_1 = m) = p_{\hat{k}}^m$ for any $m \in [n_{\hat{k}}]$;
7:         **else**
8:             Sampling a number $N_{\hat{b}}$ from $[n_k] - \{N_1, \ldots, N_{\hat{b}-1}\}$ according to the probability $\Pr(N_{\hat{b}} = m) = \frac{p_{\hat{k}}^m}{1-P}$ for any
            $m \in \left( [n_{\hat{k}}] - \{N_1, \ldots, N_{\hat{b}-1}\} \right)$;
9:         **end if**
10:         Set $S = S \cup \{v_{\hat{k}}^{N_{\hat{b}}}\}$ and $P = P + p_{\hat{k}}^{N_{\hat{b}}}$;
11:     **end for**
12: **end for**
13: Return $S$;

---

Therefore, when $L = \frac{T}{2}$, we have that

$$\mathbb{E}\big(F(\mathbf{x}(T+1))\big) \geq \Big(\frac{\gamma^2(1 - e^{-(\gamma(1-\beta)+\gamma^2)})}{\gamma(1-\beta)+\gamma^2}\Big) f(S) - \frac{2r\bar{B}\sqrt{nM_f}}{T(\gamma(1-\beta)+\gamma^2)}.$$

$\square$

**Remark 12.** If $T = L = \mathcal{O}(\frac{r\sqrt{n}}{\epsilon})$ and $S$ is the optimal subset of problem (1), we can show that, when the objective function is monotone $\alpha$-weakly DR-submodular or $(\gamma, \beta)$-weakly submodular, our **Multinoulli-SCG** algorithm can attain a value of $(1 - e^{-\alpha})f(S) - \epsilon$ or $(\frac{\gamma^2(1-e^{-(\beta(1-\gamma)+\gamma^2)})}{\beta(1-\gamma)+\gamma^2})f(S) - \epsilon$. Note that, during the process of $\mathcal{O}(\frac{r\sqrt{n}}{\epsilon})$ iterations, if $L = \mathcal{O}(\frac{r\sqrt{n}}{\epsilon})$, due to Remark 8, **Multinoulli-SCG** only requires evaluating the set function $\mathcal{O}(\frac{r^3 n^2}{\epsilon^2})$ times.

### D.2. Rounding Without Replacement

In this section, we aim to present a more effective rounding method for our proposed Multinoulli Extension $F$ when its original set function $f$ is monotone.

Given the second point of Theorem 1, we know that when the set function $f$ exhibits monotonicity, its Multinoulli Extension $F$ is also monotone. Therefore, it can be deduced that the optimal value of the relaxed problem (4) must be attained at the boundary of $\prod_{k=1}^{K} \Delta_{n_k}$. Therefore, this section primarily concentrates on how to design more effective rounding method for the points at the boundary of $\prod_{k=1}^{K} \Delta_{n_k}$. The specific details are presented in Algorithm 2.

The core of our our proposed Algorithm 2 lies in the Line 8, that is, instead of independently sampling according to each probability vector $\mathbf{p}_{\hat{k}}$, we take into account the elements previously selected within the same community, namely, $\{v_{\hat{k}}^{N_1}, \ldots, v_{\hat{k}}^{N_{\hat{b}-1}}\}$. As a result, we can prove that

**Theorem 6.** *For any point $(\mathbf{p}_1, \ldots, \mathbf{p}_K)$ at the boundary of the domain $\prod_{k=1}^{K} \Delta_{n_k}$, i.e., $\|\mathbf{p}_k\|_1 = 1$ for any $k \in [K]$, if the set function $f$ is monotone, we can show that the subset $S$ returned by Algorithm 2 satisfies:*

- *for any $k \in [K]$, $|S \cap \mathcal{V}_k| = B_k$;*

- $E\big(f(S)\big) \geq F(\mathbf{p}_1, \ldots, \mathbf{p}_K)$ *where $F$ is the Multinoulli Extension of $f$.*

*Proof.* The first point of Theorem 6 is easy to verify. We mainly focus on the second point.

At first, fixing a $\hat{k} \in [K]$ and a $\hat{b} \in [n_{\hat{k}}]$, we firstly prove that, when $S = \{v_k^{N_b} : (k, b) < (\hat{k}, \hat{b})\}$, the following inequality holds:

$$\mathbb{E}_{v_{\hat{k}}^{N_{\hat{b}}}} \left( f\big(S \cup \{v_{\hat{k}}^{N_{\hat{b}}}\}\big) \Big| S \right) \geq \mathbb{E}_{e \sim \text{Multi}(\mathbf{p}_{\hat{k}})} \left( f\big(S \cup \{e\}\big) \Big| S \right). \tag{26}$$

In order to verify Eq.(26), we first show that

$$
\begin{aligned}
\mathbb{E}_{v_{\hat{k}}^{N_{\hat{b}}}} &\left( f\big(S \cup \{v_{\hat{k}}^{N_{\hat{b}}}\}\big) \right) \\
&= \sum_{m \in [n_{\hat{k}}] - \{N_1, \ldots, N_{\hat{b}-1}\}} \left( \frac{p_{\hat{k}}^m}{1-P} f\big(S \cup \{v_{\hat{k}}^m\}\big) \right) \\
&= \frac{P}{1-P} \sum_{m \in [n_{\hat{k}}] - \{N_1, \ldots, N_{\hat{b}-1}\}} \left( p_{\hat{k}}^m f\big(S \cup \{v_{\hat{k}}^m\}\big) \right) + \sum_{m \in [n_{\hat{k}}] - \{N_1, \ldots, N_{\hat{b}-1}\}} \left( p_{\hat{k}}^m f\big(S \cup \{v_{\hat{k}}^m\}\big) \right) \\
&\geq \frac{P}{1-P} \sum_{m \in [n_{\hat{k}}] - \{N_1, \ldots, N_{\hat{b}-1}\}} \left( p_{\hat{k}}^m f(S) \right) + \sum_{m \in [n_{\hat{k}}] - \{N_1, \ldots, N_{\hat{b}-1}\}} \left( p_{\hat{k}}^m f\big(S \cup \{v_{\hat{k}}^m\}\big) \right) \\
&= P f(S) + \sum_{m \in [n_{\hat{k}}] - \{N_1, \ldots, N_{\hat{b}-1}\}} \left( p_{\hat{k}}^m f\big(S \cup \{v_{\hat{k}}^m\}\big) \right),
\end{aligned} \tag{27}
$$

where the first inequality from the monotonicity of $f$. Then, we also have that

$$
\begin{aligned}
\mathbb{E}_{e \sim \text{Multi}(\mathbf{p}_{\hat{k}})} &\left( f\big(S \cup \{e\}\big) \Big| S \right) \\
&= \sum_{m \in [n_{\hat{k}}]} \left( p_{\hat{k}}^m f\big(S \cup \{v_{\hat{k}}^m\}\big) \right) \\
&= \sum_{m \in \{N_1, \ldots, N_{\hat{b}-1}\}} \left( p_{\hat{k}}^m f(S) \right) + \sum_{m \in [n_{\hat{k}}] - \{N_1, \ldots, N_{\hat{b}-1}\}} \left( p_{\hat{k}}^m f\big(S \cup \{v_{\hat{k}}^m\}\big) \right) \\
&= P f(S) + \sum_{m \in [n_{\hat{k}}] - \{N_1, \ldots, N_{\hat{b}-1}\}} \left( p_{\hat{k}}^m f\big(S \cup \{v_{\hat{k}}^m\}\big) \right).
\end{aligned} \tag{28}
$$

With Eq.(27) and Eq.(28), we get the result of Eq.(26). Therefore, if we start by the final subset $S$ and recurrently apply the Eq.(26), we can get $E\big(f(S)\big) \geq F(\mathbf{p}_1, \ldots, \mathbf{p}_K)$. $\qquad \square$

## E. Stationary-Point Strategy for Mutinoulli Extension

### E.1. Proof of Theorem 5

In this section, we prove the Theorem 5. Firstly, we prove a lower bound about $\left\langle (\mathbf{p}_1, \ldots, \mathbf{p}_K), \nabla F(\mathbf{p}_1, \ldots, \mathbf{p}_K) \right\rangle$ for any point $(\mathbf{p}_1, \ldots, \mathbf{p}_K) \in \prod_{k=1}^{K} \Delta_{n_k}$, that is to say,

**Theorem 7.** *When the set function $f : 2^{\mathcal{V}} \to \mathbb{R}_+$ is $\alpha$-weakly DR-submodular, for any point $(\mathbf{p}_1, \ldots, \mathbf{p}_K) \in \prod_{k=1}^{K} \Delta_{n_k}$, the following inequality holds:*

$$\left\langle (\mathbf{p}_1, \ldots, \mathbf{p}_K), \nabla F(\mathbf{p}_1, \ldots, \mathbf{p}_K) \right\rangle \geq \frac{1}{\alpha} F(\mathbf{p}_1, \ldots, \mathbf{p}_K), \tag{29}$$

*where $\alpha \in (0, 1]$. Similarly, when the set function $f : 2^{\mathcal{V}} \to \mathbb{R}_+$ is $\beta$-weakly submodular from above, for any point $(\mathbf{p}_1, \ldots, \mathbf{p}_K) \in \prod_{k=1}^{K} \Delta_{n_k}$, we also can infer that*

$$\left\langle (\mathbf{p}_1, \ldots, \mathbf{p}_K), \nabla F(\mathbf{p}_1, \ldots, \mathbf{p}_K) \right\rangle \geq \beta F(\mathbf{p}_1, \ldots, \mathbf{p}_K),$$

*where $\beta \geq 1$.*

*Proof.* At first, we assume each $\mathbf{p}_k := (p_k^1, \ldots, p_k^{n_k})$ for any $k \in [K]$. Then, from the first point of Theorem 1 and the fifth equality in Eq.(13), we have that

$$
\frac{\partial F}{\partial p_k^m}(\mathbf{p}_1, \ldots, \mathbf{p}_K) = B_k \left( \mathbb{E}_{e_{\hat{k}}^{\hat{b}} \sim Multi(\mathbf{p}_{\hat{k}})} \left( f \left( v_k^m \,\middle|\, \cup_{(\hat{k}, \hat{b}) \neq (k,1)} \{e_{\hat{k}}^{\hat{b}}\} \right) \right) \right)
$$

$$
= B_k \left( \sum_{e_{\hat{k}}^{\hat{b}} \in \mathcal{V}_{\hat{k}} \cup \{\emptyset\}, (\hat{k}, \hat{b}) \neq (k,1)} \left( f \left( v_k^m | \cup_{(\hat{k}, \hat{b}) \neq (k,1)} \{e_{\hat{k}}^{\hat{b}}\} \right) \prod_{(\hat{k}, \hat{b}) \neq (k,1)} \Pr(e_{\hat{k}}^{\hat{b}} | \mathbf{p}_{\hat{k}}) \right) \right)
$$

As a result, we have that

$$
\left\langle (\mathbf{p}_1, \ldots, \mathbf{p}_K), \nabla F(\mathbf{p}_1, \ldots, \mathbf{p}_K) \right\rangle
$$

$$
= \sum_{k=1}^{K} \sum_{m=1}^{n_k} p_k^m \frac{\partial F}{\partial p_k^m}(\mathbf{p}_1, \ldots, \mathbf{p}_K)
$$

$$
= \sum_{k=1}^{K} \sum_{m=1}^{n_k} p_k^m B_k \left( \mathbb{E}_{e_{\hat{k}}^{\hat{b}} \sim Multi(\mathbf{p}_{\hat{k}})} \left( f \left( v_k^m \,\middle|\, \cup_{(\hat{k}, \hat{b}) \neq (k,1)} \{e_{\hat{k}}^{\hat{b}}\} \right) \right) \right)
$$

$$
= \sum_{k=1}^{K} \sum_{m=1}^{n_k} p_k^m B_k \left( \sum_{e_{\hat{k}}^{\hat{b}} \in \mathcal{V}_{\hat{k}} \cup \{\emptyset\}, (\hat{k}, \hat{b}) \neq (k,1)} \left( f \left( v_k^m | \cup_{(\hat{k}, \hat{b}) \neq (k,1)} \{e_{\hat{k}}^{\hat{b}}\} \right) \prod_{(\hat{k}, \hat{b}) \neq (k,1)} \Pr(e_{\hat{k}}^{\hat{b}} | \mathbf{p}_{\hat{k}}) \right) \right)
$$

$$
= \sum_{k=1}^{K} \sum_{m=1}^{n_k} \Pr(v_k^m | \mathbf{p}_k) B_k \left( \sum_{e_{\hat{k}}^{\hat{b}} \in \mathcal{V}_{\hat{k}} \cup \{\emptyset\}, (\hat{k}, \hat{b}) \neq (k,1)} \left( f \left( v_k^m | \cup_{(\hat{k}, \hat{b}) \neq (k,1)} \{e_{\hat{k}}^{\hat{b}}\} \right) \prod_{(\hat{k}, \hat{b}) \neq (k,1)} \Pr(e_{\hat{k}}^{\hat{b}} | \mathbf{p}_{\hat{k}}) \right) \right)
$$

$$
= \sum_{k=1}^{K} B_k \sum_{m=1}^{n_k} \Pr(v_k^m | \mathbf{p}_k) \left( \sum_{e_{\hat{k}}^{\hat{b}} \in \mathcal{V}_{\hat{k}} \cup \{\emptyset\}, (\hat{k}, \hat{b}) \neq (k,1)} \left( f \left( v_k^m | \cup_{(\hat{k}, \hat{b}) \neq (k,1)} \{e_{\hat{k}}^{\hat{b}}\} \right) \prod_{(\hat{k}, \hat{b}) \neq (k,1)} \Pr(e_{\hat{k}}^{\hat{b}} | \mathbf{p}_{\hat{k}}) \right) \right)
$$

$$
= \sum_{k=1}^{K} B_k \left( \sum_{m=1}^{n_k} \sum_{e_{\hat{k}}^{\hat{b}} \in \mathcal{V}_{\hat{k}} \cup \{\emptyset\}, (\hat{k}, \hat{b}) \neq (k,1)} \left( f \left( v_k^m | \cup_{(\hat{k}, \hat{b}) \neq (k,1)} \{e_{\hat{k}}^{\hat{b}}\} \right) \Pr(v_k^m | \mathbf{p}_k) \prod_{(\hat{k}, \hat{b}) \neq (k,1)} \Pr(e_{\hat{k}}^{\hat{b}} | \mathbf{p}_{\hat{k}}) \right) \right)
$$

$$
= \sum_{k=1}^{K} B_k \left( \sum_{e_{\hat{k}}^{\hat{b}} \in \mathcal{V}_{\hat{k}} \cup \{\emptyset\}, \forall \hat{b} \in [B_{\hat{k}}], \forall \hat{k} \in [K]} \left( f \left( e_k^1 | \cup_{(\hat{k}, \hat{b}) \neq (k,1)} \{e_{\hat{k}}^{\hat{b}}\} \right) \prod_{\hat{k}=1}^{K} \prod_{\hat{b}=1}^{B_{\hat{k}}} \Pr(e_{\hat{k}}^{\hat{b}} | \mathbf{p}_{\hat{k}}) \right) \right)
$$

$$
= \sum_{k=1}^{K} B_k \mathbb{E}_{e_{\hat{k}}^{\hat{b}} \sim Multi(\mathbf{p}_{\hat{k}})} \left( f \left( e_k^1 | \cup_{(\hat{k}, \hat{b}) \neq (k,1)} \{e_{\hat{k}}^{\hat{b}}\} \right) \right)
$$

$$
= \sum_{k=1}^{K} \sum_{b=1}^{n_k} \mathbb{E}_{e_{\hat{k}}^{\hat{b}} \sim Multi(\mathbf{p}_{\hat{k}})} \left( f \left( e_k^b | \cup_{(\hat{k}, \hat{b}) \neq (k,b)} \{e_{\hat{k}}^{\hat{b}}\} \right) \right),
$$

where the fourth equality follows from $\Pr(v_k^m | \mathbf{p}_k) = p_k^m$; the seventh equality comes from $e_k^1 \sim Multi(\mathbf{p}_k)$ and $f(\emptyset | B) = 0$ for any $B \subseteq \mathcal{V}$ as well as the final equality follows from that each $e_k^b$ is independently drawn from the multinoulli distribution $Multi(\mathbf{p}_k)$.

If $f$ is $\beta$-weakly submodular from above, we can show that

$$
\sum_{k=1}^{K} \sum_{b=1}^{n_k} f \left( e_k^b | \cup_{(\hat{k}, \hat{b}) \neq (k,b)} \{e_{\hat{k}}^{\hat{b}}\} \right) \leq \beta \left( f \left( \cup_{\hat{k}=1}^{K} \cup_{\hat{b}=1}^{B_{\hat{k}}} \right) - f(\emptyset) \right) \leq \beta f \left( \cup_{\hat{k}=1}^{K} \cup_{\hat{b}=1}^{B_{\hat{k}}} \right),
$$

where the final inequality follows from $f(\emptyset) \geq 0$ (Note that we define $f : 2^{\mathcal{V}} \to \mathbb{R}_+$).

As a result, when $f$ is $\beta$-weakly submodular from above, we have that

$$
\Big\langle (\mathbf{p}_1, \ldots, \mathbf{p}_K), \nabla F(\mathbf{p}_1, \ldots, \mathbf{p}_K) \Big\rangle
$$

$$
= \sum_{k=1}^{K} \sum_{b=1}^{n_k} \mathbb{E}_{e_{\hat{k}}^{\hat{b}} \sim Multi(\mathbf{p}_{\hat{k}})} \Bigg( f\Big( e_k^b | \cup_{(\hat{k},\hat{b}) \neq (k,b)} \{e_{\hat{k}}^{\hat{b}}\} \Big) \Bigg)
$$

$$
\leq \beta \mathbb{E}_{e_{\hat{k}}^{\hat{b}} \sim Multi(\mathbf{p}_{\hat{k}})} \Bigg( f\Big( \cup_{\hat{k}=1}^{K} \cup_{\hat{b}=1}^{B_{\hat{k}}} \Big) \Bigg) = \beta F(\mathbf{p}_1, \ldots, \mathbf{p}_K).
$$

Note that an $\alpha$-weakly DR-submodular function automatically satisfies the conditions for being $\frac{1}{\alpha}$-weakly submodular from above. Thus, we get the Eq.(29). $\qquad\square$

Merging Theorem 7 into Theorem 2, we also can get that

**Theorem 8.** *When the set function $f : 2^{\mathcal{V}} \to \mathbb{R}_+$ is monotone and $\alpha$-weakly DR-submodular, for any subset $S$ within the partition constraint of problem (1) and any point $(\mathbf{p}_1, \ldots, \mathbf{p}_K) \in \prod_{k=1}^{K} \Delta_{n_k}$, the following inequality holds:*

$$
\Bigg\langle \sum_{k=1}^{K} \frac{1}{B_k} \mathbf{1}_{S \cap \mathcal{V}_k} - \Big( \mathbf{p}_1, \ldots, \mathbf{p}_K \Big), \nabla F\Big( \mathbf{p}_1, \ldots, \mathbf{p}_K \Big) \Bigg\rangle \geq \alpha f(S) - (\alpha + \frac{1}{\alpha}) F\Big( \mathbf{p}_1, \ldots, \mathbf{p}_K \Big).
$$

*Similarly, when the set function $f : 2^{\mathcal{V}} \to \mathbb{R}_+$ is monotone and $(\gamma, \beta)$-weakly submodular, for any subset $S$ within the partition constraint of problem (1) and any point $(\mathbf{p}_1, \ldots, \mathbf{p}_K) \in \prod_{k=1}^{K} \Delta_{n_k}$, we also can infer that*

$$
\Bigg\langle \sum_{k=1}^{K} \frac{1}{B_k} \mathbf{1}_{S \cap \mathcal{V}_k} - \Big( \mathbf{p}_1, \ldots, \mathbf{p}_K \Big), \nabla F\Big( \mathbf{p}_1, \ldots, \mathbf{p}_K \Big) \Bigg\rangle \geq \gamma^2 f(S) - (\beta + \beta(1-\gamma) + \gamma^2) F\Big( \mathbf{p}_1, \ldots, \mathbf{p}_K \Big).
$$

From the definition of stationary point, we know that if $(\mathbf{p}_1, \ldots, \mathbf{p}_K) \in \prod_{k=1}^{K} \Delta_{n_k}$ is the stationary point over the domain $\prod_{k=1}^{K} \Delta_{n_k}$, for any point $\mathbf{y} \in \prod_{k=1}^{K} \Delta_{n_k}$,

$$
\Big\langle \mathbf{y} - \Big( \mathbf{p}_1, \ldots, \mathbf{p}_K \Big), \nabla F\Big( \mathbf{p}_1, \ldots, \mathbf{p}_K \Big) \Big\rangle \leq 0. \tag{30}
$$

Also, for any $S$ within the partition constraint of problem (1), we can easily show that $\sum_{k=1}^{K} \frac{1}{B_k} \mathbf{1}_{S \cap \mathcal{V}_k} \in \prod_{k=1}^{K} \Delta_{n_k}$ such that we know that, for any $S$ within the partition constraint of problem (1),

$$
\Bigg\langle \sum_{k=1}^{K} \frac{1}{B_k} \mathbf{1}_{S \cap \mathcal{V}_k} - \Big( \mathbf{p}_1, \ldots, \mathbf{p}_K \Big), \nabla F\Big( \mathbf{p}_1, \ldots, \mathbf{p}_K \Big) \Bigg\rangle \leq 0.
$$

Therefore, when the set function $f : 2^{\mathcal{V}} \to \mathbb{R}_+$ is monotone and $\alpha$-weakly DR-submodular, we have that $\alpha f(S) - (\alpha + \frac{1}{\alpha}) F\Big( \mathbf{p}_1, \ldots, \mathbf{p}_K \Big) \leq 0$ such that $F\Big( \mathbf{p}_1, \ldots, \mathbf{p}_K \Big) \geq \frac{\alpha^2}{1+\alpha^2} f(S^*)$ where $S^*$ is the optimal solution of problem (1). Similarly, when the set function $f : 2^{\mathcal{V}} \to \mathbb{R}_+$ is monotone and $(\gamma, \beta)$-weakly submodular,, we have that $\gamma^2 f(S) - (\beta + \beta(1-\gamma) + \gamma^2) F\Big( \mathbf{p}_1, \ldots, \mathbf{p}_K \Big) \leq 0$ such that $F\Big( \mathbf{p}_1, \ldots, \mathbf{p}_K \Big) \geq \Big( \frac{\gamma^2}{\beta + \beta(1-\gamma) + \gamma^2} \Big) f(S^*)$.

### E.2. Stochastic Gradient Ascent for Multinoulli Extension

In general, a simple strategy is to initially apply the well-established Gradient Ascent (GA) method to maximize our proposed Multinoulli Extension $F$, and subsequently, to finalize our selection by rounding the resulting continuous solution. However, the implementation of GA often requires accurately computing the gradients of $F$, which is typically computationally intensive. Fortunately, the first point of Theorem 1 indicates that it is feasible to sample a sequence of random elements to construct an unbiased estimator for each $\frac{\partial F}{\partial p_k^m}(\mathbf{p}_1, \ldots, \mathbf{p}_K)$ where $(\mathbf{p}_1, \ldots, \mathbf{p}_K) \in \prod_{k=1}^{K} \Delta_{n_k}$. Specifically, when each

---

**Algorithm 3** Stochastic Gradient Ascent for Mutinoulli Extension(**Multinoulli-SGA**)

---

**Input:** Batch size $L$, step size $\eta$, number of iterations $T$, partition $\{\mathcal{V}_1, \ldots, \mathcal{V}_K\}$ of set $\mathcal{V}$ where $\mathcal{V}_k = \{v_k^1, \ldots, v_k^{n_k}\}$ and the set function $f$

1: **Initialize** $\mathbf{x}(1) = (\mathbf{p}_1(1), \ldots, \mathbf{p}_K(1)) \in \prod_{k=1}^K \triangle_{n_k}$;
2: **for** $t = 1, \ldots, T$ **do**
3:     Generate a subset $S_t$ by rounding $\mathbf{x}(t)$;
4:     Compute the estimator $\widehat{\nabla} F(\mathbf{x}(t))$ based on Eq.(31);
5:     Set $\mathbf{y}(t+1) := \mathbf{x}(t) + \eta \widehat{\nabla} F(\mathbf{x}(t))$;
6:     $\mathbf{x}(t+1) := \arg\min_{\mathbf{z} \in \prod_{k=1}^{n_k} \triangle_{n_k}} \|\mathbf{z} - \mathbf{y}(t+1)\|_2$;
7: **end for**
8: Return $S := \arg\max_{t \in [T]} f(S_t)$;

---

$e_{\hat{k}}^{\hat{b}}(l)$ is independently drawn from the multinoulli distribution Multi($\mathbf{p}_{\hat{k}}$) for any $\hat{k} \in [K]$, $\hat{b} \in [n_{\hat{k}}]$ and $l \in [L]$, we can estimate $\frac{\partial F}{\partial p_k^m}(\mathbf{p}_1, \ldots, \mathbf{p}_K)$ as:

$$\widehat{\frac{\partial F}{\partial p_k^m}}(\mathbf{p}_1, \ldots, \mathbf{p}_K) := \frac{B_k}{L} \sum_{l=1}^L \left( f\left( v_k^m \,\middle|\, \cup_{(\hat{k},\hat{b}) \neq (k,1)} \{e_{\hat{k}}^{\hat{b}}(l)\} \right) \right). \tag{31}$$

By merging this stochastic gradient Eq.(31) into the standard GA method, we can derive a stochastic variant of gradient ascent method for our proposed ME, as detailed in Algorithm 3. Furthermore, based on the previous results of Theorem 1 and Theorem 2, we also can verify that:

**Theorem 9.** *When the set function $f : 2^{\mathcal{V}} \to \mathbb{R}_+$ is monotone and $\alpha$-weakly DR-submodular, if we set the batch size $L = \mathcal{O}(1)$, the subset $S$ output by Algorithm 3 satisfies:*

$$\mathbb{E}(f(S)) \geq \left(\frac{\alpha^2}{1+\alpha^2}\right) f(S^*) - \frac{K}{\eta T} - \eta \frac{n \bar{B}^2 M_f^2}{4},$$

*where $S^*$ is the optimal solution of problem (1), $\bar{B} = \max_{k=1}^K B_k$ is the maximum budget over the $K$ communities $\{\mathcal{V}_1, \ldots, \mathcal{V}_K\}$ and $M_f = \max_{S \subseteq \mathcal{V}, e \in \mathcal{V} \setminus S} \left( f(e|S) \right)$. Similarly, if the set function $f : 2^{\mathcal{V}} \to \mathbb{R}_+$ is $(\gamma, \beta)$-weakly monotone submodular and $L = \mathcal{O}(1)$, the returned subset $S$ of Algorithm 3 satisfies:*

$$\mathbb{E}(f(S)) \geq \left(\frac{\gamma^2}{\beta + \beta(1-\gamma) + \gamma^2}\right) f(S^*) - \frac{8K}{7\eta T} - \eta \frac{2n \bar{B}^2 M_f^2}{7}.$$

**Remark 13.** Theorem 5 shows that, when the set function $f$ is monotone and $\alpha$-weakly DR-submodular, if we set $\mathcal{O}(\frac{1}{\sqrt{T}})$, the subset $S$ output by Algorithm 3 satisfies:

$$\mathbb{E}(f(S)) \geq \left(\frac{\alpha^2}{1+\alpha^2}\right) f(S^*) - \mathcal{O}(\frac{1}{\sqrt{T}}),$$

where $S^*$ is the optimal solution of problem (1), which implies that after $\mathcal{O}(1/\epsilon^2)$ iterations, the subset output by our proposed Algorithm 1 can attain $(\frac{\alpha^2}{1+\alpha^2})OPT - \epsilon$ where $OPT$ is the maximum value of problem (1). Similarly, when $f$ is $(\gamma, \beta)$-weakly monotone submodular, if we set $\mathcal{O}(\frac{1}{\sqrt{T}})$, the subset $S$ output by Algorithm 3 satisfies:

$$\mathbb{E}(f(S)) \geq \left(\frac{\gamma^2}{\beta + \beta(1-\gamma) + \gamma^2}\right) f(S^*) - \mathcal{O}(\frac{1}{\sqrt{T}}),$$

which also means that, after $\mathcal{O}(\frac{1}{\epsilon^2})$ iterations, Algorithm 1 also can achieve $\left(\frac{\gamma^2}{\beta + \beta(1-\gamma) + \gamma^2}\right)OPT - \epsilon$. Note that, during the process of $\mathcal{O}(\frac{1}{\epsilon^2})$ iterations, Algorithm 2 only requires evaluating the set function $\mathcal{O}(\frac{1}{\epsilon^2})$ times if we set the step size $L = \mathcal{O}(1)$.

In the following part, we prove the Theorem 9. At first, we give a *easy-to-verify* lemma about the upper bound of gradients of our proposed Multinoulli Extension $F$.

**Lemma 5.** *Given a monotone set function $f : 2^{\mathcal{V}} \to \mathbb{R}_+$, if we denote the maximum marginal value of $f$ as $M_f = \max_{S \subseteq \mathcal{V}, e \in \mathcal{V} \setminus S} \left( f(e|S) \right)$, we have that, for any $k \in [K]$, $m \in [n_k]$,*

$$\left| \frac{\partial F}{\partial p_k^m}(\mathbf{p}_1, \ldots, \mathbf{p}_K) \right| \leq \bar{B} M_f,$$

*where $(\mathbf{p}_1, \ldots, \mathbf{p}_K) \in \prod_{k=1}^K \Delta_{n_k}$ and $\bar{B} = \max_{k=1}^K B_k$ is the maximum budget over the $K$ communities $\{\mathcal{V}_1, \ldots, \mathcal{V}_K\}$. Similarly, we also can infer that the gradient estimation Eq.(31) also can be bounded by $\bar{B} M_f$, i.e., for any positive integer $L$,*

$$\left| \frac{\widehat{\partial F}}{\partial p_k^m}(\mathbf{p}_1, \ldots, \mathbf{p}_K) \right| \leq \bar{B} M_f.$$

With this Lemma 5, we then prove the Theorem 9.

*Proof.* Let $S^*$ denote the optimal subset of problem 1. Then, from the Line 6 in Algorithm 3, we know that

$$\left\| \mathbf{x}(t+1) - \sum_{k=1}^K \frac{\mathbf{1}_{S^* \cap \mathcal{V}_k}}{B_k} \right\|_2^2 \leq \left\| \mathbf{y}(t+1) - \sum_{k=1}^K \frac{\mathbf{1}_{S^* \cap \mathcal{V}_k}}{B_k} \right\|_2^2$$

$$= \left\| \mathbf{x}(t+1) + \eta \widehat{\nabla} F(\mathbf{x}(t)) - \sum_{k=1}^K \frac{\mathbf{1}_{S^* \cap \mathcal{V}_k}}{B_k} \right\|_2^2$$

$$= \left\| \mathbf{x}(t) - \sum_{k=1}^K \frac{\mathbf{1}_{S^* \cap \mathcal{V}_k}}{B_k} \right\|_2^2 + 2\eta \left\langle \widehat{\nabla} F(\mathbf{x}(t)), \mathbf{x}(t) - \sum_{k=1}^K \frac{\mathbf{1}_{S^* \cap \mathcal{V}_k}}{B_k} \right\rangle + \eta^2 \left\| \widehat{\nabla} F(\mathbf{x}(t)) \right\|_2^2$$

$$\leq \left\| \mathbf{x}(t) - \sum_{k=1}^K \frac{\mathbf{1}_{S^* \cap \mathcal{V}_k}}{B_k} \right\|_2^2 + 2\eta \left\langle \widehat{\nabla} F(\mathbf{x}(t)), \mathbf{x}(t) - \sum_{k=1}^K \frac{\mathbf{1}_{S^* \cap \mathcal{V}_k}}{B_k} \right\rangle + \eta^2 n \bar{B}^2 M_f^2,$$

where the first inequality follows from $\sum_{k=1}^K \frac{\mathbf{1}_{S^* \cap \mathcal{V}_k}}{B_k} \in \prod_{k=1}^K \Delta_{n_k}$ and the final inequality comes from Lemma 5.

As a result, we have that

$$\mathbb{E} \left\| \mathbf{x}(t+1) - \sum_{k=1}^K \frac{\mathbf{1}_{S^* \cap \mathcal{V}_k}}{B_k} \right\|_2^2 \leq \mathbb{E} \left\| \mathbf{x}(t) - \sum_{k=1}^K \frac{\mathbf{1}_{S^* \cap \mathcal{V}_k}}{B_k} \right\|_2^2 + 2\eta \mathbb{E} \left\langle \widehat{\nabla} F(\mathbf{x}(t)), \mathbf{x}(t) - \sum_{k=1}^K \frac{\mathbf{1}_{S^* \cap \mathcal{V}_k}}{B_k} \right\rangle + \eta^2 n \bar{B}^2 M_f^2$$

$$= \mathbb{E} \left\| \mathbf{x}(t) - \sum_{k=1}^K \frac{\mathbf{1}_{S^* \cap \mathcal{V}_k}}{B_k} \right\|_2^2 + 2\eta \mathbb{E} \left( \left\langle \mathbb{E} \left( \widehat{\nabla} F(\mathbf{x}(t)) \big| \mathbf{x}(t) \right), \mathbf{x}(t) - \sum_{k=1}^K \frac{\mathbf{1}_{S^* \cap \mathcal{V}_k}}{B_k} \right\rangle \right) + \eta^2 n \bar{B}^2 M_f^2$$

$$= \mathbb{E} \left\| \mathbf{x}(t) - \sum_{k=1}^K \frac{\mathbf{1}_{S^* \cap \mathcal{V}_k}}{B_k} \right\|_2^2 + 2\eta \mathbb{E} \left( \left\langle \nabla F(\mathbf{x}(t), \mathbf{x}(t) - \sum_{k=1}^K \frac{\mathbf{1}_{S^* \cap \mathcal{V}_k}}{B_k} \right\rangle \right) + \eta^2 n \bar{B}^2 M_f^2.$$

$$(32)$$

Then, if **1):** the set function $f : 2^{\mathcal{V}} \to \mathbb{R}_+$ is monotone and $\alpha$-weakly DR-submodular, from Theorem 8 and Eq.(32), we have that

$$\mathbb{E} \left\| \mathbf{x}(t+1) - \sum_{k=1}^K \frac{\mathbf{1}_{S^* \cap \mathcal{V}_k}}{B_k} \right\|_2^2 \leq \mathbb{E} \left\| \mathbf{x}(t) - \sum_{k=1}^K \frac{\mathbf{1}_{S^* \cap \mathcal{V}_k}}{B_k} \right\|_2^2 - 2\eta \mathbb{E} \left( \left\langle \nabla F(\mathbf{x}(t), \sum_{k=1}^K \frac{\mathbf{1}_{S^* \cap \mathcal{V}_k}}{B_k} - \mathbf{x}(t) \right\rangle \right) + \eta^2 n \bar{B}^2 M_f^2$$

$$\leq \mathbb{E} \left\| \mathbf{x}(t) - \sum_{k=1}^K \frac{\mathbf{1}_{S^* \cap \mathcal{V}_k}}{B_k} \right\|_2^2 - 2\eta \left( \alpha f(S^*) - (\alpha + \frac{1}{\alpha}) F(\mathbf{x}(t)) \right) + \eta^2 n \bar{B}^2 M_f^2.$$

As a result, we have that

$$2\eta \sum_{t=1}^{T} \left( \alpha f(S^*) - (\alpha + \frac{1}{\alpha})F(\mathbf{x}(t)) \right) \leq \mathbb{E} \left\| \mathbf{x}(T+1) - \sum_{k=1}^{K} \frac{\mathbf{1}_{S^* \cap \mathcal{V}_k}}{B_k} \right\|_2^2 - \mathbb{E} \left\| \mathbf{x}(1) - \sum_{k=1}^{K} \frac{\mathbf{1}_{S^* \cap \mathcal{V}_k}}{B_k} \right\|_2^2 + T\eta^2 n \bar{B}^2 M_f^2.$$

Due to that $\mathbb{E}(f(S_t)) = \mathbb{E}(F(\mathbf{x}(t)))$ from Algorithm 3, we can infer that

$$\mathbb{E}(f(S)) \geq \frac{\sum_{t=1}^{T} \mathbb{E}(F(\mathbf{x}(t)))}{T}$$

$$\geq \frac{\alpha^2}{1+\alpha^2} f(S^*) - \frac{\mathbb{E} \left\| \mathbf{x}(T+1) - \sum_{k=1}^{K} \frac{\mathbf{1}_{S^* \cap \mathcal{V}_k}}{B_k} \right\|_2^2}{2\eta T(\alpha + \frac{1}{\alpha})} - \eta \frac{n\bar{B}^2 M_f^2}{2(\alpha + \frac{1}{\alpha})}$$

$$\geq \frac{\alpha^2}{1+\alpha^2} f(S^*) - \frac{K}{\eta T} - \eta \frac{n\bar{B}^2 M_f^2}{4},$$

where the final inequality follows from $\alpha + \frac{1}{\alpha} \geq 2$ and $\left\| \mathbf{x}(T+1) - \sum_{k=1}^{K} \frac{\mathbf{1}_{S^* \cap \mathcal{V}_k}}{B_k} \right\|_2^2 \leq 2(\|\mathbf{x}(T+1)\|_2^2 + \left\| \sum_{k=1}^{K} \frac{\mathbf{1}_{S^* \cap \mathcal{V}_k}}{B_k} \right\|_2^2) \leq 4K$.

Furthermore, **2):** if the set function $f : 2^{\mathcal{V}} \to \mathbb{R}_+$ is monotone and $(\gamma, \beta)$-weakly submodular, we also can show that, from Theorem 8,

$$2\eta \sum_{t=1}^{T} \left( \gamma^2 f(S^*) - (\beta + \beta(1-\gamma) + \gamma^2)F(\mathbf{x}(t)) \right) \leq \mathbb{E} \left\| \mathbf{x}(T+1) - \sum_{k=1}^{K} \frac{\mathbf{1}_{S^* \cap \mathcal{V}_k}}{B_k} \right\|_2^2 - \mathbb{E} \left\| \mathbf{x}(1) - \sum_{k=1}^{K} \frac{\mathbf{1}_{S^* \cap \mathcal{V}_k}}{B_k} \right\|_2^2 + T\eta^2 n \bar{B}^2 M_f^2.$$

Finally, we have that Due to that $\mathbb{E}(f(S_t)) = \mathbb{E}(F(\mathbf{x}(t)))$ from Algorithm 3, we can infer that

$$\mathbb{E}(f(S)) \geq \frac{\sum_{t=1}^{T} \mathbb{E}(F(\mathbf{x}(t)))}{T}$$

$$\geq \left( \frac{\gamma^2}{\beta + \beta(1-\gamma) + \gamma^2} \right) f(S^*) - \frac{\mathbb{E} \left\| \mathbf{x}(T+1) - \sum_{k=1}^{K} \frac{\mathbf{1}_{S^* \cap \mathcal{V}_k}}{B_k} \right\|_2^2}{2\eta T(\beta + \beta(1-\gamma) + \gamma^2)} - \eta \frac{n\bar{B}^2 M_f^2}{2(\beta + \beta(1-\gamma) + \gamma^2)}$$

$$\geq \left( \frac{\gamma^2}{\beta + \beta(1-\gamma) + \gamma^2} \right) f(S^*) - \frac{8K}{7\eta T} - \eta \frac{2n\bar{B}^2 M_f^2}{7},$$

where the final inequality follows from $\left\| \mathbf{x}(T+1) - \sum_{k=1}^{K} \frac{\mathbf{1}_{S^* \cap \mathcal{V}_k}}{B_k} \right\|_2^2 \leq 2(\|\mathbf{x}(T+1)\|_2^2 + \left\| \sum_{k=1}^{K} \frac{\mathbf{1}_{S^* \cap \mathcal{V}_k}}{B_k} \right\|_2^2) \leq 4K$, $\beta \geq 1$ and $\beta(1-\gamma) + \gamma^2 \geq \frac{3}{4}$(Lemma B.1 in (Thiery & Ward, 2022)). $\qquad \square$

