# OpenReview forum: "Multinoulli Extension: A Lossless Yet Effective Probabilistic Framework for Subset Selection over Partition Constraints"
_ICML.cc/2025/Conference — ICML 2025 poster_

### Official Review · Reviewer_qooM · 2025-03-14

**Overall Recommendation:** 3

**Summary:**

The paper introduces a novel algorithm called Multinoulli-SCG for solving the subset selection problem under partition constraints, particularly focusing on close-to-submodular objective functions. The core of the Multinoulli-SCG algorithm is an innovative continuous-relaxation framework named Multinoulli Extension(ME). Unlike the traditional multi-linear extension, ME provides a lossless rounding scheme for any set function, not just submodular ones.

**Claims And Evidence:**

The claims made in the submission are supported by clear and convincing evidence

**Essential References Not Discussed:**

No

**Experimental Designs Or Analyses:**

Empirical evaluation is conducted on video summarization, Bayesian A-optimal design, and maximum coverage. It is demonstrated that Multinoulli-SGA and Multinoulli-SCG outperform existing methods in terms of objective value.

**Methods And Evaluation Criteria:**

Proposed methods and evaluation criteria make sense for the problem.

**Other Comments Or Suggestions:**

No

**Other Strengths And Weaknesses:**

The Multinoulli Extension is a novel and interesting framework that provides a fresh perspective on the subset selection problem under partition constraints. The Multinoulli-SCG and Multinoulli-SGA algorithms represent significant improvements over prior continuous algorithms, particularly in terms of query efficiency and parameter-free operation. These advancements make the proposed methods highly practical for real-world applications where computational efficiency and ease of implementation are critical.

However, the paper could benefit from a more thorough analysis of the scalability of the proposed algorithms. As the scalability of the algorithms are improved compared to prior continuous algorithms theoretically, evaluating the algorithms on larger datasets would provide deeper insights into their performance in practice.

**Questions For Authors:**

No

**Relation To Broader Scientific Literature:**

The paper provides a novel continuous-relaxation framework which provides a lossless rounding scheme for any set function. This gives new direction in the literature.

**Theoretical Claims:**

No apparent issue found.

---

> ### Author Rebuttal · Authors · 2025-03-30
>
> Thank you for your detailed and insightful reviews. We sincerely appreciate the time and effort you have dedicated to reviewing our manuscript. Your feedback is invaluable to us. Below, we will respond to the concerns you have raised in **Weaknesses**.
>
> ------------------------
>
> **Weaknesses**:
>
> ---------------------
> > **W1: The paper could benefit from a more thorough analysis of the scalability of the proposed algorithms. As the scalability of the algorithms are improved compared to prior continuous algorithms theoretically, evaluating the algorithms on larger datasets would provide deeper insights into their performance in practice.**
>
> Thank you very much for your constructive suggestion.
>
> Scaling up the proposed algorithms to larger datasets and a wider range of subset selection tasks is indeed one of our long-term goals.  For example, similar to [1,2], exploring how to use distributed architectures to accelerate our algorithms or handle extremely large-scale datasets is something we are actively considering.
>
> Furthermore, we would like to emphasize that the experiments conducted in this paper are highly representative:
>
> First, to demonstrate the scalability of our proposed algorithms, we have tested them on multiple datasets and various tasks, where they consistently show strong performance in terms of both efficiency and effectiveness compared to previous methods.
>
> Second, it is worth noting that in the video summarization task, the scale of data we are dealing with is already significantly larger than that of most known subset selection tasks. For instance, in the **V2** video, which has a duration of 7 minutes and 45 seconds (465 seconds), with 30 frames per second, we have effectively selected 20 representative frames from nearly 13,900 frames.
>
> **References**
>
> [1] Mirzasoleiman, B., Karbasi, A., Sarkar, R., and Krause, A. Distributed submodular maximization. The Journal of Machine Learning Research, 17(1):8330–8373, 2016.
>
> [2] Mokhtari, A., Hassani, H., and Karbasi, A. Decentralized submodular maximization: Bridging discrete and continuous settings. In International conference on machine learning, pp. 3616–3625. PMLR, 2018b.

---

### Official Review · Reviewer_WbSo · 2025-03-15

**Overall Recommendation:** 4

**Summary:**

The paper considers maximization of a monotone close-to-sumodular objective $f$ over a partition matroid, where the notions of approximate submodularity considered are weak DR-submodularity and weak submodularity.  The authors introduce a novel continuous extension for this problem, called Multinoulli Extension (ME). They study its properties showing that it has similar nice properties as the well-known multilinear extension, but with the advantage that it allows lossless rounding on partition matroids for any set function. They also propose a novel variant of the continuous greedy algorithm, called Multinoulli-SCG, adapted to the proposed extension, and which employs the existing path-integrated differential estimator to estimate gradients of the ME. The resulting algorithm matches the best existing approximation guarantees for both classes of functions considered, with less function queries under some settings.  Experiments comparing the proposed method to existing ones are provided on video summarization, bayesian A-optimal design and coverage maximization applications.

**Claims And Evidence:**

The following claims are inaccurate or require more evidence.

1 - Claim: The proposed method uses $O(1/\epsilon^2)$ function evaluations. The best existing method, Distorted Local Search with guessing (Distorted-LS-G), uses $\tilde{O}(1/\epsilon^6)$ and $\tilde{O}(1/\epsilon^3)$  for weakly DR-submodular and weakly submodular functions respectively.

These claims ignore other non-constant parameters of the problem such as the size of the ground set $n$, the rank of the matroid $r$, and the optimal value OPT, which can be larger than $O(1/\epsilon)$ and should not be ignored. Moreover, the $\epsilon$ appearing in the query complexity of Distorted-LS-G is not the same as the one for Multinoulli-SCG. The former is equal to $\epsilon' = \epsilon / OPT$. This should be clarified, e.g., by using a different notation $\epsilon'$ for Distorted-LS-G.

2 - Related claim: The query complexity of the proposed method is better than Distorted-LS-G.

It should be clarified that this is only true under some settings: if $n < r OPT^6/\epsilon^4$ for $\alpha$-weakly DR-submodular functions, and if $n < r OPT^3/\epsilon$ for $(\gamma, \beta)$-weakly submodular functions. Both conditions are likely to hold in practice, but not always. It would be good to discuss also in each of the experiments presented if these conditions hold.

3 - Claim: the number of function evaluations required by the proposed algorithm matches the information-theoretic $O(1/\epsilon^2)$ lower bound given in (Karbasi et al., 2019; Hassani et al., 2020).

This claim is not correct. The lower bound given in (Karbasi et al., 2019; Hassani et al., 2020) is on the number stochastic gradient calls, not on the number of function calls. Either remove this claim, or show that this lower bound also applies to the number of function calls in your setting.

**Essential References Not Discussed:**

The main motivation of introducing the Multinoulli extension, instead of using multilinear extension, is that it allows for simple lossless rounding. To put things more in context, it would be good to mention that contention resolution, one of the rounding schemes for general matroid used with the multilinear extension, can still be used for $\alpha$-weakly DR-submodular functions, but it loses a factor of  $\alpha ( 1 - 1/e)$, as shown in (Gong et al., 2019).

**Experimental Designs Or Analyses:**

Yes I checked all experiments. The experiments are mostly well designed, cover three cases ($\alpha$-weakly DR-submodular, $(\gamma, \beta)$-weakly submodular, and a submodular function). I like also that one of the experiments (maximum coverage) was specifically designed to show the case where Greedy and Multinoulli-SGA get stuck in a local minimum.

Issues to address:

- Residual Greedy and Multinoulli-SCG are repeated 10 times, while Multinoulli-SGA and Distorted-LS-G are repeated 5 times. Using different number of repetition for different methods is a not a fair comparison.

- In the video summarization experiments, the DPP objective is defined as $f(S) = \det(I + X_S)$. The DPP objective does not typically include identity, a small diagonal perturbation can be added to ensure
strict positive definiteness of the kernel matrix. Is there a missing scaling factor here, i.e., $f(S) = \det(\delta I + X_S)$ for some small $\delta > 0$?

Other minor issues/suggestions:
- Include standard deviations for the reported results in Table 3 & 4, not just the average.
- Include what are the parameters $\alpha$ and $\gamma, \beta$ in each experiment.
- For the maximum coverage experiment (Appendix B.2):
	- clarify what you mean by Residual Greedy oscillates between optimal solution and local maximum, is it across different runs?
	- use a different notation to denote the small weight for $A_i$'s to avoid confusion with $\epsilon$ of the error in the optimization guarantee.
	- use a different table to present the results for the Bayesian optimal design experiment.

**Methods And Evaluation Criteria:**

Yes, the proposed method makes sense and builds on well known techniques.
The evaluation based on approximation guarantee and query complexity are also the usual evaluation criteria for this problem. Applications are also standard applications for the problem.

**Other Comments Or Suggestions:**

- Clarify in the problem setup that the partition groups are assumed to be known, as opposed to the usual assumption in submodular optimization over matroids, that we have access to the matroid via oracle calls only.

- As mentioned, the notation used is overly cumbersome. Simplifying it would significantly improve the readability of the paper. For example, why do you use $\hat{k}$  and $\hat{b}$ instead of simply $k$ and $b$ in the definition of the ME?

- The definition of $\Delta_m$ is not the standard definition of an $m$-dimensional simplex (the standard definition has $\sum_i x_i = 1$). Similarly, calling $p_k$ a "Multinoulli distribution" is also inaccurate if $\sum_m p_k^m \not = 1$.  Either use different names for $\Delta_m$  and $p_k$ or use the standard definitions for both and define $p_k$ to be the probability on $\mathcal{V}_k \cup \\{v_0\\}$, where $v_0$ is a dummy element that can be added to all groups, which corresponds to not selecting any element in the group.  I recommend the latter option, as it would simplify the notation elsewhere in the paper too.

- Discuss the motivation for using the path-integrated differential estimator, instead of directly estimating the gradient itself.

- Add a discussion on how challenging would it be to generalize the presented results to general matroids.

- Explicitly explain in the main text the simple way to round $x(T + 1)$ based on the ME.

- Use a different notation, e.g.,  $\epsilon'$, instead of $\epsilon$ for Distorted-LS-G.

- Shorten the abstract. According to instructions, it should be ideally between 4-6 sentences long.

- Use group or block to refer to $\mathcal{V}_k$ instead of community, as these the standard names used in the literature.

- One line 301-303, 1st col: the statement there is inaccurate; the inequality $\langle y, \nabla G(x) \rangle \geq G(y) - G(x)$ only holds for $y \geq x$ for the multilinear extension.

- Restate theorems in the appendix for convenience for the reader

- In Remark 7, Algorithm 2 should be Algorithm 1.

**Other Strengths And Weaknesses:**

Already highlighted above.

Other weaknesses:
- The code to reproduce experiments is not provided

**Questions For Authors:**

1 - Would using a similar implementation of the Hessian-vector product as the one described in Section 4.1 of (Karbasi et al., 2019) also lead to better computational and memory complexity for your proposed algorithm? If yes, what is the resulting complexity?

If this can lead to an improved complexity for the proposed algorithm, for example by reducing the number of functions calls by a factor $n$ (as in the case in (Karbasi et al., 2019)), it would strengthen the results of this paper.

2- Why is it an issue to select the same element when rounding as discussed in Remark 9? Isn't the resulting set simply a union of the elements obtained. What is the motivation for using a rounding-without-replacement method?

3- Why can we assume that $x(T + 1)$ is on the boundary of the domain (see details in "Theoretical claims")?

**Relation To Broader Scientific Literature:**

Existing methods for solving the problem considered either have a worse approximation guarantee, or have a worse query complexity under some settings (see Tables 1 & 2 in the paper and "Claims and Evidence" above).

The proposed Multinoulli extension has similar nice properties as the well-known multilinear extension, but with the advantage that it allows lossless rounding on partition matroids for any set function. Its disadvantage is that unlike the multilinear extension, the ME is specific to partition matroids. Nevertheless, I think this might inspire the development of other continuous extensions that incorporates constraints within the extension to allow for lossless rounding.

The proposed algorithm is a variant of the continuous greedy algorithm, adapted to the proposed extension, and which uses the existing path-integrated differential estimator to estimate gradients of the ME.

**Theoretical Claims:**

I verified the proofs of Theorems 1 and 2 but did not check those of Theorems 3–6.

The main issue is that the cumbersome notation makes the proofs difficult to follow. I strongly encourage the authors to simplify the notation.

Another concern is the assumption in Theorem 6 that $p_k$ is at the boundary of the domain, i.e., $\\|p_k\\|_1 = 1$ for all $k$, which is not well justified. The input of Algorithm 2 is expected to satisfy this assumption. Algorithm 2 is used to round the solution $x(T + 1)$ at the end of Algorithm 1. I don't see why $x(T + 1)$ is necessarily at the boundary of the domain. When the function $f$ is monotone, the gradient of the ME is non-negative, but not necessarily the estimator $g(t)$, so it's possible that $|S(t) \cap V_k| < B_k$ in some cases, and thus $x(T + 1)$ won't be at the boundary.

Other minor typos/issues:
- In the proof of Theorem 1, in the third equality in Eq. (13) (line 969-970), the third sum should be over $e_k^{\hat{b}_1}$.
- In the proof of Theorem 1, $e(X, Y, p_k, \hat{p}_k)$ should be just a function of $X$ and $p_k$.
- In the proof of Theorem 2, Eq. (20) is missing sums over $k$ and $b$.

---

> ### Author Rebuttal · Authors · 2025-03-30
>
> Thank you for your detailed and insightful reviews. We sincerely appreciate the time and effort you have dedicated to reviewing our manuscript. Your feedback is invaluable to us. In the following, we will address the concerns you have raised in **Questions**.
>
> ---
>
> **Questions**:
>
> ---
>
> >**Q3:Why can we assume that $x(T + 1)$ is on the boundary of the domain (see details in "Theoretical claims")?**
>
> Thank you very much for your question.
>
> Firstly, this paper primarily focuses on monotone set functions $f$. Secondly, as noted in Theorem 1 (2), we have shown that when $f$ is monotone, its multinoulli extension is also monotone. Specifically, for any two feasible vectors $(x_{1},\dots,x_{K})$ and $(y_{1},\dots,y_{K})$, if $x_{i}\ge y_{i},\forall i\in[K]$, then $F(x_{1},\dots,x_{K})\ge F(y_{1},\dots,y_{K})$. Thus, for any feasible vector
>
> $(x_{1},\dots, x_{K})$, we have  $F(\frac{x_{1}}{u_{1}},\dots,\frac{x_{K}}{u_{K}})\ge F(x_{1},\dots, x_{K})$
> where $u_{i}=||x_{i}||_1$ for any $i\in[K]$.
>
> Note that ($\frac{x_{1}}{u_{1}},\dots,\frac{x_{K}}{u_{K}}$) lies on the boundary of the constrained domain $\prod_{k=1}^{K}\Delta_{n_{k}}$.
>
> Therefore, even if the output of Algorithm 1 is not on the boundary, we can first normalize this output vector to the boundary and then use Algorithm 2 to round the normalized output vector. This process will not decrease the function value for the monotone set objective function $f$. This is  why we assume $x(T+1)$ is on the boundary-because we can always increase the function value by normalizing the output vector.
>
> > **Q2: Why is it an issue to select the same element when rounding as discussed in Remark 9? Isn't the resulting set simply a union of the elements obtained. What is the motivation for using a rounding-without-replacement method?**
>
> Thank you very much for your question. we will explain Q2 with a simple example.
>
> Considering $V=[3]$ and we will select at most 2 elements from $V$. Given $p=(1/3,1/3,1/3)$, if we use the definition of multinoulli extension to round
> $p$, we might end up selecting element 3 twice. This would result in a final subset containing only element 3. Since this paper focuses on monotone set functions $f$，we naturally know that randomly adding element 1 or element 2 to the resulting subset (which contains only element 3) will increase the function value without violating the constraints. This illustrates the issue of selecting the same element over multiple selections.
>
> To address this, we propose a rounding-without-replacement method. This ensures that the final resulting subset contains the maximum number of distinct elements, thereby avoiding the pitfalls of multiple selections of the same element and maximizing the function value more effectively.
> > **Q1: Would using a similar implementation of the Hessian-vector product as the one described in Section 4.1 of (Karbasi et al., 2019) also lead to better computational and memory complexity for your proposed algorithm? If yes, what is the resulting complexity?**
>
> Thank you very much for your suggestion. This is a very good question.
>
> We have also considered the same question. However, we noticed that in Section 5 of (Karbasi et al., 2019), when dealing with discrete submodular maximization, they did not use the Hessian-vector product described in Section 4.1. Instead, they employed a Hessian estimation method similar to the one used in our paper. Therefore, we suspect that the Hessian estimation method in Section 4.1 of (Karbasi et al., 2019) may not be directly applicable to our multinoulli extension.
>
> The primary issue with using the Hessian-vector product described in Section 4.1 of (Karbasi et al., 2019) is that it involves computing $\nabla\log(p(z;y))$, which may require differentiating some $log(p)$, where $p\in(0,1)$  is a parameter of interest.  Note that $(log(p))^{'}=\frac{1}{p}$ and $\frac{1}{p}$ is unbounded on the interval $(0,1)$. Therefore, we thimnk directly applying the Hessian-vector product from Section 4.1 of (Karbasi et al., 2019) to our multinoulli extension might violate the bounded gradient and smoothness assumptions in (Karbasi et al., 2019).
>
> ----
>
> **Other Comments Or Suggestions:**
>
> ---
>
> Thank you very much for your comments and suggestions. We will carefully address all the proposals and correct any typos in the final version.
>
> ----
>
> **Claims And Evidence:**
>
> ----
> * We will add a detailed remark after **Remark on Table 2**, specifically after line 153, to discuss the comparison of query complexities between multinoulli-SCG and distorted local search under different values of $n$. Moreover, in this additional remark, we will also explicitly point out that the query complexity of Distorted-LS-G uses a different error term $\epsilon^{'}$, which is related to $\epsilon$ of Multinoulli-SCG by the transformation $\epsilon' = \epsilon / OPT$.
> * We will **remove** the inappropriate claim from "Our Contributions" in RHS in line 94, i.e.," match the information-theoretic lower"

---

> > ### Comment · Reviewer_WbSo · 2025-04-03
> >
> > Thank you for the clarifications! I recommend including these explanations in the paper.

---

### Official Review · Reviewer_ua4p · 2025-03-21

**Overall Recommendation:** 3

**Summary:**

This paper considers the problem of subset selection subject to partition constraints. The objective is not fully submodular, but instead displays some degree of submodularity, e.g. is weakly submodular. Existing work on this problem relies on distorted local search methods, but these works have some shortcomings because they rely on unknown parameters such as the weak submodularity ratio, and have high query complexity (depending on a paramter $\epsilon$). To address these issues, this paper proposes the Multinoulli extension, which is a continuous extension of a submodular set function like the multilinear extension, but has some advantages for weakly-submodular objectives. The paper further proposes and analyzes algorithms for their problem using this extension.

**Claims And Evidence:**

I did not notice any problematic claims.

**Essential References Not Discussed:**

It was mentioned, and a citation was provided, that the multilinear extension has issues when being used for weakly submodular objectives. Because this is an important problem that motivated the paper, I think that they should go in to further depth in the main paper about why this is the case.

**Experimental Designs Or Analyses:**

No issues

**Methods And Evaluation Criteria:**

Yes

**Other Comments Or Suggestions:**

No

**Other Strengths And Weaknesses:**

Strengths
- Contributes to the area of weakly submodular optimization, which is of interest to the ML community
- Proposes a new continuous extension of submodular functions, which may be of use in other problems.

Weaknesses
- The multinoulli extension is advantageous specifically for weakly submodular functions, and so doesn't seem useful to those who are primarily concerned with submodular objectives.

**Questions For Authors:**

- Could you give more detail about the issues with the multilinear extension being used for weakly submodular functions?

**Relation To Broader Scientific Literature:**

This paper is of interest to those in the submodular optimization community. In particular, there are many papers exploring weakly-submodular optimization, and since this paper proposed a new continuous extension, members of that community could potentially also build upon the new extension.

**Theoretical Claims:**

I did not thoroughly check theoretical claims but I did not notice any problems.

---

> ### Author Rebuttal · Authors · 2025-03-30
>
> Thank you very much for your careful reading and constructive feedback. We are grateful for the time and effort you have dedicated to reviewing our manuscript. In what follows, we will address some of your concerns in **Questions** and **Weaknesses**.
>
> --------------
>
> **Weaknesses**:
>
> --------------
>
> > **W1: The multinoulli extension is advantageous specifically for weakly submodular functions, and so doesn't seem useful to those who are primarily concerned with submodular objectives.**
>
> Thank you very much for your question.
>
> First, it is important to note that when  $\alpha=\gamma=\beta=1$, the $(\gamma,\beta)$-weakly submodular or $\alpha$-weakly DR-submodular functions considered in this paper will degenerate into standard submodular functions. Thus, the class of set functions we study is a broader category that includes standard submodular functions as a special case.
>
> Moreover, when $\alpha=\gamma=\beta=1$, the approximation ratio $(1-e^{-\alpha})$ or $(\frac{\gamma^{2}(1-e^{-(\beta(1-\gamma)+\gamma^2)})}{\beta(1-\gamma)+\gamma^2})$ yielded by **Multinoulli-SCG** will equal $(1-1/e)$. This means that our algorithm can guarantee an optimal $(1-1/e)$-approxiation for submodular maximization over partition constraints, which is consistent with the state-of-the-art algorithms for submodular maximization problems.
>
> As a result, our proposed **Multinoulli-SCG**  algorithm not only offers advantages for weakly submodular functions but also guarantees the same approximation performance as the best existing algorithms for submodular maximization.
>
> --------------
>
> **Questions**:
>
> --------------
> >**Q1:Could you give more detail about the issues with the multi-linear extension being used for weakly submodular functions?**
>
> Thank you very much for your question.
>
> At first, it is important to note that we have provided a detailed introduction to the multi-linear extension in Appendix A.1 (lines 680-714), where we also discussed the main challenges in applying the multi-linear extension to weakly submodular functions
>
> Here, we will briefly restate the key issue.
>
> Before that, we call the defintion of multi-linear extension, that is,
>
> For a set function $f:2^{V}\rightarrow R_{+}$ where $|V|=n$ and $V:=[n]$, we define its multi-linear extension as
> 	\begin{equation}
> 		G(x)=\sum_{\mathcal{A}\subseteq V}\Big(f(\mathcal{A})\prod_{a\in\mathcal{A}}x_{a}\prod_{a\notin\mathcal{A}}(1-x_{a})\Big)=E_{\mathcal{R}\sim x}\Big(f(\mathcal{R})\Big),
> 	\end{equation} where $x=(x_{1},\dots,x_{n})\in [0,1]^{n}$ and $\mathcal{R}\subseteq V$ is a random set that contains each element $a\in V$ independently with probability $x_{a}$ and excludes it with probability $1-x_{a}$. We write $\mathcal{R}\sim x$ to denote that $\mathcal{R}\subseteq V$ is a random set sampled according to $ x$.
>
> From the previous definition, we can view multi-linear extension $G$ at any point $x\in[0,1]^{n}$ as the expected utility of independently selecting each action $a\in V$ with probability $x_{a}$.  With this tool, we can cast the previous discrete subset selection problem (1) into a continuous maximization which learns the independent probability for each element $a\in V$, that is, we consider the following continuous optimization:
> \begin{equation}
> 	\max_{x\in[0,1]^{n}} G(x),\ \ \text{ s.t.}\ \  \sum_{a\in V_{k}}x_{a}\le B_{k},\forall k\in[K]
> \end{equation}where $G(x)$ is the multi-linear extension of $f$.
>
> It is important to note that, if we round any point $x$ included into the constraint of the previous multi-linear maximization problem by the definition of multi-linear extension,  there is a certain probability that the resulting subset  will **violate** the partition constraint of the original discrete subset selection problem.
>
> -------------------------------
> > **Q2:  It was mentioned, and a citation was provided, that the multi-linear extension has issues when being used for weakly submodular objectives. Because this is an important problem that motivated the paper, I think that they should go in to further depth in the main paper about why this is the case.**
>
> Thank you very much for your insightful question.
>
> Indeed, the limitation of the multi-linear extension when applied to weakly submodular functions is a significant motivation for our work. Due to space constraints, we have detailed these limitations in Appendix A.1, where we discuss the specific challenges in using the multi-linear extension for weakly submodular functions, as shown in the answers in **Q1**.
>
> Given the importance of this issue, we will add a small subsection at the end of Section 3 to further discuss the limitations of the multi-linear extension and how our proposed multinoulli extension overcomes these challenges. This will provide a more detailed explanation directly within the main paper, addressing your concern more comprehensively.

---

### Decision · Program_Chairs · 2025-05-01

**Decision:**

Accept (poster)

**Comment:**

The paper consider the problem of maximizing a weakly submodular function subject to a partition constraint and its application to subset selection. The new algorithm attains a similar approximation factor as previous work by Thiery and Ward while having an incomparable runtime trading off the dependence on the ground set size and the error in the approximation factor. Note that the previous work is for all matroid whereas the current paper is only for partition matroid. In experiments, the new method seems to have a slight improvement in the quality while using fewer queries. All reviewers are positive about the paper. They appreciate the practical improvement from the new algorithm as well as the new theoretical perspective with a new continuous extension (though this extension is only for a partition matroid).